# A gated hydrophobic funnel within BAX binds bioactive lipids to potentiate pro-apoptotic function

Jesse D. Gelles [1,2,3], Yiyang Chen[1,2,3,4], Mark P. A. Luna-Vargas[1,2,3], Ariele Viacava Follis[5], Md Abdullah Al Noman [1,2,3], Md Kabir[2,3,6], Stella G. Bayiokos [1,2,3,4], Jarvier N. Mohammed [1,2,3], Tara M. Sebastian[1,2,3], Ngoc Dung Pham[6], Yi Shi[6], Jian Jin [2,3,6], Richard W. Kriwacki [5,7] & Jerry Edward Chipuk [1,2,3,4,8,9] ✉

Mitochondria maintain a distinct biochemical environment that cooperates with pro-apoptotic BAX and BH3-only proteins (e.g., BIM) to promote mitochondrial outer membrane permeabilization (MOMP), the key event to initiate physiological and pharmacological forms of apoptosis. The sphingosine-1-phosphate metabolite 2-*trans*-hexadecenal (2t-hexadecenal) is a bioactive lipid that supports BAX-dependent MOMP. Using integrated structural and computational approaches, we determine that 2t-hexadecenal binds within a distinct, dynamic region—a hydrophobic cavity formed by core-facing residues of α5, α6, and gated by α8—we now term the "BAX actuating funnel" (BAF). Complementary biochemical and biophysical techniques reveal that 2t-hexadecenal non-covalently interacts with the BAF and cooperates with BIM to stimulate intramolecular activation of monomeric BAX prior to membrane association. BAX α8 mobility and proline 168-mediated allostery are critical determinants for 2t-hexadecenal synergy with BAX and BIM, as is alkenal length to stimulate BAF function. Collectively, this work imparts detailed molecular insights into how pro-apoptotic BCL-2 proteins and bioactive lipids non-covalently cooperate to initiate the mitochondrial pathway of apoptosis with implications for biological and therapeutic regulation.

Developmental, homeostatic, and pharmacological pro-apoptotic signals converge by engaging the BCL-2 family of proteins to induce BAX-dependent mitochondrial outer membrane permeabilization (MOMP) and apoptosis[1]. Despite sequence and structural similarity with multiple globular BCL-2 family proteins, BAX is distinct in that it

converts from an inactive cytosolic monomer to a pore-forming oligomer at the outer mitochondrial membrane (OMM). Upon transient triggering with direct activator BH3-only proteins (e.g., BIM, BID), BAX undergoes a series of intramolecular rearrangements and structural refoldings that ultimately result in translocation to the OMM,

[1]Laboratory of Mitochondrial Biology in Human Health and Disease, Icahn School of Medicine at Mount Sinai, New York, NY, USA. [2]Department of Oncological Sciences, Icahn School of Medicine at Mount Sinai, New York, NY, USA. [3] The Mount Sinai Tisch Cancer Center, Icahn School of Medicine at Mount Sinai, New York, NY, USA. [4]The Graduate School of Biomedical Sciences, Icahn School of Medicine at Mount Sinai, New York, NY, USA. [5]Department of Structural Biology, St. Jude Children's Research Hospital, Memphis, TN, USA. [6]Department of Pharmacological Sciences, Icahn School of Medicine at Mount Sinai, New York, NY, USA. [7]Department of Microbiology, Immunology and Biochemistry, University of Tennessee Health Sciences Center, Memphis, TN, USA. [8]Department of Dermatology, Icahn School of Medicine at Mount Sinai, New York, NY, USA. [9]The Diabetes, Obesity, and Metabolism Institute, Icahn School of Medicine at Mount Sinai, New York, NY, USA. ✉e-mail: jerry.chipuk@mssm.edu

oligomerization, and MOMP[2–7]. An important structural event during this process is α9 helix mobilization from its residence within the BAX BC groove, which simultaneously supports OMM translocation, BAX: BAX interactions, and propagation of the activation process leading to proteolipid pore formation. We refer to this series of structural rearrangements as the BAX activation continuum, which is subdivided into an activation phase (triggering) and functionalization phase (pore formation)[8].

There are two general requirements for potent BAX activation: protein-protein interactions to trigger the monomer, and protein-lipid interactions to initiate and stabilize refolding of BAX into multimeric conformers. These requirements are proposed to occur in distinct phases of the activation continuum, and most insights into protein-lipid interactions focus on how BAX interacts within the OMM to form pore-like structures[9–12]. This has led to a modern conceptualization in which mitochondrial membranes actively cooperate with the BCL-2 family to control cell death commitment through the regulation of mitochondrial shape and curvature, dynamics machinery, lipid milieu, and intra-organellar communication[13–15]. Notably, mitochondrial endoplasmic reticulum contact sites (MERCS) maintain lipid homeostasis between the organelles and supply lipids that are critical for MOMP[16].

One such example is the terminal end-product of the sphingolipid pathway 2-trans-hexadecenal (2t-hexadecenal)—which is required for BAX-mediated MOMP following triggering by BIM or BID[17]. 2t-hexadecenal is formed by the irreversible cleavage of sphingosine-1-phosphate (S1P) and is further metabolized to hexadecenoyl-CoA and palmitoyl-CoA for fatty acid synthesis pathways[18,19]. The resident enzymes responsible for generating and catabolizing 2t-hexadecenal are enriched at MERCS, and genetically modulating 2t-hexadecenal levels alters sensitivity to human BAX in yeast[20]. Previously, the requirement for 2t-hexadecenal in BAX-mediated MOMP was described and subsequent investigations have proposed a covalent mechanism by which 2t-hexadecenal alkylates BAX at the membrane[17,21,22]. However, the non-covalent role of 2t-hexadecenal to cooperate with BH3-only proteins and promote BAX activation prior to membrane association remains an unanswered question proposed by the original investigation.

Here, we utilize biochemical, biophysical, structural, and computational approaches to systematically demonstrate that 2t-hexadecenal directly actuates BAX through non-covalent interactions at a previously undefined region—a funnel-shaped hydrophobic cavity formed by core-facing residues of α5, α6, and gated by α8, which we term the BAX Actuating Funnel, or BAF. Our results suggest that BIM-mediated BAX triggering mobilizes α8, making the BAF accessible, and that binding of 2t-hexadecenal promotes BAX functionalization. Furthermore, we identify chemical and structural determinants underlying 2t-hexadecenal:BAX interactions and reveal that mutation of proline 168 in the loop between α8 and α9 allosterically deforms the BAF and subsequently disrupts the function of 2t-hexadecenal. Collectively, this model advances our understanding of BAX structure-function relationships by characterizing a non-covalent protein-lipid interaction responsible for stimulating monomeric BAX activation and identifies a previously under-appreciated regulatory domain for both cell biology and therapeutic investigations.

## Results

### 2t-hexadecenal directly activates BAX through non-covalent interactions

Previous work demonstrated that 2t-hexadecenal (henceforth, hexadecenal) was required for potent BAX-mediated MOMP and that sphingolipid precursors are supplied to mitochondria via interactions with heterotypic membranes[17]. To assess whether ectopic hexadecenal exposure could engage BAX activation in cellulo, we treated SV40-transformed mouse embryonic fibroblasts (MEFs) with increasing concentrations of hexadecenal and measured the apoptotic response using our real-time multi-visitation microscopy technique, SPARKL[23]. Ectopic hexadecenal ("2t16") did not induce cell death until high concentrations were utilized (Figs. 1A and S1A). We reasoned that any pro-apoptotic signaling may have been mitigated by the repertoire of anti-apoptotic BCL-2 family proteins, and indeed, co-treatment with the BH3-mimetic ABT-737 revealed an apoptotic phenotype in response to ectopic hexadecenal (Fig. 1A). To determine if the apoptotic response required triggering by direct activators, we utilized Bim[–/–]Bid[–/–] double knockout (DKO) MEFs and observed a loss of sensitivity to hexadecenal at lower concentrations co-treated with ABT-737, suggesting that BAX activation by direct activators was part of the underlying mechanism (Fig. 1B). Finally, we replicated this experiment in Bax[–/–]Bak[–/–] DKO MEFs and observed no cell death in response to hexadecenal indicating a requirement for the effector proteins (Fig. 1C). Of note, high-concentration hexadecenal exhibited apoptosis as a single treatment and in conjunction with ABT-737 in both wildtype (WT) and Bim[–/–]Bid[–/–] DKO MEFs, but not in Bax[–/–]Bak[–/–] DKO MEFs, potentially indicating that this concentration of hexadecenal was directly activating BAX. Collectively, these data may suggest that hexadecenal induces apoptotic cell death by acting on effector BCL-2 family proteins.

There are reports that ectopic hexadecenal can form adducts with DNA and generate oxidative stress resulting in apoptosis, and we could not eliminate the possibility that ectopic hexadecenal was targeting additional substrates or stress pathways[24,25]. Therefore, we interrogated whether hexadecenal directly promoted BAX-mediated pore formation by utilizing recombinant BAX protein and large unilamellar vesicles (LUVs), which are biochemically-defined liposomes that mimic the major lipid composition of the OMM[9], and assessed BAX activation by measuring LUV permeabilization. While recent studies have incorporated hexadecenal directly into the formulation of LUVs[22], we aimed to investigate hexadecenal-mediated BAX activation without altering the biochemistry of the LUV membrane. BAX treated with hexadecenal demonstrated dose-dependent activation and LUV permeabilization (Fig. S1B). Importantly, ectopic hexadecenal alone did not disrupt or cause leakage of LUVs (Fig. S1C). In the cell, BAX activation is mediated primarily through BCL-2 family direct activators predominantly, BIM[26]—and we had observed apoptotic resistance in the Bim[–/–]Bid[–/–] MEFs; therefore, we sought to assess the cooperation of BIM and hexadecenal on BAX-mediated membrane permeabilization. We treated BAX with a range of activating concentrations of BIM-BH3 peptide and observed a dose-dependent increase and acceleration in LUV permeabilization in response to hexadecenal (Figs. 1D, F and S1D). Furthermore, we determined that the additional response to BIM-BH3 and hexadecenal was synergistic (Fig. 1G). Previous work indicated that the saturated form of 2t-hexadecenal hexadecanal ("16CHO")—did not induce BAX oligomers in cross-linking studies[17], and indeed, we did not observe BAX-mediated pore formation or synergy with BIM in response to hexadecanal (Figs. 1H–I and S1E). Similarly, we modified hexadecenal by reacting it with a hydrazide compound and determined that the resulting molecule ("2t16-NN-EtOH") did not promote BAX activation or synergize with BIM stimulation (Figs. 1J–K and S1F). These data indicate that hexadecenal promotes BAX pore formation and synergizes with BIM-mediated activation. Of note, however, we cannot exclude the possibility that the alteration of the aldehyde moiety in 2t16-NN-EtOH may have compromised the reactivity of the conserved double bond.

Despite treating BAX with hexadecenal directly, we could not entirely rule out the possibility that increased permeabilization could have been due to the lipidic aldehyde interacting with LUVs and resulting in a more permissive environment for BAX pore formation. Therefore, we utilized microscale thermophoresis (MST) to determine whether hexadecenal directly bound to BAX and observed a dose-dependent shift indicating changes to the molecular volume of BAX in

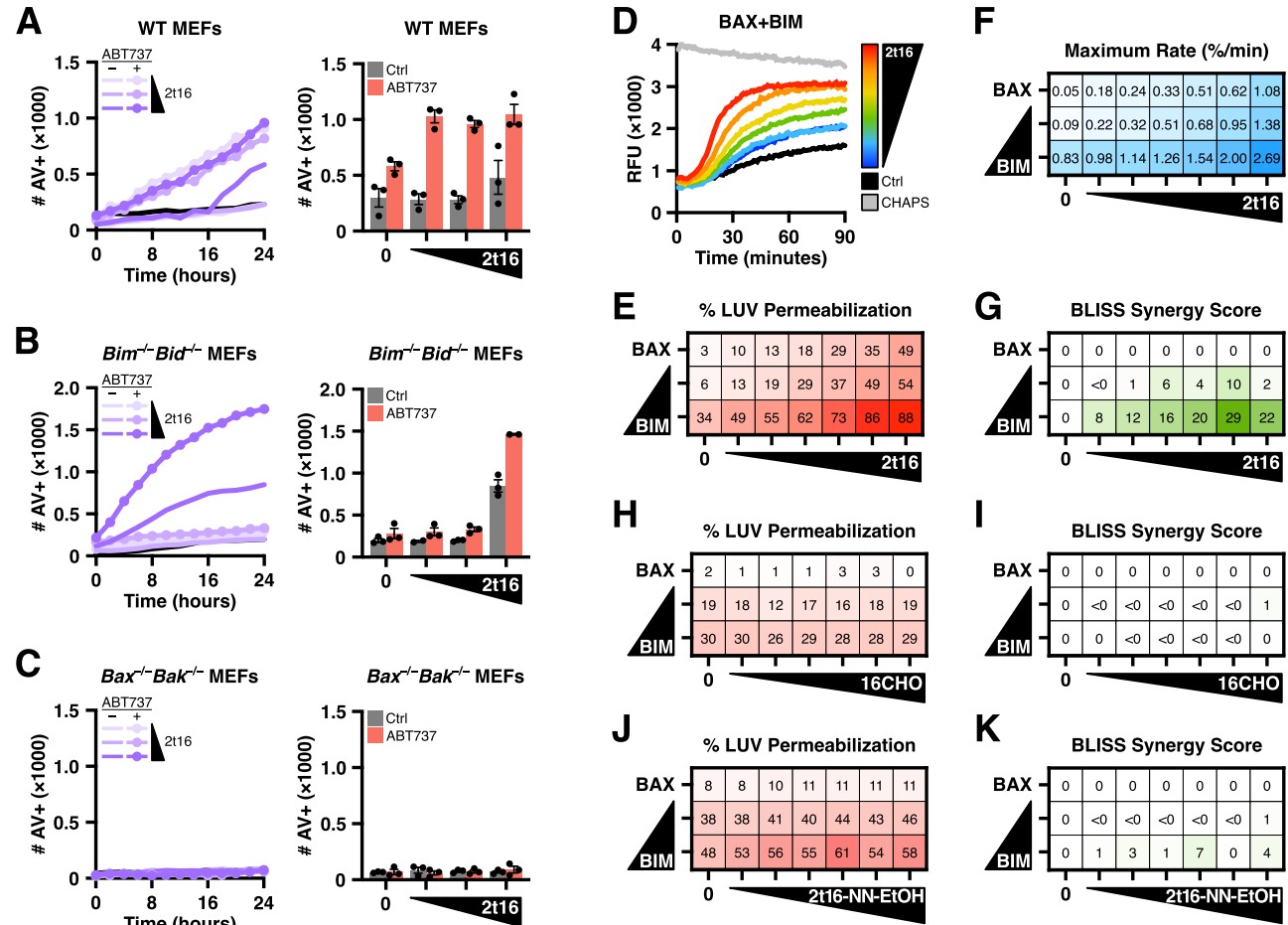

**Fig. 1 | 2t-hexadecenal induces apoptosis and membrane permeabilization by cooperating with BAX. A–C** MEFs subjected to SPARKL analysis measuring real-time labeling with fluorescently-tagged Annexin V (100 µg/ml) via imaging with an IncuCyte ZOOM. Left panels: kinetics of cell labeling in response to increasing concentrations of 2t-hexadecenal (2t16) with DMSO vehicle or co-treated with ABT-737; black line reports untreated control. Right panels: endpoint data of replicates at 24 or 26 h. Data shown are the mean of technical triplicates and error bars report SEM. **A** WT MEFs (matched to $Bax^{-/-}Bak^{-/-}$ double knockout MEFs) were treated with 2t16 (10, 20, 40 µM) and vehicle or ABT-737 (1 µM), imaged every 2 h, and quantified for number of Annexin V-positive objects. **B** Same as in (**A**) with $Bim^{-/-}Bid^{-/-}$ double knockout MEFs. **C** Same as in (**A**) with $Bax^{-/-}Bak^{-/-}$ double knockout MEFs. **D** BAX protein (120 nM) was activated with BIM-BH3 peptide in the presence of DMSO vehicle (Ctrl) or 2t16 (6.5–50 µM) followed by addition of LUVs. Changes in fluorescence were measured over time as an indication of liposome

permeabilization. Grey data report LUVs solubilized with 1% CHAPS to measure maximal signal. Data shown are the mean of technical replicates. **E** Heatmap visualization of normalized endpoint LUV permeabilization data from LUVs incubated with BAX (120 nM) treated with 2t16 (6.5–50 µM) ± BIM-BH3 peptide (0.5, 2.5 µM). Data summarized from Figure S1D. **F** Heatmap visualization of maximum permeabilization rate determined by fitting of data from (**E**) and (S1D). **G** Bliss synergy scores calculated using the endpoint LUV permeabilization data from (**E**) and (S1D). **H, I** Endpoint data and Bliss synergy scores for LUV permeabilization studies as in (**E**) with BAX (160 nM) and hexadecanal (6.5–50 µM) ± BIM-BH3 peptide (2.5 µM). Data summarized from Fig. S1E. **J, K** Endpoint data and Bliss synergy scores for LUV permeabilization studies as in (**E**) with BAX (140 nM) and the 2t16 hydrazino product (6.5–50 µM) ± BIM-BH3 peptide (2.5 µM). Data summarized from Fig. S1F. See also Fig. S1.

response to hexadecenal (Fig. 2A). Of note, the MST data determined an EC50 value of 0.182 µM, or when hexadecenal was ~180-fold the concentration of BAX, which is approximately the ratio that exhibited the greatest synergy with BIM-mediated BAX membrane permeabilization (Figs. 1G and S2A). As such, we utilized a range of hexadecenal concentrations that represented ratios aligning with the MST data in our subsequent assays. Interestingly, the saturated hexadecanal aldehyde did not induce a similar change in BAX thermophoresis, suggesting that loss of the α,β double bond disrupted interaction with BAX and subsequent activation (Figs. 2B and S2B).

Hexadecenal is an α,β-unsaturated aldehyde and is capable of modifying nucleophiles (e.g., cysteine residues) through Michael addition, and the loss-of-function of the saturated aldehyde and 2t16-NN-EtOH, could have suggested a chemical reaction mechanism[27]. Indeed, recent publications have reported that hexadecenal covalently modifies over 500 cellular proteins, including BAX, though the studies disagree on which cysteine residue was modified[21,22]. To determine if

BAX was being covalently modified within our assays, we incubated BAX with hexadecenal, subjected the sample to proteolytic cleavage, and analyzed the peptide fragments by tandem liquid chromatography-mass spectrometry (LC-MS). We did not observe alkylation by hexadecenal ($m + 238.229$ Da) on either of the BAX cysteine residues (C62, C126); as a control, we did detect cysteines modified by the alkylating agent iodoacetamide ($m + 57.021$ Da) (Fig. S2C). However, we could not eliminate the possibility that hexadecenal-modified BAX may have not been detected due to limitations of the digestion or peptide detection; therefore, we corroborated our results using native intact protein time-of-flight (TOF) mass spectrometry. We utilized an oligomerization-deficient BAX mutant (BAX$^{G108V}$) to prevent BAX aggregation and observed no mass shift in response to hexadecenal, though we did observe modification by iodoacetamide (Fig. 2C). Additionally, we replicated this experiment using an alternative oligomerization-deficient BAX mutant (BAX$^{R109D}$) to confirm that these results were not specific to either mutation (Fig. 2D).

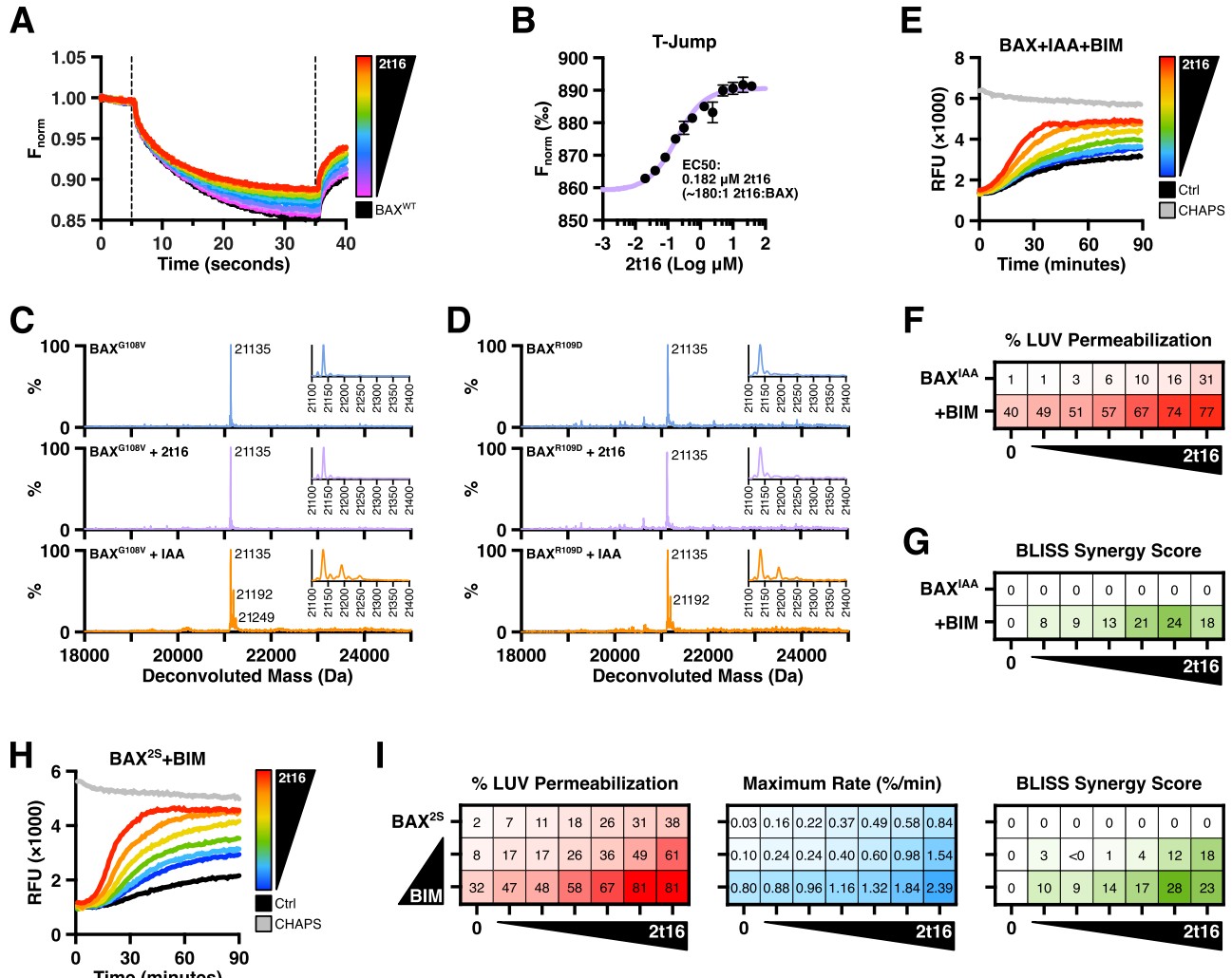

**Fig. 2 | 2t-hexadecenal activation of BAX is mediated by non-covalent interactions. A**, **B** Alexa Fluor 647-labeled recombinant BAX$^{WT}$ (1 nM) was incubated with CHAPS (0.002%) to inhibit oligomerization, treated with 2t16 (0.02–40 μM) and subjected to MST. **A** Timetrace thermal shift curves of a representative experiment. **B** Average temperature jump values from three independent experiments with error bars reporting SEM. **C**, **D** BAX protein with G108V or R109D mutations was treated with vehicle control, 2t16 (50 μM), or iodoacetamide (IAA, 50 mM) for 1 h and subjected to native intact TOF mass spectrometry. **E**–**I** LUV permeabilization studies with BAX protein treated as indicated and measured at regular intervals for changes in fluorescence as fluorophores are released from compromised liposomes. Grey data report LUVs solubilized with 1% CHAPS to measure maximal signal. Data shown are the mean of technical replicates. **E** BAX$^{WT}$ was pre-treated with iodoacetamide (IAA, 50 mM) for 1 h prior to activation with BIM-BH3 peptide (2.5 μM) in the presence of DMSO vehicle (Ctrl) or 2t16 (6.5–50 μM) followed by addition of LUVs. **F** Heatmap visualization of normalized endpoint LUV permeabilization data from (**E**). **G** Bliss synergy scores calculated using the endpoint LUV permeabilization data from (**F**). **H** BAX$^{2S}$ protein (100 nM) was activated with BIM-BH3 peptide in the presence of DMSO vehicle (Ctrl) or 2t16 (6.5–50 μM) followed by addition of LUVs and measured by fluorescence spectrometry. **I** Endpoint data (left panel), maximal rate derived from non-linear regression (middle panel), and Bliss synergy scores (right panel) from LUVs incubated with BAX$^{2S}$ (100 nM) treated with 2t16 (6.5–50 μM) ± BIM-BH3 peptide (0.5, 2.5 μM). Data summarized from Fig. S2G. See also Fig. S2.

To decisively conclude that the mechanism of hexadecenal synergy was independent of cysteine modification, we replicated our LUV permeabilization studies using cysteine-deficient BAX generated by two complementary approaches. First, we functionally inhibited the reactivity of the BAX cysteines via iodoacetamide capping and demonstrated that there was no loss of activation or synergy in response to BIM-BH3 and hexadecenal (Fig. 2E–G). Next, we utilized a cysteine-replacement BAX mutant (BAX$^{C62S,C126S}$, "BAX$^{2S}$"), which exhibited no changes to stability or melting temperature (Fig. S2D). Compared to wild-type BAX ("BAX$^{WT}$"), BAX$^{2S}$ remained sensitive to hexadecenal-mediated changes in melting temperature, pore formation, and synergy with BIM-BH3 peptide (Figs. 1D–G, 2H–I, and S2E–G). Additionally, BAX$^{2S}$ remained similarly insensitive to the saturated hexadecanal aldehyde (Fig. S2H–I). Collectively, these results indicate that there is a non-covalent mechanism by which

hexadecenal promotes BIM-mediated BAX activation through a direct interaction.

## 2t-hexadecenal cooperates with BIM to promote early-activation of monomeric BAX

The BAX activation continuum can be divided into distinct phases of activation and functionalization: the cytosolic monomer gets activated and undergoes intramolecular rearrangements and conformational changes that result in translocation to the OMM; subsequently, active BAX proteins in the OMM undergo large-scale conformational changes, oligomerize, and mature into pore-forming units. Our data thus far have measured BAX functionalization (i.e., pore formation) and demonstrated that hexadecenal synergizes with BIM to promote membrane permeabilization. However, we utilized a direct treatment model of hexadecenal and BAX protein (compared to membrane

incorporation), indicating that the mechanism of action likely occurs on BAX found in the cytosol and prior to interactions with the OMM. To investigate the effect of hexadecenal specifically on the activation phase of BAX, we utilized a technique we developed called FLAMBE, which monitors activation-induced intramolecular rearrangements within BAX (i.e., rearrangements that result in early-activation structural hallmarks such as displacement of the α1–α2 loop and mobilization of the C-terminal α9 helix)[4,8]. FLAMBE observes real-time early-activation of BAX by measuring changes in Polarization resulting from the kinetic binding and dissociation of a TAMRA-labeled BAK-BH3 peptide (BAK$^{TAMRA}$); broadly, BAX activation can be inferred by measuring a reduction of Polarization over time (Fig. S3A). Kinetic FLAMBE data can further be parameterized into time-of-maximum-Polarization (Tmax) and endpoint Polarization (EP) for trend overviews and comparative analyses between treatment conditions (Fig. S3B). As an example, BAX treated with a range of BIM concentrations exhibits a dose-dependent pattern of BAK$^{TAMRA}$ dissociation (measured as a reduction in Polarization over time) indicative of BAX activation (Fig. S3C).

Using FLAMBE, we observed BAX$^{WT}$ activation in the presence of hexadecenal as demonstrated by a reduction in Polarization over time, indicating that high concentrations of hexadecenal could induce a "direct-activator"-like effect on BAX monomers, which aligns with the LUV permeabilization data (Fig. 3A). To confirm that this phenotype was not due to covalent modification of cysteine residues, we replicated the FLAMBE experiment with BAX$^{2S}$ and observed the same activation profile (Fig. 3B). By contrast, neither BAX$^{WT}$ nor BAX$^{2S}$ activated in response to the saturated hexadecanal aldehyde, though we did observe some destabilization of the BAX:BAK$^{TAMRA}$ heterodimer at the higher concentrations (Fig. S3D–E). We hypothesized that hexadecenal cooperates with direct activators instead of activating BAX de novo within the cell, and thus we investigated whether the synergy between hexadecenal and BIM occurs during the early-activation phase. We treated BAX$^{2S}$ with a non-activating concentration of BIM-BH3 to generate a population of stable BIM:BAX:BAK$^{TAMRA}$ heterotrimers (Fig. 3C, yellow data). The addition of a non-activating concentration of hexadecenal resulted in activation of the primed BAX$^{2S}$ population (Fig. 3C, left panel); moreover, activation of primed BAX$^{2S}$ was observed with several non-activating concentrations of hexadecenal (Fig. 3C, right panel). When we induced a "triggered" BAX population (i.e., mildly activated by BIM), we similarly observed increased activation by non-activating concentrations of hexadecenal, as measured by increased kinetics of BAK$^{TAMRA}$ dissociation, greater shifts in parameterized metrics, and reduction in area under the curve (Figs. 3D and S3F). Importantly, we observed the same outcomes with BAX$^{WT}$, confirming that the mechanism was not due to the endogenous cysteines or their mutation (Fig. S3G). Critically, we observed no disruption of a BCL-xL/BAK$^{TAMRA}$ complex at comparative hexadecenal concentrations indicating that dissociation of BAK$^{TAMRA}$ from BAX was due to BAX activation and not competition for the BC groove or a non-specific detergent-like effect (Fig. S3H).

The consequence of BAX activation is translocation to the OMM and oligomerization, and while it is largely agreed that high-molecular weight oligomers are formed within the OMM, there is evidence supporting that physiologically-activated BAX forms low-order multimers (e.g., dimers) in solution prior to integrating with membranes[6,8,28]. We investigated the consequence of hexadecenal-mediated synergy with BIM-activated BAX by performing size exclusion chromatography (SEC) and observed a substantial shift from monomeric to dimeric BAX when co-treated with BIM-BH3 peptide and hexadecenal (Fig. 3E). To confirm that the earlier peak was BAX, we treated fluorescently-labeled BAX, subjected it to SEC, and screened the fractions for fluorescence. The monomeric BAX peak (fractions 28–31) shifted slightly left upon addition of hexadecenal or BIM-BH3 peptide (fractions 27–30), likely due to binding-dependent changes in molecular volume, and dimeric

species were observed in the BIM-treated sample (fractions 23–26); the co-treatment of hexadecenal and BIM-BH3 resulted in an increased shift and intensity indicating a greater percent of the BAX population formed multimeric species (fractions 22–26) (Fig. 3F). By contrast, no such shift in the monomeric peak or BIM-induced dimer peak was observed with hexadecanal (Fig. 3G). Of note, BAX activated in solution does not readily form high-molecular weight species without the stabilizing and concentrating influence of a hydrophobic environment (e.g., a membrane or micellar detergent)[8,29]; as a reference, we were able to observe BAX oligomers generated with a detergent (BAX$_O$)[30], and we interpret these results to indicate that hexadecenal is not exhibiting a detergent-like effect on BAX. These data collectively demonstrate that hexadecenal promotes monomeric BAX activation downstream of BCL-2 protein interactions, following activation by direct activators, and before interactions with a membrane.

## An alpha 8 helix-gated funnel-shaped hydrophobic cavity in the BAX core interacts with 2t-hexadecenal

Hexadecenal synergized with BIM-mediated BAX activation and we observed no evidence of competition with the BAK$^{TAMRA}$ peptide in our FLAMBE assays, suggesting that hexadecenal was binding to a site distinct from either the trigger site or BC groove—the two BH3-interacting sites, respectively[31,32]. Despite having a relatively smooth surface and no obvious "binding pocket," several studies have identified small molecules that bind to BAX, either at BH3-interacting sites or allosterically[33–35]. To identify putative interaction sites, we performed 2D $^1$H-$^{15}$N heteronuclear single quantum coherence (HSQC) nuclear magnetic resonance (NMR) of $^{15}$N-labeled BAX$^{WT}$ treated with hexadecenal and measured a multitude of peak shifts (Fig. S4A–C). Several residues exhibited significant chemical shift perturbations (CSPs) in response to hexadecenal, some of which were within unstructured regions (such as the N-terminus) or highly exposed areas (such as α4 and α9), but notably the two cysteine residues (C62 and C126) did not reach significance (Fig. 4A). Interestingly, several shifts were observed in core-facing residues of α2, α5, α6, and α8, as well as bulky residues proximal to α8 within the α4–5 loop and α7. Additionally, several residues in this region exhibited altered resonance morphology, such as peak splitting or merging (Fig. 4A, right). Critically, we did not observe any global loss of signal or peak intensities suggestive of protein loss due to either activation-induced oligomerization or degradation. To determine whether the shifted residues were specific to the activation mechanism, we compared this CSP profile against CSPs of BAX$^{WT}$ treated with the non-activating saturated aldehyde hexadecanal and observed that residues exhibiting significant shifts were more localized to accessible regions of BAX along α2, α3, and α9 (Fig. S4D). Comparing the CSPs of the two aldehydes highlighted that many of the hexadecenal-induced shifts were not observed with the saturated aldehyde (Fig. S4E). These core-facing residues are not believed to be readily accessible and therefore were likely to be meaningful for the mechanism of hexadecenal interaction and function.

Inspection of the BAX structure revealed a cavity formed by hydrophobic residues in the core of the protein. The cavity shape resembles a funnel and is comprised of two topographies: a wide and shallow "mouth" formed by residues in α1, α2, α5, and α8; a narrow "neck" extending into the BAX core between α1, α5, and α6 (Fig. 4B). Of note, the cavity is conserved in each of the structures of the NMR ensemble, albeit with differing shapes and connectivity, suggesting that it is a highly dynamic feature of monomeric BAX (Fig. S4F). We hypothesized that this hydrophobic "funnel" could be a desirable interaction site for hexadecenal—an inherently lipidic aldehyde—and therefore we performed unconstrained in silico docking calculations using the SwissDock web service to model interactions with BAX. Binding modalities of hexadecenal were clustered into a few distinct regions on BAX, but 59.4% of the poses were proximal to α8, aligning with the CSPs observed by NMR (Fig. 4C).

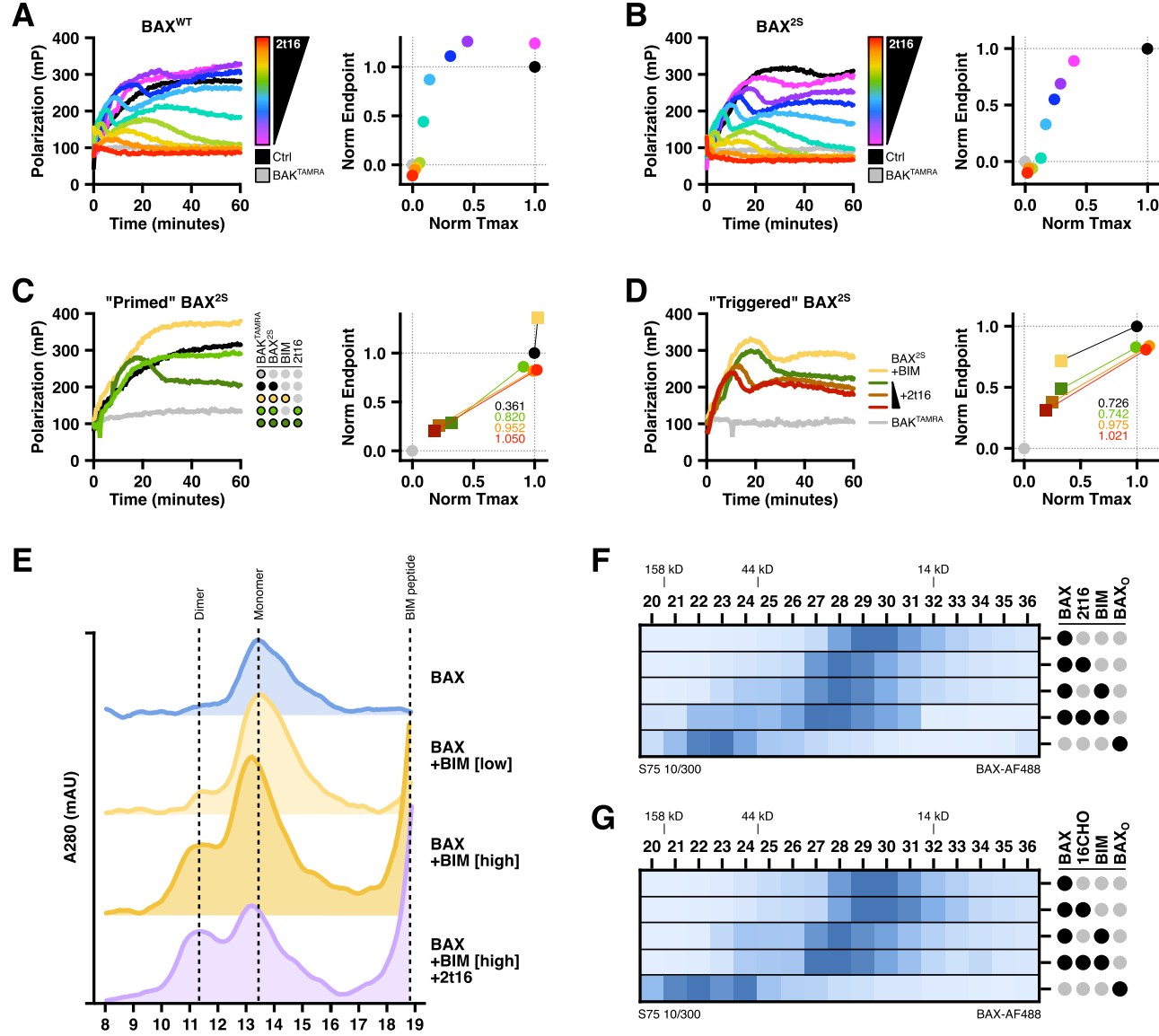

**Fig. 3 | 2t-hexadecenal synergizes with BIM to induce intramolecular rearrangements and early-activation of monomeric BAX. A–D** BAX was treated as indicated, combined with a TAMRA-labeled BAK-BH3 peptide (BAK$^{TAMRA}$), and immediately subjected to FLAMBE analysis in which the association and activation-induced dissociation of BAK$^{TAMRA}$ is monitored via changes in Polarization. Left panels: kinetic Polarization data; right panels: two-dimensional plot of parameterized FLAMBE data comparing Tmax and endpoint Polarization metrics normalized to BAK$^{TAMRA}$ (grey data) and the vehicle-treated BAX control (black data). Data shown are the mean of replicates. **A** BAX$^{WT}$ (60 nM) was treated with 2t16 (2–50 μM), combined with BAK$^{TAMRA}$ (50 nM), and subjected to FLAMBE. **B** Same as in (**A**) with BAX$^{2S}$ (60 nM). **C** Left: BAX$^{2S}$ (60 nM) was combined with non-activating concentrations of BIM-BH3 peptide (0.15 μM) and 2t16 (4.5 μM), followed by BAK$^{TAMRA}$ (50 nM), and subjected to FLAMBE. Right: Parameterized FLAMBE data including three concentrations of 2t16 (green: 4.5 μM; orange: 6.5 μM; red: 10 μM) in the absence or presence of BIM-BH3 (circle and square datapoints, respectively). Annotations report the magnitude of shift between data with and without BIM-BH3. **D** Left: Same as in (**C**) with an activating concentration of BIM-BH3 and three concentrations of 2t16 (green: 4.5 μM; orange: 6.5 μM; red: 10 μM). Right: Parameterized FLAMBE data reporting shift in samples with or without BIM-BH3. Conditions without BIM-BH3 provided in Fig. S3F. **E** BAX$^{WT}$ (800 nM) was treated with BIM-BH3 (2.5, 10 μM) and 2t16 (50 μM) for 1 h, subjected to size exclusion chromatography, and measured by 280 nM absorbance (A280) as eluate flowed out of the column. **F** Alexa Fluor 488-labeled BAX$^{WT}$ (400 nM) was treated with 2t16 (50 μM) and/or BIM-BH3 (10 μM) for 1 h and subjected to size exclusion chromatography. Samples of fractions were analyzed by fluorescence spectroscopy to monitor distribution of BAX. Oligomeric BAX (BAX$_O$) was generated via overnight treatment with DDPC (1 mM). Color scale denotes relative percent fluorescence in each fraction. **G** Same as in (**F**) with 16CHO (50 μM). See also Fig. S3.

Our functional interrogations indicated that hexadecenal synergizes with BIM-mediated BAX activation, and we reasoned that BIM-induced intramolecular arrangements may induce flexibility and/or mobility of α8, making the funnel accessible to hexadecenal. Swiss-Dock utilizes rigid receptor docking, which results in the α8 helix firmly blocking the funnel. To create a funnel-accessible structure, we removed α8 (BAX$^{Δα8}$) and docking against the BAX$^{Δα8}$ structure revealed a clear preference for the funnel with 97.3% of hexadecenal poses positioned within the funnel (Fig. 4D). Interestingly, when we examined the solution NMR structure of BAX bound to a stapled BIM-BH3[31]—a structure representing a quasi-active, "triggered" BAX—we observed an enlargement of the hydrophobic funnel, most notably in the neck of the funnel, and this was conserved across the structures within the ensemble (Figs. 4E and S4G). In the inactive BAX structure, the neck of the funnel is cinched into discontinuous cavities by residues of α5 and α6; in contrast, the BIM-bound active monomer

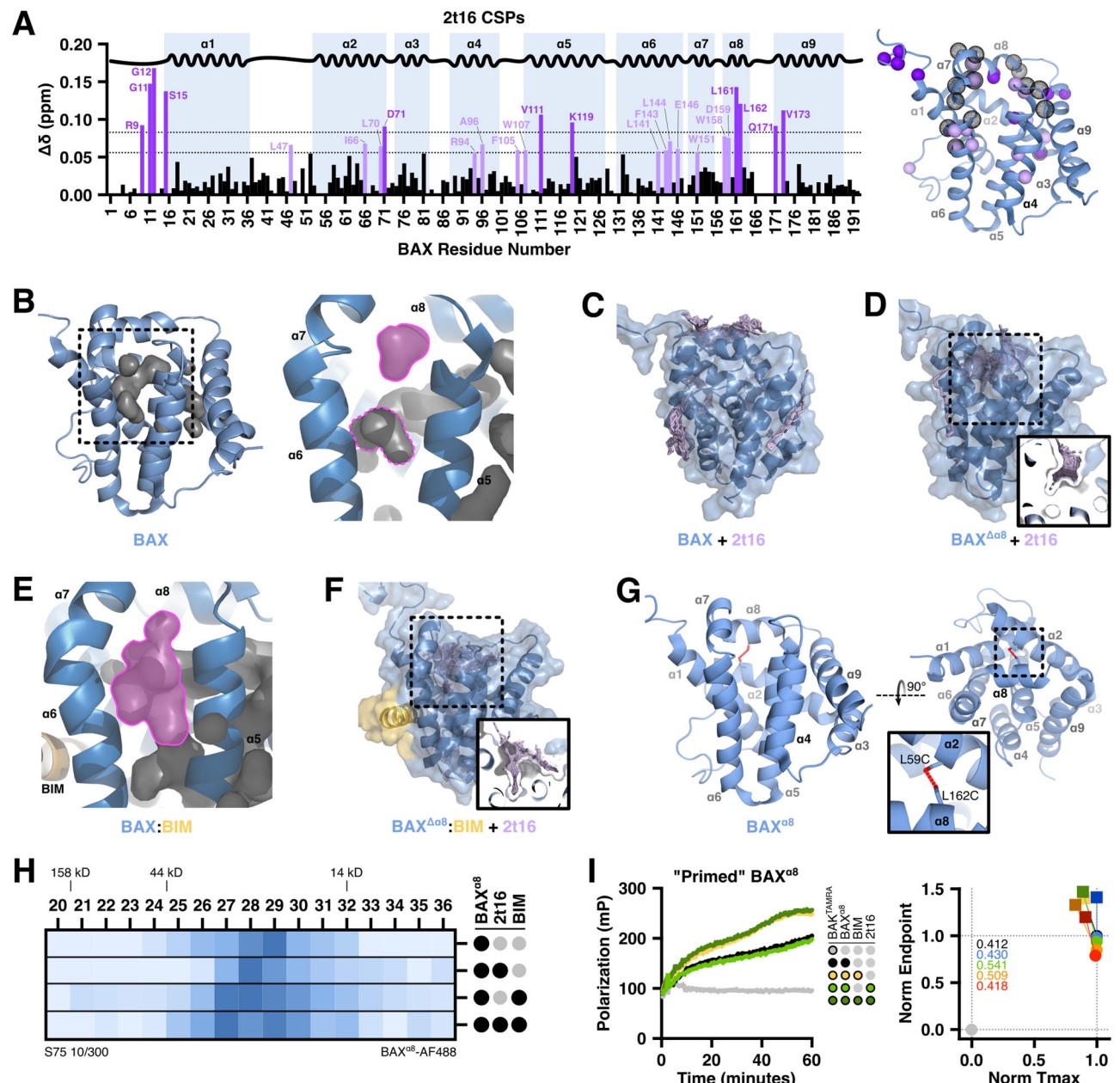

**Fig. 4 | 2t-hexadecenal interacts with an alpha 8 helix-gated hydrophobic cavity in the core of BAX. A** [15]N-labeled BAX[WT] (40 µM) was treated with vehicle or 2t16 (50, 150 µM) and subjected to [1]H-[15]N HSQC NMR. Left: Chemical shift perturbations (CSPs) are plotted as a function of BAX residues with shifts greater than 1 or 2 standard deviations above the average noted. Right: Residues exhibiting significant CSPs (light and dark purple spheres) or alterations to peak morphology (hollow spheres) are modeled on BAX (PDB: 1F16). **B** BAX structure (PDB: 1F16) and the hydrophobic cavity formed by core helices. The cavity proximal to α8 is highlighted; a distinct cavity in the core is outlined. Cavity determination and visualization was performed with PyMOL using a cavity radius of 3 and a cavity cutoff of −5.5. **C**, **D** Unconstrained in silico rigid docking of 2t16 against the unmodified (PDB: 1F16) or α8-removed (Δ157–163; "BAX[Δα8]") BAX using the Swiss-Dock web service. Inset displays cross-section of 2t16 docking within the hydrophobic funnel. **E** Unmodified structure of BAX bound to a BIM-BH3 peptide (PDB:

2K7W; "BAX:BIM") with the enlarged, connected cavity visualized as in (**B**). **F** Results of 2t16 docking on the structure of BAX:BIM with α8 removed (PDB: 2K7W, Δ157–163; "BAX[Δα8]:BIM") as in (**C**). **G** Visualization of the BAX[α8] mutant with residues L59 and L162 mutated to cysteines and oxidized to form a disulfide tether immobilizing α8. **H** Fluorescently-labeled BAX[α8] (400 nM) was treated with 2t16 (50 µM) and/or BIM-BH3 (10 µM) for 1 h and subjected to SEC. Fraction samples were analyzed by fluorescence spectroscopy to identify BAX distribution. Color scale denotes relative fluorescence in each fraction. **I** Left: BAX[α8] (225 nM) was combined with non-activating concentrations of BIM-BH3 peptide (0.15 µM) and 2t16 (4.5 µM), followed by BAK[TAMRA] (50 nM), and subjected to FLAMBE analysis. Right: Parameterized FLAMBE data including four concentrations of 2t16 (blue: 3 µM; green: 4.5 µM; orange: 6.5 µM; red: 10 µM) ± BIM-BH3 (circle and square datapoints, respectively). Annotations report the magnitude of shift ± BIM-BH3. Kinetic data provided in Fig. S4J. See also Fig. S4.

structure has a connected cavity due to a shift in the residues lining the funnel, increasing both the depth and width. Docking hexadecenal against the BAX:BIM structure with α8 removed (BAX[Δα8]:BIM) revealed that the aldehyde was positioned within the funnel, frequently posed as inserted into the neck of the funnel (Fig. 4F).

To substantiate our in silico conclusions that hexadecenal interacts within this cavity, we engineered a structural mutant of BAX to restrict the mobility of α8 and the subsequent access to the hydrophobic funnel. We introduced two cysteine residues into BAX[2S] (L59C, C62S, C126S, L162C; "BAX[α8]") that could be oxidized to induce a

disulfide tether between α2 and α8 (Fig. 4G). We characterized the consequence of the new mutations by activating BAX$^{\alpha 8}$ with BIM-BH3 peptide and assessing LUV permeabilization. Reduced BAX$^{\alpha 8}$ (i.e., "unlocked") remained functional and exhibited BIM-induced LUV permeabilization (Fig. S4H); in contrast, oxidized BAX$^{\alpha 8}$ (i.e., "locked") demonstrated no permeabilization (Fig. S4I). Several studies have characterized that BAX functionalization and pore formation require large-scale conformational reorganization, which involves the separation of α8 from core helices, and therefore it is not surprising that locked BAX$^{\alpha 8}$ was a "functionally dead" structural mutant[5–7,36]. However, we recently demonstrated that several historically "dead" BAX mutants remain sensitive to BH3-induced activation and demonstrate early-activation structural hallmarks despite not maturing into functional pore-forming conformations[8]. SEC experiments revealed that locked BAX$^{\alpha 8}$ exhibited a shift in response to BIM-BH3 but was unaffected by either hexadecenal alone or coupled with BIM-mediated activation (Fig. 4H). To further investigate whether the immobilized α8 helix would disrupt sensitivity to hexadecenal, we studied BAX$^{\alpha 8}$ activation using our FLAMBE assay. Both the unlocked and locked forms of BAX$^{\alpha 8}$ demonstrated activation by BIM-BH3, indicating that that the mutant still exhibits activation-induced intramolecular rearrangements (Fig. S4J–K). Critically, locked BAX$^{\alpha 8}$ did not activate when treated with hexadecenal (Fig. S4L). Moreover, BIM-primed BAX$^{\alpha 8}$ displayed no activation or synergy in the presence of hexadecenal (Figs. 4I and S4M). These data substantiate the results of our docking investigations and the conclusion that hexadecenal-induced BAX activation is mediated through non-covalent interactions with the hydrophobic funnel in the BAX core, which is exposed following BIM triggering and subsequent α8 mobilization. Given that this hydrophobic cavity potentiates BAX activation, we termed this site the "BAX Actuating Funnel", or BAF.

## Aldehyde length and BAF topography are determinants of BAX activation

Neither the saturated aldehyde hexadecanal nor the hydrazide-modified hexadecenal activated BAX suggesting that the mechanism is distinct to the structure of 2t-hexadecenal. We next considered the relevance of aldehyde length on interactions with the BAF by utilizing a panel of 2t-alkenals ranging from 5 to 13 carbon chains (henceforth, "alkenals"). HSQC experiments with a mid- or short-chain alkenal, 2t-nonenal ("2t9") and 2t-pentenal ("2t5"), respectively, demonstrated CSP profiles that were similar to hexadecenal, but were also more diffuse across BAX residues (Fig. 5A). While there was some conservation of CSPs observed in residues within or proximal to α8, the shorter alkenals exhibited a plethora of shifts within α4 and α9 as well as a general trend of interacting with solvent-exposed residues (Fig. S5A). These data may also indicate that shorter alkenals are more promiscuous, as nonenal and especially pentenal interacted with several accessible regions of BAX, exhibited decreasing specificity at higher concentrations, and displayed weaker CSPs (Fig. S5B–C). Docking investigations with alkenals ranging from 5 to 15 carbon chains supported this hypothesis by demonstrating a length-dependent specificity for the α8 region (in the full-length structure, BAX$^{FL}$) or the BAF (in the BAX$^{\Delta \alpha 8}$ structure) (Figs. 5B and S6A, B). When docked against the BAX$^{\Delta \alpha 8}$:BIM structure, most alkenal species were predicted to interact with the larger BAF, though many were not positioned in the depth of the funnel and short-chain aldehydes could not occupy both the mouth and neck of the BAF in the same pose (Fig. S6C). In the BIM-bound BAX structure, mobilization of the α1–α2 loop created a region that was predicted to be an interaction site, and longer aldehyde structures were disproportionately positioned in this region; we believe this was an artifact of rigid receptor docking and not a physiological phenomenon, and therefore we did not quantify the percentage of binding poses localized to the BAF. The precursor of 2t-hexadecenal, sphingosine 1-phosphate (S1P), was previously

identified as a requirement for BAK-mediated MOMP but did not directly promote BAX-mediated MOMP[17]. We docked the structure of S1P against BAX models and observed a reduced preference for the BAF and the tail could not occupy the neck of the BAF when modeled against BAX$^{\Delta \alpha 8}$:BIM structure, likely due to the bulky and negatively charged phosphate group (Figs. 5B and S6A–C). Collectively, NMR and unconstrained, global docking assessments supported our hypothesis that aldehyde length and structure promote specificity for the BAF.

Next, we tested our hypothesis by investigating which, if any, alkenals could phenocopy hexadecenal function and BAX activation. When subjected to MST, BAX$^{WT}$ exhibited altered thermophoresis with each of the alkenals; though the fit of the data demonstrated a trend in which the slope increased with aldehyde length (Figs. 5C and S7A). To interrogate the functional consequence of alkenals interacting with BAX, we next subjected each alkenal to FLAMBE analysis and observed a dependency on aldehyde length for BAX$^{2S}$ activation (Fig. 5D). Interestingly, at the concentrations we tested, 2t-undecenal ("2t11") and 2t-dodecenal ("2t12") displayed some disruption of BAX$^{2S}$:BAK$^{TAMRA}$ interactions, similar to the results with hexadecanal, but BAK$^{TAMRA}$ was able to re-bind and did not exhibit an activation profile (Figs. S3E and S7B). When compared to hexadecenal, only 2t-tridecenal ("2t13") exhibited a clear dose-dependent activation trend, albeit less potently than hexadecenal (Figs. 5D–E and S7B). To definitively determine alkenal-induced BAX activation, we performed LUV permeabilization studies and confirmed a length-dependent sensitivity for BAX$^{2S}$-mediated pore formation (Figs. 5F and S7C). Additionally, these results were conserved in experiments using BAX$^{WT}$. These observations align with a prior study demonstrating that short-chain alkenals exhibited greater reactivity with BAX but were paradoxically less proficient at inducing BAX activation compared to long-chain alkenals[22]. It has been reported that 2t-alkenals exhibit reduced reactivity in a length-dependent manner, despite having similar electrophilic indices, possibly due to differences in steric hindrance or relative solubility/hydrophobicity, though this was not specifically measured for 2t-hexadecenal or long-chain alkenals[37,38]. Taken together with our structural and modeling data, these results indicate that chain length is a determinant of α,β-unsaturated alkenals to specifically target the BAF and activate BAX.

Having identified determinants of ligand specificity, we next sought to characterize the residues of the BAF that regulate interactions with, and sensitivity to, hexadecenal. We performed molecular docking of hexadecenal against the BAX:BIM structure using the Glide software with a docking grid centered on the BAF. As we expected, the residues lining the BAF that interact with hexadecenal were predominantly hydrophobic residues of α5 and α6 and all predicted poses were positioned into the neck of the BAF in a "tail-first" orientation (Fig. 6A, B). Compared against the docked position of the inactive 2t16-NN-EtOH molecule, the modified headgroup induced a pose that did not reside in the BAF (Fig. S8A). When we analyzed the top ten scored poses for 2t16-NN-EtOH, we observed several possible poses including the neck of the BAF, extensions towards α9, or entirely nested in the mouth of the BAF, indicating a departure from the residues predicted to stabilize the hexadecenal pose (Fig. S8B). Therefore, these in silico determinations suggest that 2t16-NN-EtOH fails to synergistically activate BAX due to altered binding conferred by the modified structure.

Interestingly, when we introduced additional positional constraints on the oxygen atom of hexadecenal, we were able to generate a "head-first" orientation, which did exhibit slightly better binding scores according to Glide; despite this, we could not replicate this orientation without dictating the position of the oxygen within the BAF. As such, we cannot decisively conclude which orientation may be preferred or representative of physiological interactions, but the conclusion that hexadecenal resides within the depth of the BAF is conserved in both models.

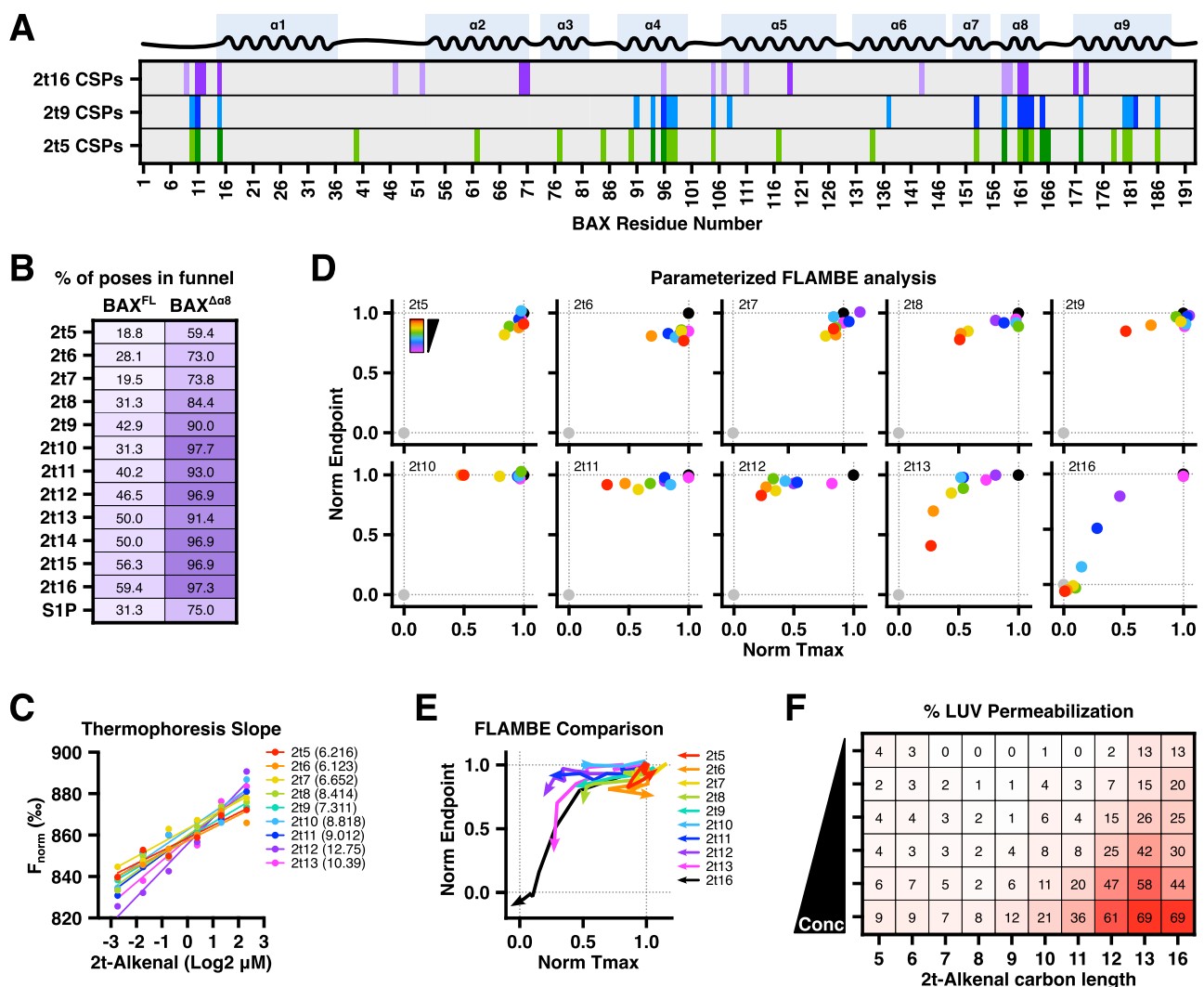

**Fig. 5 | Aldehyde length controls BAF function and BAX activation. A** [15]N-labeled BAX[WT] (40 µM) treated with 2t16, nonenal (2t9), or pentenal (2t5) and subjected to [1]H-[15]N HSQC NMR. Residues exhibiting significant CSPs averaged across concentrations for each of the indicated 2t-alkenal are highlighted. Each row is color-coded to report residues exhibiting shifts greater than 1 (light color) or 2 (dark color) standard deviations above the average across measurable shifts. Data summarized from Fig. S5A. **B** Results of unconstrained in silico rigid docking 2t-alkenals or S1P against the full-length BAX or BAX[Δα8] structure using the SwissDock web service. Binding modalities were inspected for positioning proximal (for BAX[FL]) or within (BAX[Δα8]) the BAF and reported as the percent of total poses. Visualizations of docking results provided in Fig. S6A, B. **C** Summary of MST analyses with BAX[WT] incubated with 2t-alkenals of differing carbon lengths (5–13 carbons) at a range of concentrations (0.16–5 µM). The thermophoresis datapoints were fit and the slope is provided for each alkenal. Original timetrace data provided in Fig. S7A. **D** Parameterized data from FLAMBE studies with BAX[2S] (60 nM) treated with 2t-alkenals of differing carbon lengths (5–13, 16 carbons) at several concentrations (3–50 µM). Black and grey datapoints report vehicle-treated BAX[2S] and BAK[TAMRA], respectively. Original data provided in Fig. S7B. **E** FLAMBE data trends for each of the 2t-alkenal titration from (**D**). **F** Heatmap visualization of endpoint permeabilization data from LUVs incubated with BAX[2S] (100 nM) treated with 2t-alkenals of differing carbon lengths (5–13, 16 carbons) at several concentrations (6.5–50 µM). Each experiment was normalized to the matching vehicle-treated BAX[2S] condition to control for variability. Original data provided in Fig. S7C. See also Figs. S5–7.

We next performed a virtual mutagenesis screen aimed at disrupting the BAF and identified that V110L, L113M, or L144F mutations were likely to alter the BAF (Fig. 6C). Indeed, performing molecular docking on these mutant structures demonstrated worse binding scores and altered hexadecenal positioning due to the disruption of BAF topography (Fig. 6D, E). Of note, we ran clustering analysis on all the docked poses for each mutant and found that the poses were sufficiently similar to be clustered together with the exception of BAX[L113M], which still positioned one pose into the shallower BAF; however, this pose had worse scoring compared to the top L113M hits.

To corroborate our in silico predictions, we generated recombinant protein for these BAF mutants and tested them for BAX functionality. Several established BAX point mutants result in a deficient or functionally dead protein, and so we first determined the melting

temperature of our BAF mutants. Compared to WT, each BAF mutant exhibited a lower melting temperature, suggesting that these mutations would not be functionally deficient (Fig. 6F). While the lower melting temperature may also suggest reduced stability of the monomeric protein, we did not observe any increased redistribution to oligomeric species during protein purification and storage. Furthermore, we confirmed that each BAF mutant responded to BIM activation with similar kinetics and endpoint, noting only slight insensitivity of BAX[L113M] at low BIM concentrations but similar activation at high concentrations (Fig. S8C, D). Critically, each BAF mutant demonstrated complete insensitivity to hexadecenal and exhibited a loss of synergy with BIM-primed BAX (Fig. 6G, H and S8E, F). It is worth noting that BAX[L113M] did retain some synergy of BIM and hexadecenal, albeit substantially reduced, and we suggest that this may be a

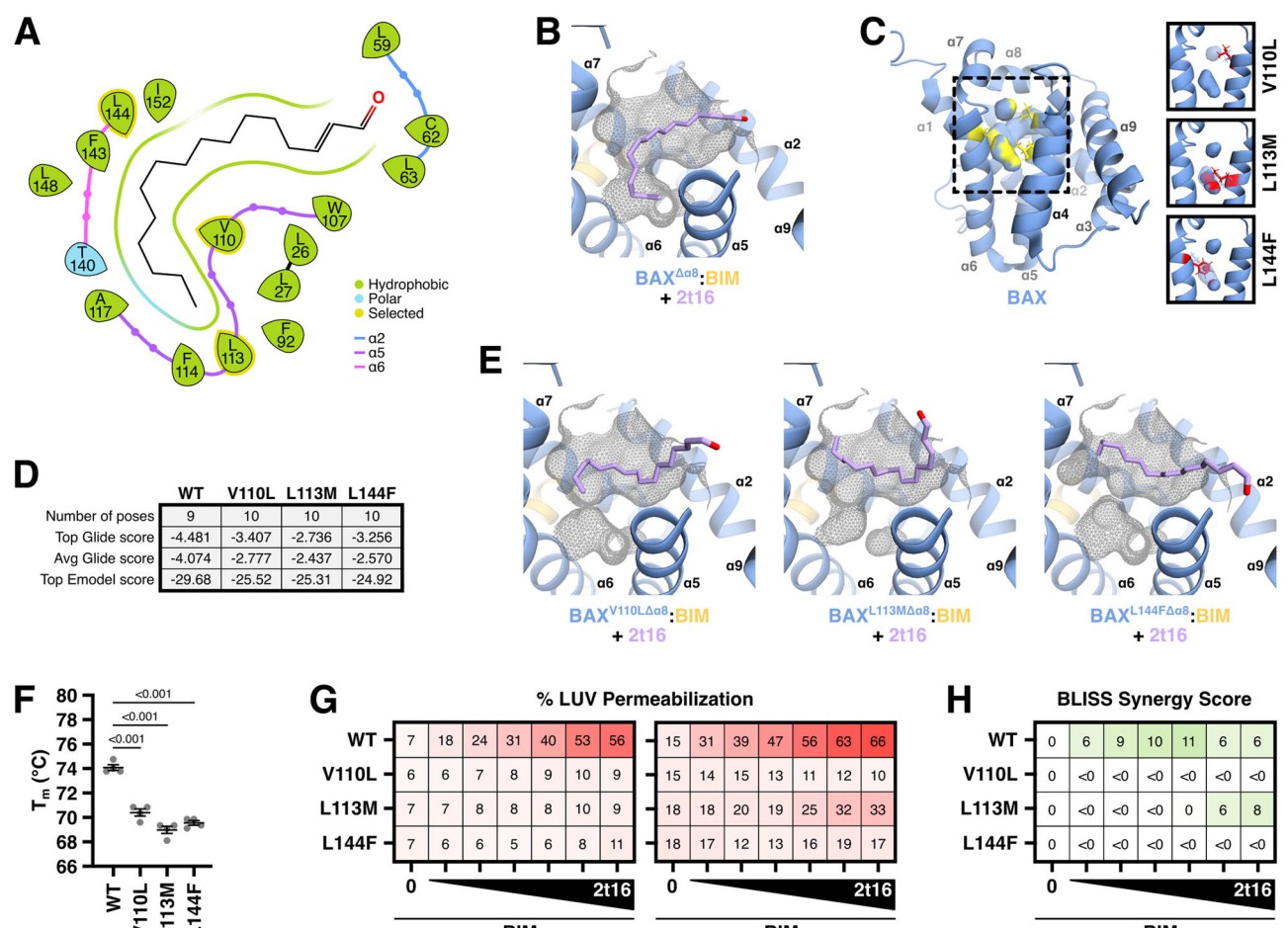

**Fig. 6 | Proximal BAF mutations disrupt 2t-hexadecenal synergy with BAX and BIM. A** Molecular docking and modeling of BAX$^{\Delta\alpha8}$:BIM (PDB: 2K7W, Δ157–163) with 2t16 using Schrödinger Glide with a binding region localized to the BAF. The interaction diagram identified several hydrophobic residues lining the BAF. **B** The highest scoring pose from (**A**) is shown and is representative of all generated poses. Surface of exposed BAF shown in grey wireframe and clipped to aid in visualization. **C** Cartoon visualization of full-length BAX (PDB: 1F16) with BAF residues identified from (**A**) highlighted in yellow. A virtual mutagenesis screen was performed with PyMOL to identify BAF-disrupting mutations. Right: candidate side chain replacements altering the BAF are shown in red and compared to the WT BAF. Cavity determination and visualization was performed with PyMOL using a cavity radius of 3 and a cavity cutoff of -5.5. **D** Molecular docking as in (**A**) against structures containing the indicated mutations. Generated poses of 2t16 had reduced

interaction scores against mutant structures compared to the wild type structure resulting from the altered BAF. **E** The highest scoring poses from (**D**) are shown and representative of generated poses. **F** The melting temperature of each mutant was measured by thermal shift differential scanning fluorimetry using SYPRO orange and compared using four replicates. Individual values, the mean, and SEM are shown. Statistical significance was determined by two-way ANOVA with multiple comparisons corrected using Dunnett T3 test and $P$ values are reported. Data shown are representative of replicated experiments. **G** Heatmap visualization of normalized endpoint LUV permeabilization data from LUVs incubated with the BAF mutants treated with 2t16 (6.5–50 μM) ± BIM-BH3 peptide (0.15 μM). Data summarized from Fig. S8E, F. **H** Bliss synergy scores calculated using the endpoint LUV permeabilization data from (**G**) and (**S8F**). See also Fig. S8.

consequence of a shallower, but relatively intact, BAF in the BAX$^{L113M}$ structure (Fig. 6E). As an aside, we did generate and screen candidate mutations altering the bulky BAF residues (i.e., F114 and F143), but observed reduced stability, oligomerization during purification, and substantially increased sensitivity to BIM, which obfuscated interpretations with hexadecenal. Interestingly, a prior study also noted that mutation of F114 is highly sensitizing[39], and we posit that substituting these bulky residues enlarges the BAF and subsequently destabilizes BAX (see Discussion). Collectively, these investigations indicate that the activating mechanism of hexadecenal is the ability of the aldehyde to reside in the depth of the BAF, and alteration to the BAF shape or aldehyde size disrupts this fundamental interaction.

### Proline 168 allosterically controls the BAF and 2t-hexadecenal function

Our modern understanding of the BAX structure-function relationship includes the concept of allostery, in which interactions with BAX can

cause structural rearrangements in distal regions. This concept is observed during BIM-mediated activation, which elicits mobilization of α9 from the BC groove[31], or binding of a sensitizing molecule to the hairpin pocket, which provokes exposure of the α1−2 loop[34]—both structural events are hallmarks of BAX activation[4]. Recently, the structure of BAX containing a mutation of proline 168 (BAX$^{P168G}$) was proposed to cause rotation of bulky sidechains in the distal α4−5 and α7−8 loops[40]. Despite identification of P168 as critical for BAX activation, translocation, and pore-forming ability over 20 years ago[41], and its recent identification as an arising loss-of-function mutation conferring resistance to Venetoclax in acute myeloid leukemia[42], a mechanistic explanation for its requirement has remained elusive. We hypothesized that the alternative rotamers in bulky α8-proximal residues may consequently alter the BAF and, indeed, the neck of the BAF is lost in the BAX$^{P168G}$ structure (Fig. 7A). Furthermore, molecular docking against the BAX$^{P168G}$ structure positioned hexadecenal with the carbon chain now pivoting towards α1 residues (Fig. 7A, B). Directly

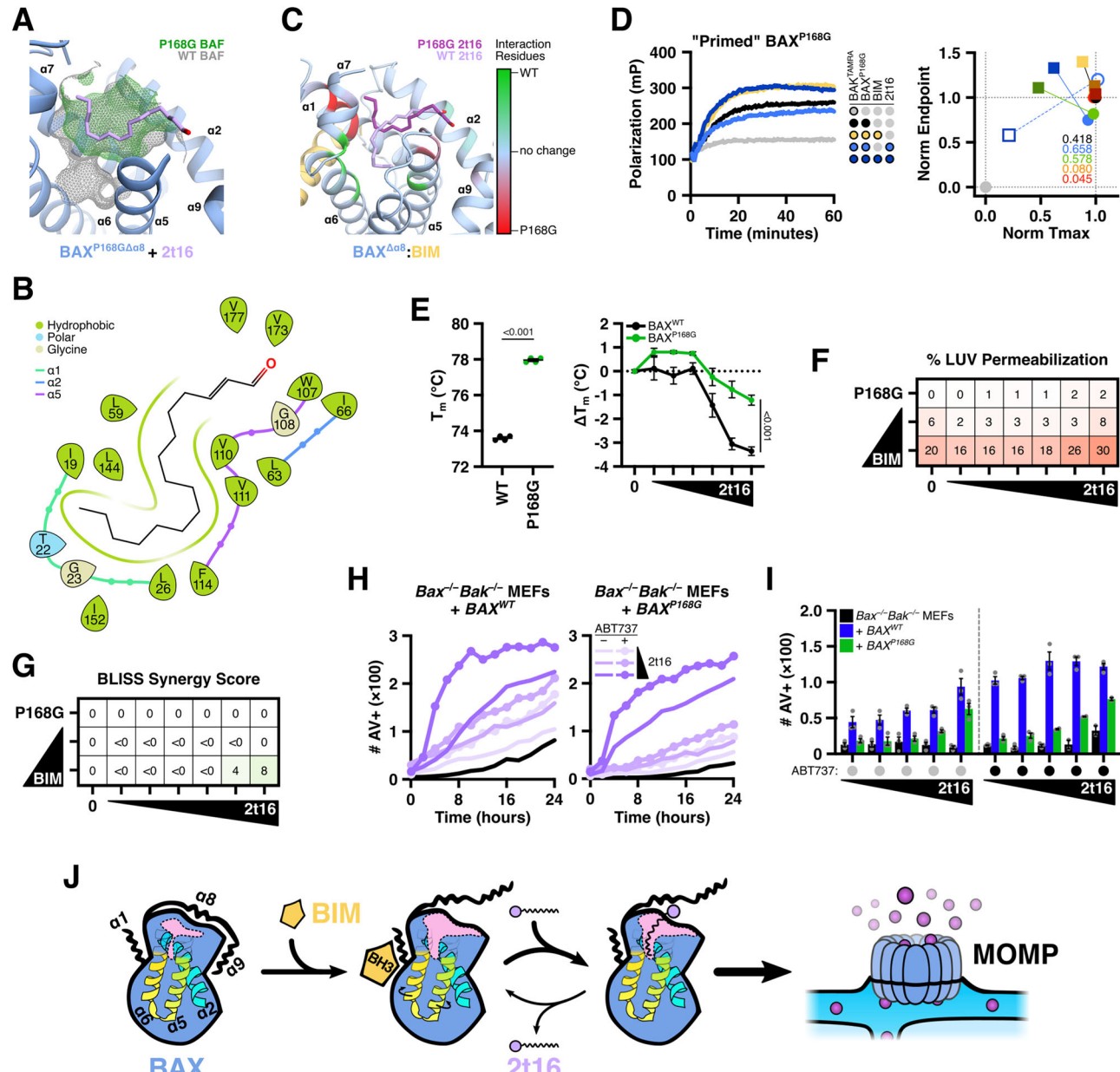

**Fig. 7 | Mutation of proline 168 allosterically disrupts the BAF and 2t-hexadecenal function. A–C** Molecular docking and modeling of BAX^P168GΔα8 (PDB: 5W60, Δ157–163) with 2t16 using Schrödinger Glide with a binding region localized to the BAF. **A** The BAX^P168G structure includes divergent rotamers of bulky core residues that removes the neck of the BAF. The highest scoring pose is shown and is representative of generated poses. Surface of exposed BAF shown in green wireframe and clipped to aid in visualization. The BAF surface from WT BAX is overlaid in grey for comparison. **B** The interaction diagram from (**A**) reveals that 2t16 interacts with distinct residues, including sidechains of α1. **C** Residues interacting with 2t16 were compared between WT and P168G BAX and colored green or red, respectively, according to the number of poses and specificity for one of the isoforms. Structure of BAX^Δα8:BIM included for visualization as well as top scoring 2t16 poses. **D** Left: BAX^P168G (65 nM) was combined with non-activating concentrations of BIM-BH3 peptide (0.15 μM) and 2t16 (3 μM), followed by BAK^TAMRA (50 nM), and subjected to FLAMBE. Right: Parameterized FLAMBE data including four concentrations of 2t16 (blue: 3 μM; green: 4.5 μM; orange: 6.5 μM; red: 10 μM) in the absence or presence of BIM-BH3 (circle and square datapoints, respectively). Annotations report the magnitude of shift between data with and without BIM-BH3. Empty symbols denote parameterized data for BAX^WT shown in Fig. S9A and is included for comparison. **E** The melting temperature of BAX^WT and BAX^P168G ± 2t16 (6.5–50 μM) was measured by thermal shift assay using SYPRO orange and

compared using four replicate samples. Individual values, the mean, and SEM are shown. Statistical significance was determined by paired t test and the *P* value is reported. Data shown are representative of replicated experiments. **F** Heatmap visualization of normalized endpoint LUV permeabilization data from LUVs incubated with the BAX^P168G treated with 2t16 (6.5–50 μM) ± BIM-BH3 peptide (0.15 μM). Data summarized from Fig. S9C. **G** Bliss synergy scores calculated using the endpoint LUV permeabilization data from (**F**) and (**S9C**). **H, I** *Bax⁻/⁻Bak⁻/⁻* double knockout MEFs reconstituted to express BAX^WT or BAX^P168G were subjected to SPARKL analysis measuring real-time labeling with fluorescently-tagged Annexin V (100 μg/ml) via imaging with a Cytation 7. Data are reported as fluorescence-positive events per image. **H** Kinetics of cell death for reconstituted MEFs treated with 2t16 (10 – 30 μM) and co-treated with DMSO vehicle or ABT-737 (1 μM); black lines report vehicle control. Data are the mean of replicates and error bars are omitted for visualization. **I** Comparison of Annexin V labeling at 18 h for parental *Bax⁻/⁻Bak⁻/⁻* MEFs and BAX^WT or BAX^P168G reconstituted MEFs. Data shown are the mean of independently-treated triplicates with error bars reporting SEM and are representative of replicated experiments. **J** A cohesive model of protein and lipid contributions to BAX activation. A cross section of BAX is illustrated to visualize changes to the hydrophobic funnel shape and accessibility (highlighted in pink). See also Fig. S9.

comparing between poses of hexadecenal docked to WT and P168G structures revealed distinct interaction signatures favoring interactions with α5/α6 or α1 residues, respectively (Fig. 7C).

We reasoned that disruption of the BAF in BAX[P168G] would inhibit interactions with hexadecenal. Using our FLAMBE assay, we demonstrated that BIM-primed BAX[P168G] did not activate in response to hexadecenal as compared to BAX[WT] (Figs. 7D and S9A). Furthermore, BAX[P168G] exhibited a reduced shift in melting temperature due to hexadecenal (Fig. 7E). As previously reported, BAX[P168G] was less sensitive to BIM activation and resulted in attenuated membrane permeabilization, but notably was not entirely functionally dead (Fig. S9B). In contrast, BAX[P168G] demonstrated no activation in response to hexadecenal and only minor synergy with BIM at activating concentrations, suggesting that activation-induced molecular rearrangements are the limiting factor for the synergistic function of hexadecenal (Figs. 7F, G and S9C). As detailed by others[40], we also observed a significantly higher melting temperature in BAX[P168G] compared to BAX[WT], suggesting a stabilization of the monomeric conformer (Fig. 7E). Finally, we replicated our apoptosis experiments in Bax[-/-]Bak[-/-] DKO MEFs transduced to stably express either BAX[WT] or BAX[P168G] (Fig. S9D). The DKO MEFs reconstituted with BAX[WT] exhibited greater sensitivity to ectopic hexadecenal compared to BAX[P168G]-expressing MEFs (Fig. 7H). Furthermore, this difference was maintained in MEFs co-treated with hexadecenal and ABT-737 (Fig. 7H, I). Taken together, these results suggest that BAX[P168G] is insensitive to hexadecenal and synergy with BIM due to dysfunction of the BAF. Furthermore, we posit that the increased flexibility in the α8–9 loop (by substituting a rigid proline for glycine) may also alter the mobilization of α8 and subsequent BAF exposure following BIM triggering in addition to deforming the BAF structure through allosteric sidechain reorganization.

## Discussion

Several decades have been devoted to characterizing the protein-protein interactions that govern the BCL-2 family and mediate the onset of MOMP following pro-apoptotic signaling. More recently, a growing focus on the mitochondrial contributions to potentiate MOMP identified the sphingolipid metabolites S1P and 2t-hexadecenal as signaling lipid species to modulate the BCL-2 effector proteins, BAK and BAX, respectively[17]. Specifically, several approaches suggested that hexadecenal cooperated with direct activators to activate native BAX prior to membrane associations, and therefore we further investigated the underlying mechanism. Here, we demonstrate a non-covalent interaction by which 2t-hexadecenal directly promotes BAX activation (Figs. 1 and 2) and synergizes with BIM to enhance early-activation steps of monomeric BAX prior to translocation or oligomerization (Fig. 3). We determined that 2t-hexadecenal interacts within a hydrophobic funnel-shaped cavity in the core of BAX—formed residues of α1, α5, α6, and capped by α8, which we now term the "BAX Actuating Funnel" or BAF—and that this region is exposed and enlarged following BIM triggering (Fig. 4). We identified chemical and structural determinants of the 2t-hexadecenal:BAF interaction, including aldehyde length (Fig. 5) and specific BAF residues (Fig. 6), and further revealed that proline 168 allosterically regulates the BAF and subsequent mutation reduces BAF shape and availability (Fig. 7). We therefore propose a cohesive model for successive BAX activation: BIM-mediated triggering induces α9 mobilization as well as tandem α8 movement, resulting in accessibility of the BAF, interactions with 2t-hexadecenal, and promoting subsequent conformational changes and pore formation (Fig. 7J).

Previously, two studies reported that 2t-hexadecenal covalently modifies BAX cysteine residues through Michael addition, albeit on different cysteines, and we propose a few theories as to why the investigations yielded divergent conclusions. One study observed alkylation by 2t-hexadecenal at BAX C126 in solution and aldehyde-containing LUVs[22]. A possible explanation for the observed

modification in solution could be the use of millimolar concentrations of 2t-hexadecenal, which the authors report induces the reaction and BAX oligomerization in solution. When studying BAX pore formation, model liposomes were formulated with upwards of 10% 2t-hexadecenal, which may have facilitated covalent modification and altered the behavior of the membrane in response to BAX activation. Another study synthesized a clickable alkyne analogue of 2t-hexadecenal and observed modification of BAX C62 in cell lysates and recombinant C-terminal truncated protein[21]. This work utilized buffers containing NP-40 nonionic detergent at micelle-forming concentrations, which is known to directly promote monomeric BAX activation[2,43]. Interestingly, C62, which sits within the BH3 residues of BAX α2, is predominantly core-facing and not likely to be solvent exposed in inactive BAX structures, but is likely exposed along with BH3 residues following triggering[4,44]. As such, perhaps either the NP-40 detergent and/or removal of BAX α9 facilitated C62 availability to the alkyne probe. Moreover, both works utilized auxiliary chemistries as part of their aldehyde capture and detection (e.g., hydrazide labeling, click chemistry), and we cannot rule out the possibility that additional differences in solvents, buffer formulations, reagents, or pH limited our ability to replicate the covalent reaction within our investigation.

Critically, we do not dispute the findings of these prior studies nor the conclusion that cysteine modification modulates BAX activity[45]. There is substantial evidence that 2t-hexadecenal and α,β-unsaturated aldehydes are inherently reactive molecular species that form adducts with a variety of macromolecules[24,27]. In fact, over 500 proteins were identified to be modified by the 2t-hexadecenal analogue probe[21], highlighting the importance of regulating cellular 2t-hexadecenal. In a cell, 2t-hexadecenal is promptly metabolized by FALDH/ALDH3A2 and as such, cellular BAX is unlikely to experience millimolar concentrations of 2t-hexadecenal during activation in the cytosol[18,46,47]. By comparison, the confluence of sphingolipid metabolism, lipid transfer, and BAX apoptotic foci forming at MERCS likely serves to regulate and concentrate the availability of 2t-hexadecenal during BAX functionalization at the OMM[48]. The structure of active BAX multimers reveals that C62 is readily solvent accessible—where it could be targeted by the ectopic hexadecenal alkyne and C126 faces the membrane—where it could react with membrane-integrated hexadecenal as previously described (Fig. S10A–C)[22]. Therefore, we posit that the prior works identified covalent modification of functionalized BAX at the OMM, differing in cysteine target based on their chemical approach, while we identified a non-covalent interaction mechanism by which 2t-hexadecenal synergizes with BIM to active monomeric BAX. Of note, the precursor of 2t-hexadecenal, S1P, is not localized only to membranes and does function as a soluble moiety, including reports as a non-covalent cofactor; as such, it is reasonable to expect that 2t-hexadecenal can similarly operate in the cytoplasm and distinct from membranes[49–51].

So how does 2t-hexadecenal binding to the BAF promote BAX activation? One possible explanation is that residence of 2t-hexadecenal in the BAF occupies space that would otherwise accommodate bulky aromatic residues of α8. Activated BAX monomers can be "reset" by anti-apoptotic BCL-2 protein (e.g., BCL-xL)[52,53] and perhaps the steric interference of 2t-hexadecenal prevents replacement of α8 and α9 into the BAF and BC groove, respectively. Another explanation could be that binding of 2t-hexadecenal, particularly in the neck of the BAF, which resides deep in the core of BAX, is inherently destabilizing to the packing of sidechains and core helices. Therefore, 2t-hexadecenal binding may aid separation of the core and latch domains and facilitate forming multimeric conformers. Additionally, residence in the BAF may also aid BAX activation by promoting exposure of the BAX BH3 residues, which are core-facing in the inactive conformer, through steric interference with α2[4,44,54].

Interestingly, the concept of a lipid-interacting funnel cavity capped by a gatekeeper helix is also present in the only other protein

known to non-covalently bind 2t-hexadecenal−FALDH. In FALDH, the substrate site is a deep hydrophobic funnel capped with a short gatekeeping helix, which is hypothesized to participate in substrate specificity[55]. The gatekeeper helix has several aromatic residues and is stabilized through interactions with the preceding helix, which has a perpendicular orientation, and a lateral loop region, in a parallel orientation (similar to BAX α7 and α4−5 loop, respectively). Like BAX, the FALDH gatekeeper helix is the penultimate helix, and is followed by a short linker and a C-terminal helix that serves as a transmembrane domain, orienting the entrance of the funnel towards the residential membrane. Of note, FALDH functions as a symmetrical homodimer with the stabilizing helix, gatekeeping helix, and transmembrane helix folding on the *trans* subunit, bearing a striking similarity to the domain-swapped BAX dimer structure[6]. The specificity of FALDH is attributed to the cooperation of the hydrophobic loop, the stabilizing helix, and the gatekeeper helix that collectively surround the substrate funnel; and it is tempting to speculate that the similar features in the BAX structure may likewise play a role in 2t-hexadecenal recruitment and specificity.

Identification of the BAF provides additional insights into the BAX structure-function relationship. Monomeric BAX conformers must undergo substantial unfolding to adopt multimeric structures, and therefore, from a structural perspective, "activating" BAX must sufficiently destabilize the monomer to enable refolding into domain-swapped conformations. Interestingly, several features of the BAX structure support this conceptualization of activation. For example, BAX has no disulfide bonds and evolved to fold entirely through hydrophobic/hydrophilic and electrostatic interactions, which decreases the energy requirement to transition between stable conformers (e.g., monomer to domain-swapped dimer). Additionally, cavities or voids destabilize globular proteins[56,57], and BAX studies mutating bulky core residues with smaller side chains have observed increased sensitivity to activation[39], likely due to enlarging the BAF and reducing stability[58,59]. Indeed, we observed this phenomenon when we mutated F114 or F143 during our attempts to identify selective BAF mutants. Of note, the cavity has also been observed in the BH3-in-groove:BAX domain-swapped dimer structures (e.g., PDBs 4BD2, 4ZIE, and 4ZIH)[6,60]; albeit the location is more central to α2, α5, and α8 (i.e., the "mouth" of the BAF), and may be influenced by the removal of the C-terminus (Fig. S10D). One interesting insight is that BAX$^{\Delta C26}$ and BAX$^{\Delta C28}$ exhibited a diminished cavity due to a shift in the α2 helix[60], suggesting that bulky residues within and proximal to α8 (e.g., F165 and W170) may aid in maintaining the cavity through interactions with α2. Additionally, we explored the allosteric influence of P168 on BAF shape and availability in this work. The presence of the cavity in several solved apo and BH3-bound BAX structures strongly suggests its importance to the structure-function relationship of BAX and supports the destabilization concept of BAX activation.

Our model suggests that activation-induced mobilization of α9 also results in displacement of α8, which in turn exposes the BAF and hydrophobic core. Therefore, we propose that α8 functions as a gatekeeper helix capping the cavity, limiting solvent exposure of hydrophobic residues, and ultimately stabilizing the monomeric conformation of BAX through interactions within the helical bundle. This model is supported by the fact that the P168G mutation, which increases flexibility in the α8−9 loop and we propose limits α8 mobility, exhibits increased protein stability as the inactive, globular monomer. Furthermore, several studies have demonstrated that disruption of α8 is deleterious to monomeric BAX stability: a proline substitution in α8 (L161P) caused oligomerization and cell death[61]; removal of α8 and α9 (BAX$^{\Delta C35}$) resulted in oligomerization[62]; C-terminal truncations (BAX$^{\Delta C21}$, BAX$^{\Delta C25}$, BAX$^{\Delta C28}$) remained monomeric but deletion of Y164 (BAX$^{\Delta C29}$) caused oligomerization[6]. Furthermore, one of our BAX$^{\alpha8}$-locked mutagenesis strategies (BAX$^{W107C,Y164C}$) resulted in recombinant protein purifying primarily as

oligomeric species (data not shown). Notably, W107 and Y164 are predicted to exhibit π-stacking as are several aromatic residues localized to the α4−5 loop and α7/α8. Interestingly, these regions of BAX are the BH1 and BH2 domains, suggesting that their conservation within multidomain BCL-2 family proteins is due, at least in part, to their role in stabilizing the helical bundle via the α8 cap.

In conclusion, we describe a cohesive model of BAX activation, in which successive contributions of protein-protein and protein-lipid interactions promote intramolecular maturation of activated BAX monomers into functional oligomers. We propose that interactions between 2t-hexadecenal and the exposed BAF can occur either in the cytosol, as we have modeled herein, or at the OMM. Furthermore, BAX appears to be regulated by 2t-hexadecenal both biophysically (i.e., interactions with the BAF) and biochemically (i.e., cysteine modification) at discrete stages and environments throughout the BAX activation continuum. Interestingly, while activated BAK structures bind and incorporate lipid moieties, analogous studies have not identified a similar mechanism in BAX multimers suggesting that there may be a divergent mechanism[63,64]. BAX is distinct in that much of its regulation occurs in solution prior to membrane association and therefore the lipid contributions to BAX activation may function primarily during monomeric activation instead of oligomerization. Finally, this work describes the function and importance of the BAF, which had previously been identified but its significance was never determined[6,60]. The BAF represents an unexplored regulatory region that is critical for understanding monomeric BAX structure-function biology as well as providing a space for pharmacologically targeting BAX. The BAF is exposed following BIM- or BID-induced BAX activation and thus we suggest that BAF-targeting molecules would be particularly adept at modulating BAX in cellular contexts that actively maintain direct activator BH3-only protein function (e.g., "primed" cells) while being well tolerated in healthy tissues where the BAF is unexposed.

## Methods

### Experimental models
**Bacterial cell lines.** For recombinant protein generation, One Shot$^{TM}$ BL21 (DE3) Chemically Competent *E. coli* were purchased from Invitrogen/Thermo Fisher Scientific (Cat. No. C600003). Cells were grown in BD Difco$^{TM}$ Terrific Broth (Cat. No. BD243820, Thermo Fisher Scientific) media supplemented with 1% glycerol at 37 °C with shaking at 220 rpm.

### Cell lines
*Bax$^{+/+}$Bak$^{+/+}$* and *Bax$^{-/-}$Bak$^{-/-}$* SV40-transformed MEFs were obtained from ATCC (Manassas, VA, USA); *Bim$^{-/-}$Bid$^{-/-}$* SV40-transformed MEFs were provided by Dr. Douglas Green (St. Jude Children's Research Hospital); *Bax$^{-/-}$Bak$^{-/-}$* double knockout MEFs reconstituted to express wild type or P168G BAX were provided by Dr. Evripidis Gavathiotis (Albert Einstein College of Medicine) and BAX expression was confirmed by western blot and GFP positivity from the pBabe IRES-GFP vector backbone. Cells were cultured in high-glucose DMEM (Cat. No. 10-017-CV, Corning, Manassas, VA, USA) containing 10% FBS (Cat. No. 10438-026, Life Technologies, Carlsbad, CA, USA), 2 mM L-glutamine (Cat. No. 25030-081, Life Technologies, Carlsbad, CA, USA), and 1× Penicillin/Streptomycin (Cat. No. 10378-016, Life Technologies, Carlsbad, CA, USA), and grown in humidified incubators at 37 °C with 5% CO2. All cell cultures were maintained in mycoplasm-free conditions as verified by the HEK-Blue Detection Kit (Cat. No. hb-det2, Invivogen, San Diego, CA, USA).

### Recombinant protein and peptides
**Expression vectors and site-directed mutagenesis.** Recombinant human BAX was expressed and purified using an intein-chitin binding domain (CBD) tag from a pTYB1 vector (BAX cDNA inserted into Nde/SapI cloning site, with a C-terminal tag)[44]. BAX mutants (e.g., C62S and

C126S double mutant, $BAX^{2S}$; L59C, C62S, C126S, and L162C quadruple mutant, $BAX^{\alpha8}$) were generated by site-directed mutagenesis of the WT construct. Amplification of the de novo plasmids was accomplished using CloneAmp HiFi PCR Premix (Cat. No. 639298, Takara Bio, Mountain View, CA, USA) and thermo-cycled as follows: 1× [98 °C for 2 min]; 30× [98 °C for 10 s, 55 °C for 30 s, 72 °C for 8 min]; and 1× [72 °C for 10 min]. Parental plasmids were digested by 1 µl DpnI enzyme (Cat. No. R0176, New England Biolabs, Ipswich, MA, USA) at 37 °C for 15 min, followed by purification using the QIAquick PCR Purification Kit (Cat. No. 28104, Qiagen, Hilden, Germany). Sequences were verified by Genewiz Sanger sequencing using the Universal T7 primer (5′-TAA-TACGACTCACTATAGGG-3′).

**Recombinant BAX expression and purification.** One Shot™ BL21 (DE3) chemically competent *E. coli* (Cat. No. C601003, Thermo Fisher Scientific) were transformed with the pTYB1-BAX construct in LB broth and grown on agar plates supplemented with 100 µg/mL carbenicillin at 37 °C. Starter cultures were grown in Terrific Broth (TB) supplemented with 0.4% glycerol and 100 µg/mL carbenicillin at 30 °C for 14−16 h to an optical density at 600 nM ($OD_{600}$) of approximately 1.5; then diluted with 4× volume of TB broth and expanded at 37 °C for 3−4 h until the culture achieves a target $OD_{600}$ of 2−3. Recombinant BAX expression was induced with 1 mM IPTG at 30 °C for 6 h. Bacteria were pelleted, resuspended in lysis buffer (50 mM $K_2PO_4$, 50 mM $NaH_2PO_4$, 500 mM NaCl, 1 mM EDTA, 0.1 mM AEBSF) supplemented with a Pierce protease inhibitor tablet (Cat. No. A32965, Thermo Fisher Scientific), and lysed with a probe sonicator (e.g., Dismembrator 505, Thermo Fisher Scientific) for 20 min. Lysates were centrifuged at $42,000 \times g$ at 4 °C for 1 h to pellet debris and recombinant intein-CBD-tagged BAX was captured from the supernatant by chitin affinity chromatography at 4 °C according to the manufacturer's instructions (Cat. No. S5551, New England Biolabs, Ipswich, MA, USA). On-column intein cleavage of the CBD tag was achieved by incubation with DTT (50 mM) at 4 °C for at least 16 h, followed by elution in Gel Filtration Buffer (20 mM HEPES [pH 7.4], 150 mM NaCl). Recombinant BAX protein (now without a tag or exogenous residues) was further purified by size-exclusion chromatography using Gel Filtration Buffer on a HiLoad 16/600 or 26/600 Superdex 200 pg column (Cat. No. 28989335, Cytiva) using an ÄKTA pure™ 25 L1 (Cat. No. 29018225, Cytiva) at 4 °C according to the manufacturer's instructions. Fractions containing BAX protein were combined and concentrated using Amicon Ultra-4 centrifugal filter (Cat. No. UFC801024, Millipore Sigma) to a stock concentration of ~30 µM, prior to snap freezing with liquid nitrogen, and storage at −80 °C. A detailed version of this protocol has been published[65].

**Oxidizing disulfide tether BAX mutants.** For the $BAX^{\alpha8}$ mutant, the nascent protein was purified as the reduced form due to incubation with DTT during on-column intein cleavage; to generate a disulfide tether ("locked $BAX^{\alpha8}$"), BAX protein was oxidized in 1 mM Dichloro(1,10-Phenanthroline) Copper (II) (Cat. No. 362204, Millipore Sigma-Aldrich) at 4 °C for 15 min, purified by Superdex 75 Increase 10/300 GL column (Cat. No. 29148721, Cytiva), and only monomeric species were pooled and concentrated to make protein stocks.

**Other reagents.** Additional recombinant proteins and peptides were purchased from commercial sources: 5-TAMRA labeled BAK-BH3 (Cat. No. AS-64590, AnaSpec); recombinant Bcl-xL$^{\Delta C}$, aa 2−212 (Cat. No. 894-BX, R&D Systems); BIM-BH3, Peptide IV (Cat. No. AS-62279, AnaSpec).

**Lipid aldehyde preparation**
2-*trans*-hexadecenal (16:1 aldehyde) was purchased in powder form (Cat. No. 857459 P, Avanti Research) and reconstituted in pre-warmed (37 °C) anhydrous DMSO (Cat. No. D12345, Thermo Fisher Scientific) to a concentration of 50 mM using a Hamilton syringe. The reconstituted

2t-hexadecenal was aliquoted into single-use glass aliquots and stored at −80 °C. To avoid precipitation/coagulation of 2t-hexadecenal, stocks were serially diluted first in pre-warmed DMSO, then diluted into pre-warmed aqueous assay buffer to use as a stock for the experiment. This was typically accomplished as follows: 2.5 µl of 50 mM 2t-hexadecenal stocks were two-fold diluted four times in DMSO (to 40 µl of 3.13 mM), then diluted twice in the selected assay buffer (first with 40 µl, then with 45 µl) to achieve a working stock of 125 µl of 1 mM 2t-hexadecenal in 32% DMSO/buffer. Working stocks of 1 mM 2t-hexadecenal were placed in a water bath sonicator for 15 min to aid in solubilization and then further diluted to the desired concentration and acceptable DMSO background for experiments. For cell death experiments, 2t-hexadecenal was prepared in 0% FBS media to prevent binding to serum proteins (see SPARKL method below). For aldehyde modification, (2-hydroxyethyl)hydrazine (Cat. No. 043793.09, Thermo Fisher Scientific) was added to stock 2t-hexadecenal at a ratio of 30−50 molar excess and reacted at 37 °C for 1 h. The mass of the resultant hexadecenal-hydrazino product (2t16-NN-EtOH) was confirmed by mass spectrometry (see below) and used to validate the reagent for BAX function assays.

Hexadecanal (16:0 aldehyde) was purchased as 1 mg/ml (4.16 mM) in methylene chloride solution (Cat. No. 857458M, Avanti Research), aliquoted into 25 µl, and stored at −80 °C. Prior to use, hexadecanal was first diluted with pre-warmed DMSO to a concentration of 3 mM, then diluted to 1 mM with pre-warmed assay buffer of choice, sonicated for 15 min, and then further diluted for experiments. The remaining alkenals were purchased in liquid form (see Table 1), and their molar concentrations were calculated based on their density, purity, and molecular weight. Short-chain alkenals (2t5−2t10) were diluted with DMSO to a stock concentration of 1 M, aliquoted, and stored at −80 °C. Prior to use, stocks were further diluted with DMSO to 320 mM and then in assay buffer to a final concentration of 1 mM with 32% DMSO (v/v). Long-chain alkenals (2t11−2t13) were more likely to coagulate and thus were prepared following the exact same procedure as the 2t-hexadecenal preparation.

**Real-time live-cell detection of apoptosis (SPARKL)**
**Cell culture and treatments.** MEFs were seeded at $2−4 \times 10^3$ cells per well in 96-well tissue culture treated plates and left to adhere for 18−24 h. Prior to live-cell imaging, growth media was replaced with phenol red-free DMEM as follows: first, 50 µl (50% total volume) of 0% FBS DMEM containing 2t-hexadecenal treatments was added to wells for 15−30 min to allow uptake; next, 50 µl of 10% FBS DMEM containing additional treatments and fluorescently-labelled Annexin V (50 nM or ~1.8 ng/µl) was added to wells to reconstitute serum levels. Recombinant Annexin V was generated in-house as previously described[23,66,67]. Immediately following treatments, plates were subjected to real-time fluorescence microscopy, automated data collection, and analysis.

**Table 1 | Aldehyde reagents**

| Aldehyde | Supplier | Cat. No. |
|---|---|---|
| trans-2-Pentenal (2t5) | Sigma-Aldrich | W321818 |
| trans-2-Hexen-1-al (2t6) | Sigma-Aldrich | W256005 |
| trans-2-Heptenal (2t7) | Sigma-Aldrich | W316504 |
| trans-2-Octenal (2t8) | Sigma-Aldrich | W321508 |
| trans-2-Nonenal (2t9) | Sigma-Aldrich | W321399 |
| trans-2-Decenal (2t10) | Sigma-Aldrich | W236608 |
| trans-2-Undecenal (2t11) | Sigma-Aldrich | W342300 |
| trans-2-Dodecenal (2t12) | Sigma-Aldrich | W240206 |
| trans-2-Tridecenal (2t13) | Sigma-Aldrich | W308218 |
| 16:1 aldehyde (2E-hexadecenal, 2t16) | Avanti Research | 857459 P |
| 16:0 aldehyde (hexadecanal, 16CHO) | Avanti Research | 857458 M |

**Data acquisition and event detection with an IncuCyte ZOOM.** SPARKL experiments performed using an IncuCyte ZOOM (Model 4459, Sartorius, Göttingen, Germany) were housed in a humidified tissue culture incubator maintained at 37 °C with 5% $CO_2$. Images were collected every 2 h for 24 h using a 10× objective, and a single field of view was collected per well. Bright field and green channels (Ex: 440/80 nm; Em: 504/44 nm; acquisition time: 400 ms) were collected at 1392 × 1040 pixels at 1.22 μm/pixel. Automated event detection was accomplished using the ZOOM software (v2018A) and user-defined processing definition using images collected using the relevant cell lines and fluorescent reporters. Processing definition settings were as follows: Parameter, Top-Hat; Radius (μm), 25; Threshold (RFU), 3; Edge Sensitivity, −30; Area (μm²), >100. Kinetic data are graphed as the calculated events per well metric provided by the ZOOM software. The y-axis scale was determined for each experiment using parallel internal control treatments to assess maximal apoptotic death. Data are the mean of replicates and are representative of at least three repeated and reproduced assays. A detailed explanation of this method[23] and protocol[68] have been published.

**Data acquisition and event detection with a Cytation 7.** SPARKL experiments performed using a Cytation 7 equipped with an inverted imager (Model CYT7UW-SN, BioTek/Agilent, Santa Clara, CA, USA) tethered to a BioSpa 8 automated incubator (Model BIOSPAG-SN, BioTek/Agilent, Santa Clara, CA, USA), where cells were maintained in a humidified environment at 37 °C with 5% $CO_2$. Bright field and red channel images were collected every 2 h for at least 24 h using a 10× objective with a 75% wide field of view crop (1045 × 1045 μm), and a single field of view was collected per well using a laser autofocus (Part No. 1225010). Bright field images were acquired with the following settings: LED intensity, 7; Integration Time, 100 msec; Camera Gain, 19. Red channel images (Ex: 586/18 nm; Em: 647/57, Part No. 1225102) were acquired with the following settings: LED intensity, 8; Integration Time, 20 msec; Camera Gain, 19. Automated red event detection was accomplished using the Gen5 software (v3.12) and a cellular analysis data reduction profile with the following settings: Threshold, 5000 RIU; Background, Dark; Split Touching Objects, yes; Fill holes, yes; Background Flattening Size, auto (270 μm); Image Smoothing Strength, 0; Background Percentage, 5%; Minimum Object Size: 5 μm; Maximum Object Size: 90 μm. Kinetic data are graphed as events per image. The y-axis scale was determined for each experiment using parallel internal control treatments to assess maximal apoptotic death. Data are the mean of replicates and are representative of at least three repeated and reproduced assays.

**Large unilamellar vesicle (LUV) permeabilization assays**
**LUV composition and preparation.** LUVs and permeabilization assays were prepared as similarly described[9,69]. Briefly, chicken egg phosphatidylcholine (Cat. No. 840051C, Avanti Research), chicken egg phosphatidylethanoloamine (Cat. No. 840021C, Avanti Research), porcine brain phosphatidylserine (Cat. No. 840032C, Avanti Research), bovine liver phosphatidylinositol (Cat. No. 840042C, Avanti Research), and cardiolipin (18:1) (Cat. No. 710335C, Avanti Research) were combined at a ratio of 48:28:10:10:4 (5 mg total), dried under $N_2$ gas, and resuspended in 500 μl LUV buffer (10 mM HEPES [pH 7], 200 mM KCl, 5 mM $MgCl_2$, 0.2 mM EDTA) containing a polyanionic dye (12.5 mM ANTS: 8-aminonaphthalene-1,3,6-trisulfonic acid) and cationic quencher (45 mM DPX: p-xylene-bis-pyridinium bromide) using a water bath sonicator. Unilamellar vesicles were formed by 31 extrusions of the suspension through 1.0 μm polycarbonate membranes (Cat. No. 610010, Avanti Research, Alabaster, AL, USA). The unincorporated DPX and ANTS were removed by using a 10 ml Sepharose S-500 gravity flow column. LUVs were used within 2 weeks of preparation to avoid significant liposome degradation.

**LUV assays.** Working stocks of recombinant proteins, peptides, and lipids were prepared in LUV buffer at 4× their intended final concentrations. LUVs from the preparation stock (20×) were diluted five-fold in LUV buffer to generate a working stock (4×) for assays (equivalent to each sample receiving 5 μl undiluted LUVs). In an opaque black polystyrene 96-well plate (Cat. No. 3915, Corning), lipids or peptide titrations were generated via in-well serial dilutions and LUV buffer was added to any control wells requiring volume compensation. Samples were assayed in 100 μl total volume with 25 μl of each component sequentially added to achieve desired 1× concentrations. A typical assay was prepared as follows: 25 μl of titrant is prepared in the well; then 25 μl of peptide, reagent, or buffer is added; then 25 μl of recombinant BAX is added; and finally, 25 μl of diluted LUVs. Plates were immediately subjected to fluorescence analyses for 90 min using either a Synergy H1 or Cytation 5 multi-mode microplate reader (Agilent/BioTek, Santa Clara, CA, USA) using the following parameters: Read interval, 55 sec; Protocol, [37 °C, 2 sec linear shake, read]; Excitation, 355/20 nm; Excitation, 520/20 nm; Gain (voltage), 100; Optics position, top; Read height, 5.5 mm. Kinetic data represents the mean of triplicate samples and are representative of at least three repeated and reproduced assays using separate protein aliquots and LUV preparations. Every assay included a 1% CHAPS positive control to determine maximum signal. Normalized endpoint data (% permeabilization) were calculated in Prism (Graphpad) using the minimum value of the buffer control (as 0%) and the mean value of LUVs solubilized in 1% CHAPS (as 100%) measured during the entire assay (Equation 1). Appropriate concentrations of BAX were determined by extensive protein titrations for each preparation of recombinant protein and LUVs; typically, the highest BAX concentration exhibiting minimal permeabilization was selected for subsequent assays to ensure adequate signal and dynamic range for activation studies. Data are the mean of replicates and are representative of at least three repeated and reproduced assays.

**LUV data analysis.** Kinetic LUV permeabilization was fit with Prism (Graphpad) nonlinear regression analysis using a modified 5-parameter sigmoidal model (Equation 2), with S ≤ 10 and T ≥ B. The formula for maximal rate, or slope at the inflection point (IP), was derived from the second derivative of the function and incorporated into the analysis report as an exported metric (Equation 3). Bliss synergy scores were determined for normalized endpoint LUV permeabilization data using the SynergyFinder Plus web application (www.synergyfinder.org) and graphed in Prism[70].

Equation 1: LUV data normalization

$$\% \, permeabilization = \left(\frac{F - F_0}{F_{100} - F_0}\right) \cdot 100$$

Equation 2: LUV kinetic data nonlinear regression fitting

$$y = B + \left(\frac{T - B}{\left(1 + \left(\frac{E}{X}\right)^H\right)^S}\right)$$

Equation 3: Maximal rate of fitted LUV kinetic data

$$\left(\frac{\delta y}{\delta x}\right)_{X = X_{IP}} = \frac{H \cdot S \cdot (T - B) \cdot \left(\frac{H+1}{(H \cdot S)-1}\right)^{\left(\frac{H+1}{H}\right)}}{E \cdot \left(1 + \frac{H+1}{(H \cdot S)-1}\right)^{S+1}}$$

**Microscale thermophoresis (MST)**
MST experiments were carried out on a NT.115 (NanoTemper GmbH, Munich, Germany) instrument and performed in MST buffer (50 mM Tris-HCl pH 7.4, 150 mM NaCl, 10 mM $MgCl_2$, 0.05% Tween-20). EC50 values were calculated using the NanoTemper software. The recombinant human BAX protein was labeled with Alexa Fluor-647 NHS ester

by incubating 60 μM of the fluorescent dye with a 20 μM of BAX protein for 30 min in the dark. The labeling mixture was subsequently applied to a G-25 gravity flow column (GE Healthcare) that had been equilibrated with MST buffer. After elution from the G-25 column, the concentration of the labeled protein was quantified spectro-photometrically, snap frozen in liquid nitrogen, and stored as single use aliquots at −80 °C. Labeled BAX protein was treated with 0.002% CHAPS to prevent activation-induced BAX aggregation upon ligand binding. Curve fitting was conducted using Prism (Graphpad) and only applied to X values from 5–35 s using Equation 4, and the K constant was calculated using Equation 5. Data are representative of at least three repeated and reproduced assays.

Equation 4: One-phase decay function for MST data fitting

$$y = (Y_0 - Y_{Plateau})(e^{-K \cdot x}) + Y_{Plateau}$$

Equation 5: Decay constant function

$$K = \frac{-\ln\left(\frac{y - Y_{Plateau}}{Y_0 - Y_{Plateau}}\right)}{x}$$

### Liquid chromatography-mass spectrometry (LC-MS)

**Sample preparation and handling.** Working stocks of recombinant BAX protein and 2t-hexadecenal were prepared in Gel Filtration Buffer. 30 μl of 5 μM recombinant BAX (~3 μg) was treated as indicated in duplicate and incubated at 37 °C for 2 h. Samples were then incubated in 8 M urea buffer (with 50 mM $NH_4HCO_3$) at 57 °C for 1 h, and alkylated in the dark with 30 mM iodoacetamide for 30 min at 25 °C. Samples were then digested using 1:100 (w/w) trypsin at 37 °C for 4 h, with an additional round of 1:100 trypsin at 37 °C for 16–18 h. After proteolysis, the peptide mixtures were desalted by self-packed stage-tips or Sep-Pak C18 columns (Waters) and analyzed with a Vanquish Neo UHPLC System that is coupled online with a Orbitrap Exploris 480 mass spectrometer (Thermo Fisher). Briefly, desalted Nb peptides were loaded onto an analytical column (C18, 1.7 μm particle size, 150 μm × 5 cm; IonOpticks) and eluted using a 20 min liquid chromatography gradient (3–10% B, 0–2 min; 10–40% B, 2–12 min; 40–80% B, 12–14 min; 3% B, 14–20 min). Mobile phase A consisted of 0.1% formic acid (FA), and mobile phase B consisted of 0.1% FA in 80% acetonitrile (ACN). The flow rate was 1 μl/minute. The Orbitrap Exploris 480 instrument was operated in the data-dependent mode, where the top 20 most abundant ions (mass range 200–1800, charge state 2–6) were fragmented by high-energy collisional dissociation (HCD). The target resolution was 120,000 for MS and 15,000 for tandem MS (MS/MS) analyses. The maximum injection time for MS/MS was set at 100 ms. Data were replicated using different protein batches.

**Proteomic data analysis.** Raw data collected by LC-MS was searched against the Uniprot reviewed Human protein sequences database retrieved on 01 March 2024 with decoys and common contaminants appended using FragPipe (v22.0). A labile search was performed in MSFragger[71] without diagnostic and Y ions specified. Mass offsets were set restricted to cysteines. The offset list was set at 0 (no modification) and 238.22967 (monoisotopic mass of 2t-hexadecenal) and replacing the fixed cysteine carbamidomethylation with a variable one. "Write calibrated MGF" was turned on for the PTM-Shepherd[72] diagnostic feature mining module, and "Diagnostic Feature Discovery" in PTM-Shepherd was enabled with default parameters. Peptide and protein levels were performed using label-free quantification (LFQ) algorithms in IonQuant[73]. After MS search with MSFragger, raw files and identification files were imported into PDV[74] (https://github.com/wenbostar/PDV) for MS spectra annotation.

### Time of flight (TOF) mass spectrometry

**Small molecule mass spectrometry.** Small molecules were analyzed by using an Agilent 1200 series system with DAD detector and a 2.1 × 50 mm InfinityLab Poroshell 120 EC-C18 column for chromatography. Samples (0.5 μl) were injected onto a C18 column at room temperature with the flow rate of 0.6 ml/min. Chromatography was performed with the solvent as follows: water containing 0.1% formic acid and 3% acetonitrile was designated as Solvent A while acetonitrile containing 0.1% formic acid was designated as solvent B. The linear gradient was set such that 5% B was used from 0–0.5 min, 5–100% B from 0.5–2.8 min, 100% B from 2.8–4.2 min, 100–5% B from 4.2–4.3 min, and 5% B from 4.3–5 min. High-resolution mass spectra (HRMS) data were acquired in positive ion mode using an Agilent 6230 TOF with an electrospray ionization (ESI) source. Results were replicated in three repeated experiments.

**Native intact protein mass spectrometry.** Working stocks of recombinant BAX protein (10–20 μM) were treated with DMSO vehicle, 2t-hexadecenal (10– 200 μM), or iodoacetamide (50 mM) were prepared in 20 μl of 10 mM HEPES buffer and reacted for 1 h. Intact MS was performed by an Agilent 1200 series system with DAD detector and a 2.1 mm × 150 mm Zorbax 300 SB-C18 5 μm column for chromatography. Samples (10 μl) were injected onto a C18 column at room temperature with the flow rate of 0.4 ml/minutes. Chromatography was performed with the solvent as follows: water containing 0.1% formic acid and 3% acetonitrile was designated as Solvent A while acetonitrile containing 0.1% formic acid was designated as solvent B. The linear gradient was set such that 5% B was used from 0–1 min, 5–70% B from 1–6 min, 70% B from 6–8 min, and 5% B from 8–12 min. Mass spectra data was acquired in positive ion mode using an Agilent 6230 TOF with an electrospray ionization (ESI) source. MS spectra (m/z 600 – 1600) were deconvoluted by Agilent MassHunter BioConfirm v12.0 using a mass range of 16000 – 25000 daltons. Results were replicated in four repeated experiments.

### Thermal shift differential scanning fluorimetry

Working stock of recombinant BAX proteins, SYPRO Orange dye (Cat. No. S6650, Thermo Fisher Scientific), and 2t-hexadecenal were prepared in gel filtration buffer. The assay was performed in 100 μl 96-well PCR plates using a total assay volume of 50 μl. Twenty microliter of 2.5× 2t-hexadecenal titrations were generated via in-well serial dilutions and buffer was added to any control wells requiring volume compensation. 20 μl of 2.5 μM BAX and 10 μl of 20× SYPRO Orange were sequentially added to the wells, thoroughly mixed by pipetting, and the plate was centrifuged to recollect sample in the bottom of the well and remove any trapped bubbles. The PCR plate was then sealed with optically clear adhesive sheet and subjected to fluorescence spectroscopy using an Applied Biosystems ViiA7 real-time PCR instrument (Thermo Fisher Scientific). Temperature started at 25 °C and increased to 95 °C at a rate of 1% per minute. Data was collected as normalized fluorescence at each step of the thermal ramp and the melting temperature was determined as the maximum first derivative value. Data shown are the average of at least three replicates with error bars denote the SEM, and representative of at least three repeated and reproduced assays.

### Fluorescence polarization ligand assay to monitor BAX early-activation (FLAMBE)

**Assay setup.** Working stocks of recombinant BAX protein, peptides, and lipids were prepared in 0.5× PBS buffer at 4× their intended final concentrations. 5-TAMRA labeled BAK-BH3 peptide (Cat. No. AS-64590, AnaSpec) was diluted to a 200 nM working stock (4×). In an opaque black polystyrene 96-well plate (Cat No. 3915, Corning), lipids or peptide titrations were generated via in-well serial dilutions and buffer was added to any control wells requiring volume compensation.

Samples were assayed in 100 µl total volume with 25 µl of each component sequentially added to achieve desired 1× concentrations. A typical assay was prepared as follows: 25 µl of titrant is prepared in the well; then 25 µl of peptide, reagent, or buffer is added; then 25 µl of recombinant BAX; and finally, 25 µl of BAK$^{TAMRA}$. Plates were immediately subjected to spectrometry analyses for 60 min using a Synergy H1 multi-mode microplate reader equipped with a red polarization filter (8040562, Agilent/BioTek, Santa Clara, CA, USA) using the following parameters: Read interval, 30 s; Protocol, [25 °C, 2 s double orbital shake, read]; Excitation, 530/25 nm; Excitation, 590/35 nm; Mirror, 570 nm; Gain (voltage), 50; Optics position, top; Read height, 7.0 mm. Polarization (expressed as milli-Polarization (mP) units) is derived from measured parallel (∥) and perpendicular (⊥) emission intensities (I) and was calculated by the Gen5 software (Agilent/BioTek); Polarization can also be calculated manually using Equation 6. Kinetic data represents the mean of triplicate samples and are representative of at least three repeated and reproduced assays using separate protein aliquots. Appropriate concentrations of BAX and peptides were determined by extensive protein titrations for each preparation of recombinant protein; typically, the highest BAX concentration exhibiting no BAK$^{TAMRA}$ dissociation was selected for subsequent assays to ensure adequate signal and dynamic range for activation studies. Similarly, concentrations of direct activators were selected from extensive titration experiments to determine the appropriate concentration for each experimental setup. Data are the mean of replicates and are representative of at least three repeated and reproduced assays.

**Kinetic FP data parameterization.** Parameterized FLAMBE data was derived from the average of replicates to reduce data noise. For each condition, Tmax was identified as the timepoint with the highest Polarization and EP was the endpoint Polarization value recorded. Tmax of the BAK$^{TAMRA}$ control or any sample exhibiting no binding kinetics during the assay was set to 0 to avoid misidentification due to noise. Each parameterized metric was normalized to metrics from the BAK$^{TAMRA}$ and BAX control conditions, as 0 and 1, respectively. The shift magnitude of parameterized FLAMBE data was calculated using the distance between conditions without and with BIM-BH3 treatment (Equation 7). Area under the curve (AUC) was calculated using Prism (Graphpad) and normalized to the controls. A detailed explanation of this method[8] and protocol[75] have been published.

**Competition FP assay.** BCL-xL competitive binding FP assays were conducted similarly to FLAMBE assays with some modifications. BCL-xL, 2t-hexadecenal, and BAK$^{TAMRA}$ were prepared as 4× working stocks in a modified FP assay buffer (20 mM sodium phosphate buffer [pH 7.4], 50 mM NaCl, 1 mM EDTA, 0.05% pluronic F-68)[76]. 2t-hexadecenal titrations were prepared in-well and diluted with 25 µl buffer; separately, recombinant BCL-xL and BAK$^{TAMRA}$ stocks are combined at a 1:1 volume ratio (resulting in a 2× stock) and 50 µl is added to sample wells immediately followed by spectrometry.

Equation 6: Polarization calculation

$$mP = \left( \frac{I_{\parallel} - I_{\perp}}{I_{\parallel} + I_{\perp}} \right) \cdot 1000$$

Equation 7: Distance equation for parameterized FLAMBE data

$$\Delta s = \sqrt{(\Delta Tmax)^2 + (\Delta EP)^2}$$

## Fluorescent size-exclusion chromatography

Alexa Fluor™ 488 NHS ester (AF488)-labeled BAX was generated according to the manufacturer's instructions (Thermo Fisher Scientific, Cat. No. A20000) at a 20:1 dye:protein molar excess. Excess dye was removed by gel filtration with a Superdex 75 Increase 10/300 GL column (Cat. No. 29148721, Cytiva) equilibrated with Gel Filtration Buffer using an ÄKTA pure™ 25 L1 (Cat. No. 29018224, Cytiva) at 4 °C according to the manufacturer's instructions. 0.5 ml fractions were collected and fractions containing monomeric AF488-labelled BAX were combined, quantified, aliquoted, and stored at −80 °C. For experiments, 600 µl of 400 nM 488-labelled BAX was treated as indicated at 25 °C for 1 h and protected from light. Following treatment, BAX was subjected to size exclusion chromatography using the above parameters. For detection, 0.2 ml of eluate from the indicated fractions was transferred to a black 96 well-plate and analyzed for fluorescence by a Cytation 5 multi-mode plate reader (BioTek/Agilent, Santa Clara, CA, USA) using the following parameters: Excitation, 495/10 nm; Emission, 520/10 nm; Gain (volage), 100. Results were replicated in at least three repeated experiments.

## 2D HSQC NMR spectroscopy

$^{15}$N-labeled BAX protein was prepared at 40 µM in 15 mM sodium phosphate (pH 7.0), 50 mM NaCl. Samples were prepared and immediately subjected to spectroscopy. Data were acquired on Bruker 600-MHz spectrometer equipped with a cryoprobe. Two-dimensional $^{1}$H-$^{15}$N correlation spectra were acquired at 25 °C using standard Bruker pulse sequences using 128 scans, 2,048 × 200 complex points and spectral windows of 14 × 35 p.p.m. in the $^{1}$H and $^{15}$N dimensions, respectively. Samples did not exhibit any treatment-specific loss of signal or peak intensities suggestive of protein loss (due to either degradation or activation-induced multimerization). Spectra were processed using TOPSpin (Bruker Biospin, MA, USA) and analyzed with CARA software (cara.nmr.ch)[77]. Peaks were assigned using assignments from a previous publication[44]. The weighted average chemical shift perturbation (CSP) difference (Δδ) was calculated using Eq. 8. The absence of a value indicates no CSP difference, the presence of a proline, or a resonance that is overlapped or missing and therefore could not be confidently assigned and not used in the analysis. The significance threshold for CSPs was calculated based on the average chemical shift across all measurable residues plus 1 or 2 standard deviations, in accordance with standard methods. Mapping of CSP data onto the BAX structure (PDB: 1F16) was performed with PyMOL (Schrödinger, LLC).

Equation 8: CSP difference equation

$$\Delta\delta = \sqrt{\frac{\left(\Delta^1 H\right)^2 + 0.2\left(\Delta^{15}N\right)^2}{2}}$$

## In silico docking and molecular modeling

**Unconstrained global rigid-body docking.** The molecular interactions between the alkenals and BAX were modeled using the EADock DSS version of the SwissDock webserver (www.swissdock.ch)[78,79]. The structure PDB files and the alkenal mol2 files were obtained from the Protein Data Bank and the ZINC database, respectively (see Table 2). Output files were split into individual clusters and visualized using PyMOL (Schrödinger, LLC). Quantification of alkenal binding was accomplished by manual inspection of each cluster group and the total number of binding modality outputs for each job. Cavity determination and visualization was performed with PyMOL using a cavity radius of 3 and a cavity cutoff of −5.5; additional spaces were excluded to enhance visualization.

**Virtual mutagenesis and guided molecular docking.** Virtual mutagenesis of BAX structures was performed using the mutagenesis wizard in PyMOL (v3.1, Schrödinger, LLC) and selecting rotamers that did not conflict with surrounding sidechains; the resulting structures were utilized in downstream investigations. NMR-guided molecular docking of 2t-hexadecenal on BAX was performed using Glide (v2023-1 build 128, Schrödinger, LLC) with extra precision (XP) and a receptor

**Table 2 | List of resources for structural investigations**

| Structure | Identifier |
|---|---|
| BAX | PDB: 1F16[44] |
| BAX:BIM-BH3 | PDB: 2K7W[31] |
| BAX-P168G | PDB: 5W60[40] |
| 2t-hexadecenal | ZINC08217876 |
| hexadecanal | ZINC08216082 |
| 2t-pentenal | ZINC02031161 |
| 2t-hexenal | ZINC01531148 |
| 2t-heptenal | ZINC02017189 |
| 2t-octenal | ZINC02013450 |
| 2t-nonenal | ZINC01571215 |
| 2t-decenal | ZINC01571216 |
| 2t-undecenal | ZINC01849946 |
| 2t-dodecenal | ZINC01589935 |
| 2t-tridecenal | ZINC01613339 |

grid generated in the center of the BAF ($13 \times 13 \times 13$ Å inner box, 10 Å outer box), with no additional constraints. Wild type and mutant structures had α8 residues (157–163) removed and were prepared using the Protein Prep Wizard, assigning partial charges with EPIK (Schrödinger, LLC), and aligned against the 1F16 structure to ensure consistency of BAF coordinates for grid generation. The 2t-hexadecenal structure was converted to 3D and prepared for docking using LIGPREP (v2023-1 build 128, Schrödinger, LLC); the structure for the 2t-hexadecenal-hydrazino-product (2t16-NN-EtOH) was drawn in MAESTRO and similarly prepared. The lowest energy docking pose is visualized, was representative of all output poses, and various scoring metrics are reported. Interaction fingerprints were generated for residues exhibiting overall interaction with any of the output poses. Interaction comparisons were calculated as the fraction of total poses interacting with a residue, relative to the WT:2t-hexadecenal interaction, and color-coded for visualization in PyMOL (Schrödinger, LLC). Interaction diagrams were generated and exported from MAESTRO (Schrödinger, LLC) and then redrawn in Inkscape (www.inkscape.org) to aid in visualization.

### SDS-PAGE and western blotting

*Bax⁻/⁻Bak⁻/⁻* MEFs, or MEFs reconstituted to express WT or P168G BAX, were harvested with 0.25% Trypsin-EDTA and pelleted at $800 \times g$ for 5 min. Cells were lysed with 1× RIPA lysis buffer supplemented with protease inhibitors (HALT Tablet, Cat. No. 87786) and phosphatase inhibitors (Cat. No. k1015b, APExBIO) on ice for 20 min, pelleted at $21,000 \times g$ for 10 min at 4 °C, and supernatant was collected for protein quantification by BCA assay (Cat. No. 23225, Thermo Fisher Scientific, Waltham, MA, USA) according to the manufacturer's protocol. Sample concentrations were equilibrated in lysis buffer, combined with 4× Laemmli sample loading buffer supplemented with DTT, subjected to SDS-PAGE in a 12.5% polyacrylamide gel, and transferred to nitrocellulose by standard western blot conditions. Membranes were blocked in 5% milk in 1× TBS buffer supplemented with 0.1% Tween-20, incubated with primary antibodies (1:500-1000 in blocking buffer; incubated for 16–18 h at 4 °C) and secondary antibodies (1:4000 in blocking buffer; incubated for 1 h at 25 °C), followed by standard enhanced chemiluminescence detection (Cat. No. WBLUF0100, Sigma-Aldrich, St. Louis, MO, USA). Antibodies: BAX, 1:500 dilution (Clone 2D2, Cat. No. sc-20067, Santa Cruz Biotechnology, Dallas, TX, USA); GAPDH, 1:1000 dilution (Clone 1E6D9, Cat. No. 60004, Proteintech, Rosemont, IL, USA); m-IgGk BP-HRP secondary antibody (Cat. No. sc-516102, Santa Cruz Biotechnology, Dallas, TX, USA).

### Reporting summary

Further information on research design is available in the Nature Portfolio Reporting Summary linked to this article.

## Data availability

The data supporting the findings of this study are included in the Source Data file. Additional data are also included within the Supplementary Information files. Materials generated as part of this study are available from the corresponding author upon request. Structures corresponding to PDB 1F16, 2K7W, and 5W60 were analyzed within this study; structures corresponding to PDB 4BD2, 4BD6, 4BDU, 4ZIE, 6L8V, 8SPZ, and 9IXU were included as visual aids but were not used for investigation. Mass spectrometry data can be accessed using the identifier MSV000100737. Previously published BMRB codes utilized: BMRB 4632. Source data are provided with this paper.

## Code availability

No code was generated for this study.

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

## Acknowledgements

This work was supported by NIH grants F31-AA024681 (J.D.G.), R01-GM083159 (R.W.K.), R01-CA237264 (J.E.C.), R01-CA267696 (J.E.C.), and R01-CA271346 (J.E.C.); a Collaborative Pilot Award from the Melanoma Research Alliance (J.E.C.); a Department of Defense—Congressionally Directed Medical Research Programs—Melanoma Research Program: Mid-Career Accelerator Award (ME210246; J.E.C.); an award from the National Science Foundation (2217138); a Translational Award Program from the V Foundation (T2023-010); and the Mount Sinai Tisch Cancer Center (MSTCC) Support Grant (P30-CA196521). The authors would like to acknowledge the MSTCC Shared Resources and the Department of Oncological Sciences for research support, and thank Drs. Douglas Green (St. Jude Children's Research Hospital) and Evripidis Gavathiotis (Albert Einstein College of Medicine) for generously providing $Bid^{-/-}Bim^{-/-}$ and BAX-reconstituted $Bax^{-/-}Bak^{-/-}$ cell lines, respectively.

## Author contributions

Conceptualization: J.D.G. and J.E.C.; methodology: J.D.G., Y.C., M.P.A.L.-V., A.V.F., and J.N.M.; validation: J.D.G., Y.C., M.A.N., J.N.M., and T.M.S.; investigation: J.D.G., Y.C., M.P.A.L.-V., A.V.F., M.A.N., M.K., S.G.B., J.N.M., T.M.S., and N.D.P.; resources: Y.C., M.P.A.L.-V., and M.A.N.; writing: J.D.G. and J.E.C.; visualization: J.D.G.; supervision: Y.S., J.J., R.W.K., and J.E.C.; funding acquisition: J.D.G., J.J., R.W.K., and J.E.C.

## Competing interests

The authors declare no competing interests.
