## [Transparent Peer Review File · Nature Communications]

A gated hydrophobic funnel within BAX binds bioactive lipids to potentiate pro-apoptotic function

Corresponding Author: Dr Jerry Chipuk

Version 0:

Reviewer comments:

Reviewer #1

(Remarks to the Author)

In this paper, the authors aim to characterize the mechanism of activation of the BAX protein by 2t-hexadecenal. The authors discovered a binding pocket, which they termed the "BAX actuating funnel", where they propose lipid molecules binds, which leads to conformational change, a key step in BAX activation.

The authors used several types of biophysical technique to characterize BAX-2t-hexadecenal interaction, such a microscale thermophoresis (MST), nuclear magnetic resonance spectroscopy, mass spectrometry, and fluorescence polarization method (referred here as FLAMBE). The authors also made and purified several mutants of BAX and present some mammalian cell culture work.

Overall, the paper address one key question: how is BAX activated, and how is it becoming anchored to the membrane in the mitochondria. Previous studies have shown that a cysteine present in BAX is being covalently modified by the lipid, which lead to conformational change and membrane anchoring, and eventual MOMP in the mitochondria. This is paper challenges this concept and propose an alternative mechanism of BAX activation anchorage to the membrane, through non-covalent interaction of lipids in the "BAF" hydrophobic pocket.

The article is well written, the experiments sound, and the figures are very nicely presented. The authors also present a good discussion presenting their results in the context of previously contradictory results. Finally, while this work is perhaps a little controversial, I believe it is worth publishing and propose an alternative avenue of BAX activation.

Comments:

- Fig. 1B. Why is the control DMSO (for 40 uM) in the SPARKL analysis show such strong signal, more than the ABT 20 uM? This seems concerning.
- Line 111. "[we] observed a loss of apoptosis at lower concentrations of ectopic hexadecenal co-treated with ABT-737, suggesting that BAX activation was part of the underlying mechanism (Figure 1B). Finally, we replicated this experiment in Bax^{-/-}Bak^{-/-} DKO MEFs and observed no cell death (Figure 1C). Collectively, these data support the conclusion that hexadecenal induces apoptotic cell death by acting on effector BCL-2 family proteins."
 While there is nothing wrong with these results, I am puzzled at the conclusion. My conclusion is that ABT-737 obviously doesn't kill Bax^{-/-}Bak^{-/-} DKO.
- Line 131, Fig. 1F. "Furthermore, the synergy with hexadecenal was observed with mildly activating concentrations of BIM-BH3 as well (Figures 1F, S1C)."
 Could it simply be additive? In order to claim synergy, I think the authors need to present the synergistic calculation...
- Line 152. "We detected peptide fragments covering both cysteines (C62, C126) and observed no alkylation by hexadecenal (m+238.229 Da);"
 Is it possible trypsin can't cleave after the lysine at close proximity to the modification?
- The authors report chemical shift changes – but what about comparing line broadening between the mutants? At the residue level, or simply using 1D HN relaxation envelop? There seem to be a loss in chemical shift dispersion (Supp. Fig. 4A)
- Line 330 "I don't like the term "unbiased docking assessments". Nothing is unbiased.
- Line 399. Typo. "Directly comparison"
- Line 475. "We also observed that short-chain alkenals did not activate BAX, which aligns with a prior study that also

demonstrated increased reactivity with short-chain alkenals.”
 Increased reactivity doesn't lead to increase BAX activation?

Reviewer #2

(Remarks to the Author)

The study by Gelles et al. presents a multidisciplinary investigation into how the lipid metabolite 2t-hexadecenal enhances BAX activation, a central event in mitochondrial apoptosis. The authors combine NMR, computational modeling, biochemical and cellular assays, and mutagenesis to provide mechanistic insight into this interaction. They show that 2t-hexadecenal binds a novel binding region on BAX, a hydrophobic cavity formed by helices $\alpha 5$, $\alpha 6$, and gated by $\alpha 8$, which they term the BAX Actuating Funnel (BAF). Their data suggest that 2t-hexadecenal binds non-covalently at this site, priming BAX for activation, particularly in the presence of the BH3-only activator BIM. Further, they show that $\alpha 8$ helix mobility and alkenal chain length are critical determinants of lipid-mediated BAX activation. The study is thorough and well-controlled, offering a rational model for how 2t-hexadecenal lipid metabolite can modulate BAX conformational dynamics to facilitate BH3-mediated BAX activation leading to mitochondrial outer membrane permeabilization (MOMP). Overall, the manuscript offers a novel conceptual model for lipid-mediated BAX activation and a strong case for the biological relevance of the newly identified BAF site, that may be exploited for therapeutic modulation using small molecules. However, I have listed several issues below that should be addressed to strengthen the manuscript and validate the conclusions before publication.

1. The authors report a dissociation constant (K_d) of 0.182 μM for 2t-hexadecenal binding to BAX (MST assay), yet concentrations $\gg 6.5\text{--}10\ \mu\text{M}$ are used in FLAMBE, liposomal, and cellular assays, including in the presence of BIM BH3. The rationale for this discrepancy should be clearly explained. Have the authors used concentrations near the K_d concentration? Why functional effects require supersaturating conditions?
2. The authors report a non-covalent interaction between 2t-hexadecenal and BAX. However, prior studies from two independent groups have provided convincing evidence for covalent modification of BAX by 2t-hexadecenal. Did the authors use comparable experimental conditions and mass spectrometry techniques to exclude the possibility of covalent binding in their assays?
3. Given the known reactivity of alkenal metabolites, can the authors confirm the purity and structural integrity of the 2t-hexadecenal used in their experiments? Can they provide batch consistency and analytical validation (e.g., NMR, MS) to help rule out variability of the 2t-hexadecenal as a factor in observed differences compared to previous studies.
4. The manuscript briefly suggests that NP-40 (a non-ionic detergent) may explain some discrepancies with earlier findings. The authors should elaborate on this point. How might NP-40 influence lipid-protein interactions in lysates or buffers in a way that affects covalent modification?
5. The authors note that the BAF cavity is present in previously reported structures of BAX bound to BID BH3 or BAX BH3 domains and the domain-swapped dimers of BAX. The authors should include a comparative structural figure highlighting the BAF site across these BAX structures for additional clarity and support of this conclusion.
6. Figures lack detailed information about the number of replicates and statistical analyses performed. The authors should include this information to figure legends wherever applicable.

Reviewer #3

(Remarks to the Author)

Gelles et al. investigate the mechanism by which the bioactive sphingolipid metabolite trans-2-hexadecenal (2t-16) promotes BAX activation, a process relevant to mitochondrial outer membrane permeabilisation and apoptotic cell death. Employing biochemical, structural, and computational approaches, the authors identify a hydrophobic region of BAX as a putative binding site for non-covalent interaction with 2t-16. The authors propose that this interaction, particularly in the presence of the BH3-only activator BIM, potentiates early BAX conformational changes and promotes membrane permeabilisation, without invoking the covalent mechanism elucidated in prior studies.

Whilst the study employs several creative indirect methods and identifies a potential lipid interaction site, the central claim that 2t-16 operates principally through a non-covalent mechanism rests on incomplete and, at times, internally inconsistent evidence. Given the well-established reactivity of α,β -unsaturated aldehydes and the membrane-embedded nature of lipid regulation, much of the data—derived largely from high-concentration solution-phase lipid treatments and rigid-body docking to an inactive BAX structure—appears inconsistent with the proposed model. The findings may ultimately align better with a dual mechanism involving both covalent and non-covalent features, as is commonly observed in physiological and pharmacological covalent protein regulation.

Moreover, the study is marked by a considerable degree of speculation, with data interpreted selectively to support the authors' model, whilst alternative explanations receive insufficient consideration.

Major Concerns:

1. Departure from Physiological Context:

Prior work established that 2t-16, generated by ER lipid metabolism, is shuttled to mitochondria to regulate BAX. Building on these findings, subsequent studies documented a covalent mechanism of regulation for this lipid electrophile. In contrast, the present study departs from the physiologically relevant membrane context by applying high-dose 2t-16 in solution to cells and liposomes (a “direct treatment model” that is insufficiently justified and moves further away from physiological relevance). Further, the membrane environment may facilitate BAX conformations amenable to covalent derivatization, which would not be readily recapitulated by a solution system. The lipid, solubilised using DMSO and prewarming techniques, likely exists as heterogeneous micelles or aggregates, raising concerns regarding its bioavailability and mechanism of intracellular delivery. Intracellular entry beyond plasma membrane incorporation appears doubtful, and any 2t-16 reaching the cytoplasm would likely be rapidly reduced or react with multiple off-target proteins. The observed cellular effects might therefore reflect a non-specific stress response culminating in apoptosis rather than direct BAX regulation. Furthermore, the use of serum starvation—a known pro-apoptotic stimulus—as the delivery context for 2t16 further complicates the interpretation. Similar concerns extend to the *in vitro* data, where the observed effects could more plausibly derive from non-specific detergent-like action, as evidenced by the lack of distinction between wild-type and cysteine-mutant BAX.

2. Limitations of the Mass Spectrometry Analysis:

The MS experiments relied on to discount covalent modification are not sensitive for detecting lipidated BAX due to technical limitations. Lipidated peptides are very hydrophobic and often poorly recovered in MS workflows, and activated BAX oligomers are typically resistant to proteolytic digestion and mass spectrometric detection. Thus, the absence of detectable lipid-modified peptides does not exclude the presence of covalent modifications, particularly given the methodological constraints and the limited experimental detail (e.g., lipid-to-protein ratios, incubation times) provided.

3. Inappropriate Use of Rigid-Body Docking:

The docking studies are performed using the structure of inactive BAX, whereas the proposed binding site is thought to emerge following BIM-induced conformational activation. As such, the docking is based on an inappropriate structural template, and the conclusions drawn are therefore questionable. For a protein as conformationally dynamic as BAX, molecular dynamics simulations would offer a more appropriate approach than rigid-body docking.

4. Technical Concerns with the NMR Data and Its Interpretation Based on the Structure of Inactive BAX:

The interpretation of the NMR data raises additional concerns. First, the data are analyzed in the context of the inactive BAX structure, whereas the proposed model places non-covalent interaction of 2t-16 with BAX downstream of BIM activation. This is problematic because BIM activation is known to induce conformational changes that could alter the proposed binding site, currently mapped to and evaluated based on the inactive state. Second, the timing parameters of the NMR experiments—specifically, the duration of protein-ligand incubation and the time point of NMR data acquisition—are not disclosed, limiting the ability to assess the relevance of the observations to the proposed model. Since BAX activation by 2t-16 promotes dimer (as suggested by the SEC data) and ultimately oligomer formation, such species that could be induced by covalent modification would not be visible by NMR due to signal attenuation, biasing the solution NMR data toward evaluable monomeric species only for the undisclosed conditions that were applied.

5. Control Compound and Binding Orientation Issues:

Based on the authors' non-covalent ligand binding model, there is little mechanistic rationale for why the reduced form (16CHO) would fail to bind and activate BAX, whereas the α,β -unsaturated aldehyde (2t-16) would. Both compounds are featureless hydrophobic chains differing only at the terminal functional group. If the entirety of the binding mechanism was dependent on the presence of the double bond (however unlikely), a key control beyond 16CHO would be the alcohol derivative (trans-2-hexadecen-1-ol), which maintains the double bond but removes the electrophilic aldehyde functionality. Furthermore, the docking poses suggest an unnatural orientation, with the lipid tail (ordinarily embedded within the membrane) proposed to insert into the hydrophobic groove—an orientation inconsistent with membrane biophysics. Instead, a model involving covalent reaction between a cysteine on the BAX surface and a membrane-exposed and reactive lipid head group appears more plausible, as previously detected and validated in the native membrane context.

6. Interpretation of Mutagenesis Data:

The mutagenesis experiments do not conclusively validate the proposed lipid-binding mechanism. Mutations within the putative binding site could alter BAX activation through a range of alternative structural or functional mechanisms. For example, the P186G mutant is known to exhibit reduced activity, but attributing this to impaired lipid binding is speculative without further rigorous characterisation.

Summary:

This manuscript offers interesting insights into potential non-covalent interactions between 2t-16 and BAX that may influence BAX function. However, the evidence provided does not sufficiently support the conclusion that 2t-16 acts predominantly via a non-covalent mechanism. The data appear more consistent with a complementary non-covalent contribution within an overarching covalent regulatory process, particularly in light of established principles of lipid electrophile biology (including diminished activity of the reduced analog) and lipid orientation within membranes.

The study would benefit from additional experimentation in membrane systems, a broader exploration of covalent interactions under physiologically relevant conditions, cellular proof of direct non-covalent interaction with BAX, and a more cautious framing of its conclusions.

Finally, the characterisation of prior work as limited or flawed, whilst employing even more supra-physiological and indirect methodologies in solution, appears unnecessarily dismissive and detracts from the overall scientific narrative. Further,

indicating that “structural and molecular mechanisms remain largely unknown” does not accurately reflect the prior literature on 2t-16 and BAX. A more balanced and integrative perspective—that covalent and non-covalent features likely cooperate in BAX regulation by 2t-16—would better reflect the available data and provide a stronger foundation for future studies.

Version 1:

Reviewer comments:

Reviewer #1

(Remarks to the Author)

The authors have addressed my comments adequately.

Reviewer #2

(Remarks to the Author)

The authors have satisfactorily addressed my comments with new data, clarifications, and revisions of the text. They investigated two BAX mutants with reduced oligomerization capacity for evidence of covalent adducts formed by 2t-hexadecenal binding and evaluated the BAX-2S mutant (C62S/C126S), which lacks reactive cysteines and therefore cannot form covalent adducts. These new experiments and data strengthen the authors' conclusions and underscore a role for 2t-hexadecenal in mediating early BAX activation in the cytosol. This is also consistent with the structures of BAX conformations available. While prior studies demonstrated covalent lipid adduction to BAX, the present work offers an alternative and potentially complementary mechanism for 2t-hexadecenal. The authors have clearly articulated the study's conclusions and limitations in the revised manuscript and reviewer's responses. Overall, this is a coherent and well-executed study and I recommend publication.

Reviewer #3

(Remarks to the Author)

I appreciate the revision efforts and the clarifying remarks with respect to rationale and methodology. The work introduces interesting tools and concepts regarding early BAX activation, but several interpretive and methodological aspects remain overstated and highly speculative relative to the data.

1. Positioning of prior work. It would be appropriate to acknowledge up front that covalent modification of BAX by 2t-hexadecenal (2t16) at Cys126 has already been demonstrated in membrane and cellular contexts with functional consequences for MOMP (as opposed to indicating in the abstract and introduction that the molecular mechanism involving 2t16 is “largely unknown” and “remains an open question”). Introducing and acknowledging prior work does not undermine the current work. In accordance with this handling, the discussion concentrates chiefly on the in vitro concentrations used in earlier covalent studies whilst overlooking the accompanying physiological findings that validated these mechanisms in native mitochondria and intact cells in a cysteine-specific manner. The selective emphasis on mM dosing in prior in vitro studies gives an incomplete picture and may inadvertently mislead readers regarding the existing mechanistic evidence and sets a tone that authors are trying hard to “sell” their current work as a controversial alternative. It would be more objective and balanced to frame the present work as exploring non-covalent contributions that complement covalent mechanisms already confirmed in physiologic settings.

2. Plausibility of cytosolic delivery. The proposal that exogenously applied 2t16 freely enters the cytosol and engages BAX to account for the observed results remains speculative and unlikely – and the authors provide no evidence that lipid-BAX binding in the cytosol is actually occurring (nor that 2t16 even reached the cytosol in reacted or unreacted form). It is so well known that α,β -unsaturated aldehydes are highly reactive electrophiles which, if present in the media and if even able to dislodge from the cell membrane to access the cytosol, would be expected to form adducts with numerous nucleophilic targets, including glutathione and abundant soluble proteins (the authors refer to hundreds of potential reactive targets). In this light, a BAX-related effect to explain the Figure 1A-B data appears so implausible as to undermine the subsequent structure-function work. Starting the manuscript on such thin ice (as in, this is our interpretation yet here are the multitude of caveats that more likely make it not a reliable interpretation) is undermining. The early data is among the weakest data (lack of actual dose-responses, many alternative interpretations, etc.).

3. Clarification regarding lipid orientation and membrane context. In the response to point 5, the authors appear to have misunderstood an important aspect of the critique. The concern was not about Schiff-base formation with an aldehyde given the Michael addition mechanism, but instead about lipid orientation: the suggestion that the lipid tail of 2t16 inserts into the hydrophobic cleft of BAX (the hydrophobic BAF site not the BC groove) is difficult to reconcile with the disposition of endogenous ER-to-mitochondrial transfer of these lipids, whose hydrophobic tails are embedded within or associated with membranes. A soluble, tail-inserted lipid electrophile acting on cytosolic BAX is conceptually inconsistent with known membrane biology. I appreciate that the authors believe that perhaps 2t16 can find its way to the solution phase, but they never test or show it by fractionation MS or otherwise. In addition, there is no data presented from cells or cell lysates that show this lipid is associated with soluble BAX. Such results are so fundamental to the study – so proceeding without them makes this a really hard sell.

4. Non-covalent versus covalent framing. The failure of non-reactive analogues (saturated or hydrazide-modified aldehydes) to activate BAX supports the relevance of the α,β -unsaturated moiety to a covalent mechanism, not a non-covalent binding mechanism. The reader will know this, so to interpret this result otherwise is inconsistent with biochemical principles. Along these lines, the cysteine-mutant data do not disprove a covalent mechanism, they only allow for study of an isolated non-covalent contribution. Therefore, the result section subtitles and interpretations in the abstract and conclusion should emphasize the study of a non-covalent contribution rather than position the results as proving a non-covalent only mechanism, which the data simply do not support (and more often suggest the opposite). The NMR data show that the non-reactive 16CHO compound causes NMR CSPs, which would be expected as the non-covalent component of a covalent process, because the specificity of physiologic covalent reactivity and signaling relies on non-covalent interaction to template and reinforce selectivity of the covalent component of the interaction.

5. Mass-spectrometry limitations and docking language. The authors' negative MS data do not exclude covalent modification (particularly since two independent groups demonstrated otherwise), and the docking studies would be better presented as hypothesis-generating rather than structural evidence. Since MD simulations are not performed, "docking simulations" terminology should be changed to "rigid-body docking" or "static docking calculations".

6. Clarifying the dosing context. The authors emphasize that they are using micromolar rather than millimolar dosing for in vitro studies, but in the response the authors also indicate that in one set of experiments the molar excess is up to 40,000-fold compared to BAX, which undermines the argument that this study is more conservative or physiologic with respect to dosing levels compared to prior ones. The in vitro setting is just that – it is non-physiologic by definition and is used to hypothesis generate for ultimate testing and validation in cells and tissues.

7. Lingering structural methodology concerns. The manuscript remains heavily reliant on static docking calculations, which constitute a minimal and low-rigor approach given the central role of computation in their conclusions. While the authors argue that molecular dynamics (MD) simulations are beyond scope, this is unpersuasive; MD is precisely the method needed to capture conformational variability, address structural perturbations (e.g., $\alpha 8$ deletion), and evaluate stability of binding modes. The current strategy does not adequately represent the relevant ensembles or intermediates, and the terminology conflating docking with simulation is misleading. At minimum, the authors should employ explicit solvent MD sampling of existing structures, relax mutant models, and dock to representative ensembles. Without this, the work does not meet basic standards for a high-impact publication.

To summarize, the authors work so hard at justifying the new model that they undermine their credibility by making comparisons and justifications that aren't objectively effective or necessary based on the data provided. Making the suggested textual refinements, with an overall more objective approach to data interpretation would allow the manuscript to stand as a more constructive extension of previous models rather than a competing reinterpretation, whilst fully crediting established physiological data and situating the proposed mechanism within the known biochemistry of lipid electrophiles. It may be "OK" to directly say, this is what we speculate based on our results despite the caveats, some of our own data (and prior data) are inconsistent with our new model, and even quite inconsistent with what the reader might expect from lipid electrophiles and how they are used in the study. At the end of the day, a more circumspect handling seems not only justified, but could ultimately come across more genuine and compelling.

Version 2:

Reviewer comments:

Reviewer #3

(Remarks to the Author)

The authors' responses address several points, but many rely more on rhetorical framing than on directly engaging the underlying scientific issues. To be constructive and avoid unnecessary back-and-forth, I offer a few specific examples where clarification or correction would strengthen the manuscript. My intention is sincerely to help refine the discourse to ensure that the authors' interpretations align with biochemical principles and the data presented.

Line 4, Title: Consider whether "lipids" should be singular, as the study focuses on one lipid species.

Lines 34–35, Abstract: The phrase "mechanisms remain underexplored" is not strictly accurate, as prior work included detailed structure–function studies. The sentence reads clearly without this clause.

Line 84, Introduction: The phrase "requirement for 2t-hexadecenal in BAX pore formation" mischaracterizes earlier covalent studies, which examined BAX monomer activation rather than strictly pore formation.

Lines 106–107, Results: The external-treatment approach remains difficult to interpret given the compound's high intrinsic reactivity. It is unknown whether sufficient cytosolic levels can be reached to achieve a non-covalent interaction with BAX, as opposed to non-specific induction of stress. The term "validated ectopic hexadecenal" is not supported experimentally.

Lines 130–131, Results: The statement that the authors use LUVs "without altering the biochemistry of the LUV membrane" doesn't really apply; LUVs are synthetic systems defined entirely by their constructed compositions.

Lines 142–144, Results: This is a key area needing clarification. Both control compounds—hexadecanal (double bond reduced) and the hydrazone (aldehyde eliminated)—lack electrophilicity because the distinctly reactive α,β -unsaturated aldehyde moiety is disrupted in each case. Although the hydrazone species still contains a double bond, it is not “reactive” (electrophilic) without conjugation to the aldehyde. Readers will recognize that both controls remove Michael acceptor capacity. Thus, these negative results are much more consistent with loss of covalent reactivity than with selective loss of a non-covalent interaction that happens to require both of these molecular features. This is important because both compounds are treated as specificity controls across multiple assays (thermophoresis [lines 158-159], FLAMBE [lines 216-217], SEC [lines 247-248], results paragraph introduction [lines 335-336], etc.), so accurate interpretation here is especially important.

Line 196, Results: The distinction made regarding “direct treatment” is not meaningful, as prior studies also used direct treatment models.

Lines 254–255, Results: A concluding statement about lipid-BAX association at a specific stage of the activation pathway is made without evidence of the interaction in a cellular or cell lysate context.

Lines 354–355, 402–404, 433–435, Results: The authors appropriately acknowledge the limitations of rigid docking, including artifact risk and inability to determine ligand orientation. However, given these limitations, it is premature to draw a declarative conclusion that the aldehyde “resides in the depth of the BAF” without more informative structural or computational approaches. Incorporating even basic molecular dynamics simulations is widely considered an industry standard for such claims in a high-impact journal.

Lines 500–501 and 522–523, Discussion: The suggestion that prior results may reflect “the use of millimolar concentrations of 2t-hexadecenal”, coupled with the assertion that BAX is “unlikely to experience millimolar concentrations...during activation in the cytosol,” is misleading. Earlier studies also employed micromolar concentrations as well as endogenous, physiologic levels present in mitochondrial membranes. It is worth noting that covalent mechanisms generally require far lower effective concentrations than non-covalent interactions to exert biological effects in a cellular context, so the focus on millimolar levels does not meaningfully address the underlying issue.

RESPONSE TO REVIEWERS

Reviewer #1:

In this paper, the authors aim to characterize the mechanism of activation of the BAX protein by 2t-hexadecenal. The authors discovered a binding pocket, which they termed the "BAX actuating funnel", where they propose lipid molecules binds, which leads to conformational change, a key step in BAX activation.

The authors used several types of biophysical technique to characterize BAX-2t-hexadecenal interaction, such a microscale thermophoresis (MST), nuclear magnetic resonance spectroscopy, mass spectrometry, and fluorescence polarization method (referred here as FLAMBE). The authors also made and purified several mutants of BAX and present some mammalian cell culture work.

Overall, the paper address one key question: how is BAX activated, and how is it becoming anchored to the membrane in the mitochondria. Previous studies have shown that a cysteine present in BAX is being covalently modified by the lipid, which lead to conformational change and membrane anchoring, and eventual MOMP in the mitochondria. This is paper challenges this concept and propose an alternative mechanism of BAX activation anchorage to the membrane, through non-covalent interaction of lipids in the "BAF" hydrophobic pocket.

The article is well written, the experiments sound, and the figures are very nicely presented. The authors also present a good discussion presenting their results in the context of previously contradictory results. Finally, while this work is perhaps a little controversial, I believe it is worth publishing and propose an alternative avenue of BAX activation.

We greatly appreciate this Reviewer's support of our work and conclusions, even as we provide a divergent model of 2t16-mediated BAX activation than prior publications. While we do strive to make a compelling argument for our findings and model, we do not intend to discount the conclusions of prior publications, instead we propose explanations for how (and when) each model may be relevant to BAX activation mechanisms. We are encouraged that this Reviewer recognizes our efforts in this regard. Responses to the questions and comments are below; we also extensively edited the manuscript to reduce ambiguity and increase clarity as suggested. We appreciate the Reviewer for highlighting these points in the text and we believe that the revised manuscript is much improved.

Comments:

- Fig. 1B. Why is the control DMSO (for 40 uM) in the SPARKL analysis show such strong signal, more than the ABT 20 uM? This seems concerning.

*The dark purple line in Figure 1B represents 40 μM 2t16 without cotreatment of ABT-737. As the reviewer notes, 40 μM 2t16 is sufficient to induce some apoptosis as a solo treatment (and is heightened in the presence of ABT-737 – data line with circles), while 20 μM 2t16+ABT-737 does not induce death in the *Bim*^{-/-}*Bid*^{-/-} MEFs. When compared with WT MEFs (Figure 1A), we interpret this decreased sensitivity (at 20 μM 2t16) to the lack of direct activators and therefore an inability of BAX to be activated, even if the anti-apoptotic reservoir is neutralized by ABT-737. The 40 μM 2t16 treatment is pro-apoptotic in both WT and *Bim*^{-/-}*Bid*^{-/-} MEFs (Figures 1A–B) and we interpret this as a condition that*

can induce BAX activation directly. As we describe later in Figure 1 using LUV permeabilization studies, while high dose 2t16 can induce BAX activation directly, the major effect of 2t16 is observed in contexts where BAX is activated by BIM. Therefore, we believe the cell death studies align with the subsequent permeabilization studies and that the interpretations throughout Figure 1 are consistent.

In case this comment arose from misunderstanding our Figure labels, we modified the labels and legends for Figures 1A–C to increase clarity and adjusted the text in the results section (lines 101–120).

• Line 111. “[we] observed a loss of apoptosis at lower concentrations of ectopic hexadecenal co-treated with ABT-737, suggesting that BAX activation was part of the underlying mechanism (Figure 1B). Finally, we replicated this experiment in Bax^{-/-}Bak^{-/-} DKO MEFs and observed no cell death (Figure 1C). Collectively, these data support the conclusion that hexadecenal induces apoptotic cell death by acting on effector BCL-2 family proteins.”

 While there is nothing wrong with these results, I am puzzled at the conclusion. My conclusion is that ABT-737 obviously doesn't kill Bax^{-/-}Bak^{-/-} DKO.

The key comparison is that 40 μM 2t16 (highest dose), which demonstrated cytotoxicity in MEFs expressing BAX (Figures 1A–B), did not induce cell death in the Bax^{-/-}Bak^{-/-} MEFs. Since there is no BAX to be activated by 2t16, the addition of ABT-737 did not reveal a death response as it had in BAX-expressing cells. We adjusted the text to better articulate this point and draw the reader to the relevant comparisons (lines 101–120).

• Line 131, Fig. 1F. “Furthermore, the synergy with hexadecenal was observed with mildly activating concentrations of BIM-BH3 as well (Figures 1F, S1C).”

 Could it simply be additive? In order to claim synergy, I think the authors need to present the synergistic calculation...

To substantiate the use of "synergy" within this work, we confirmed a synergistic effect using the Bliss model and added new data panels reporting the Bliss score for our endpoint LUV permeabilization data throughout the manuscript (Figures 1G, 1I, 1K, 2F, 2H, 6H, and 7G).

Additionally, since the original submission, we identified a mathematical model to perform non-linear regression data fitting of the LUV permeabilization kinetics, which permits us to extract certain parameters for comparative analysis. In the revised manuscript, we performed data fitting of our LUV experiments (Figures S1D, S2G) and report the maximal rate for comparisons across several concentrations of BIM-BH3 and 2t16 (Figures 1F, 2H). Details of the synergy determination and data fitting have been added to the Methods section (lines 1027–1033).

We thank the reviewer for this suggestion and believe these additional metrics further support our interpretations and strengthen the revised manuscript.

- Line 152. “We detected peptide fragments covering both cysteines (C62, C126) and observed no alkylation by hexadecenal (m+238.229 Da);”

 Is it possible trypsin can't cleave after the lysine at close proximity to the modification?

*While we cannot exclude this possibility, we note that Jarugumilli et al., 2018 also utilized trypsin digestion within their work. Additionally, if the lack of observed m+238 cysteine residues were due solely to a limitation of the technique, the prevalence of iodoacetamide-modified fragments (m+57) would have been noticeably different in the DMSO vs 2t16 samples, which was not observed (these data are now **Figure S2C**).*

*In the original submission, we utilized "bottom-up" HCD LC-MS analysis; however, multiple reviewers have raised questions regarding the potential limitations of the digestion and detection of "lipidated" BAX fragments. Therefore, we conducted additional intact mass spectrometry experiments to assess the status of BAX modifications. Due to limitations of sample preparation (e.g., protein concentrations, sample injection, etc.), experiments utilizing wildtype BAX protein did not provide consistent and accurate data. Instead, we utilized oligomerization-deficient BAX mutants (BAX^{G108V}, BAX^{R109D}) to confirm no modification in response to 2t16; as a control, we did observe cysteine modification in response to iodoacetamide treatment (**Figures 2B–C; lines 167–174**).*

*While we believe the data from both mass spectrometry approaches are compelling, we agree that they are not conclusive. As such, we corroborated our conclusions by utilizing a cysteine-deficient BAX mutant (BAX^{2S}) in mechanistic and functional assays throughout the study, thereby demonstrating that synergistic activation of BAX by 2t16 is independent of cysteines (**Figures 2, 3, 5, S2, S3, and S7**). Additionally, we added experiments demonstrating that wildtype BAX modified with iodoacetamide remains sensitive to 2t16 (**Figures 2D–F; lines 175–179**). Collectively, these diverse approaches all corroborate our conclusion that, at this concentration range, 2t16 results in BAX activation through a non-covalent interaction.*

- The authors report chemical shift changes – but what about comparing line broadening between the mutants? At the residue level, or simply using 1D HN relaxation envelop? There seem to be a loss in chemical shift dispersion (Supp. Fig. 4A)

*We did observe that several residues peripheral to the BAF (α 4–5 loop, α 5, α 6, α 7, α 8) exhibited altered peak morphology, such as splitting or merging. We have included this analysis in the revised manuscript (**Figure 4A, right panel; lines 267–270**).*

*In addition to reporting the chemical shift perturbations (CSPs) for BAX+2t16 (which is calculated from a weighted average of both 1H and 15N shifts) we included an analysis depicting the shifts for each individual dimension (**Figure S4C**). For the most part, residues exhibiting a significant shift in either the 1H or 15N dimension were also determined to be significant in the CSP calculation, though this approach did reveal that our analysis may preference towards N shifts.*

The suggestion to compare NMR results across the BAX mutants is interesting. At present, we have not conducted NMR studies with our BAF mutants. The BAX mutants may exhibit different spectra compared to wildtype and therefore would require substantial effort to

reassign the peaks (potentially requiring additional dimensions or analyses). We agree that this could be informative and hope to explore this in future studies.

- Line 330. I don't like the term "unbiased docking assessments". Nothing is unbiased.

We have replaced "unbiased" with "unconstrained" throughout the main text and figure legends, which we agree is a more accurate description of the approach.

- Line 399. Typo. "Directly comparison"

Thank you for pointing out this typo.

- Line 475. "We also observed that short-chain alkenals did not activate BAX, which aligns with a prior study that also demonstrated increased reactivity with short-chain alkenals."

 Increased reactivity doesn't lead to increase BAX activation?

*We apologize for not explaining this point sufficiently. Cohen et al., 2020 demonstrated that short-chain alkenals were more proficient at modifying BAX compared to long-chain alkenals. Utilizing BAX laddering assays, they demonstrated that high molecular weight species of BAX were formed proportional to alkenal length, with long-chain alkenals exhibiting less monomeric and more oligomeric bands. Our results provide an explanation for this reciprocal relationship between reactivity and BAX activation proportional to alkenal chain length. Our BAX activation assays corroborate that short-chain alkenals fail to activate BAX compared to 2t16, and our structural and in silico approaches suggest that this is due to reduced specificity for the BAF (**Figures 5, S5, S6**). We have modified the text to better articulate how our model fits with this prior study (**lines 375–381**).*

Reviewer #2:

The study by Gelles et al. presents a multidisciplinary investigation into how the lipid metabolite 2t-hexadecenal enhances BAX activation, a central event in mitochondrial apoptosis. The authors combine NMR, computational modeling, biochemical and cellular assays, and mutagenesis to provide mechanistic insight into this interaction. They show that 2t-hexadecenal binds a novel binding region on BAX, a hydrophobic cavity formed by helices $\alpha 5$, $\alpha 6$, and gated by $\alpha 8$, which they term the BAX Actuating Funnel (BAF). Their data suggest that 2t-hexadecenal binds non-covalently at this site, priming BAX for activation, particularly in the presence of the BH3-only activator BIM. Further, they show that $\alpha 8$ helix mobility and alkenal chain length are critical determinants of lipid-mediated BAX activation. The study is thorough and well-controlled, offering a rational model for how 2t-hexadecenal lipid metabolite can modulate BAX conformational dynamics to facilitate BH3-mediated BAX activation leading to mitochondrial outer membrane permeabilization (MOMP).

Overall, the manuscript offers a novel conceptual model for lipid-mediated BAX activation and a strong case for the biological relevance of the newly identified BAF site, that may be exploited for therapeutic modulation using small molecules. However, I have listed several issues below that should be addressed to strengthen the manuscript and validate the conclusions before publication.

We are delighted that this Reviewer supports the nature of our work and its conclusions. As suggested, we incorporated new data, analyses, and discussion points to address all comments and questions. Thank you for the opportunity to improve this study and its communication.

1. The authors report a dissociation constant (K_d) of 0.182 μM for 2t-hexadecenal binding to BAX (MST assay), yet concentrations $\gg 6.5\text{--}10\ \mu\text{M}$ are used in FLAMBE, liposomal, and cellular assays, including in the presence of BIM BH3. The rationale for this discrepancy should be clearly explained. Have the authors used concentrations near the K_d concentration? Why functional effects require supersaturating conditions?

*This is a good point. The MST software reported this value as a K_d , but it appears to be largely derived from the inflection point of the dose curve (these data are now **Figure S2A**). As such, the concentration of the target (BAX) is relevant to this determination. In this assay, we were using 1 nM BAX and so 2t16 was used in 20–40,000 fold excess (to ensure saturation and accurate curve fitting), with the midpoint at ~180 fold excess. In our functional assays (e.g., FLAMBE, LUVs), the concentration of BAX is regularly 100–150 nM and so using 2t16 at 5–50 μM represents approximately 33–500 fold excess, which aligns with MST results. Also of note, the MST experiments were conducted with low-level detergent – 0.5% Tween-20 in the MST buffer to prevent capillary adsorption and 0.002% CHAPS to inhibit BAX activation – and that may also have affected the determination of this value.*

*For these reasons, we avoided putting much emphasis on this value in the text, but we have added a statement to address the above points (**lines 151–155**). Additionally, we will now label the metric as an EC_{50} value instead of a K_d , which is likely more accurate and should avoid over-interpretation of the metric. We believe that addressing this comment has clarified the explanation of these data and thank the reviewer for bringing it to our attention.*

2. The authors report a non-covalent interaction between 2t-hexadecenal and BAX. However, prior studies from two independent groups have provided convincing evidence for covalent modification of BAX by 2t-hexadecenal. Did the authors use comparable experimental conditions and mass spectrometry techniques to exclude the possibility of covalent binding in their assays?

We did our best to properly address this in our Discussion by providing sufficient consideration and contemplation to explain the divergence in phenotypes, but it is delicate balance advocating for our findings without appearing to disparage the prior works. We describe some of the major differences in experimental conditions below, and modified part of our Discussion to capture some of these points (lines 494–513).

The paper that identified modification at C62 (Jarugumilli et al., 2018) utilized an alkyne derivative of 2t16 and subjected their samples to additional chemical agents as part of their click chemistry and pull-down methods. Additionally, their recombinant BAX (Cat #: BAX-6976H, Creative Biomart) is a C-terminal truncated mutant (aa1-171) with two additional His tags, of which the N-terminal tag is 20 residues long. For cell lysates, they subjected their samples to various detergents which may have altered the availability of the cysteine residues (see our response to comment 4).

The paper that identified modification at C126 (Cohen et al., 2020) predominantly used 2t16 and other alkenals in millimolar range (0.5–5 mM), which according to their own data induced the chemical modification. The hydrazide labeling of BAX activated to permeabilize LUVs was a very clever approach, but appears to also label activated BAX in LUVs without 2t16 (albeit to a lesser extent), and there was no control demonstrating that BAX-C62A/C126A could not be labeled by the hydrazide. Additionally, some assays utilized auxiliary chemistries and pull-down methods and their 2t16 was solubilized in ethanol instead of DMSO – we do not know if these differences are consequential to the chemistry they describe.

Ultimately, there came a point at which we had to decide whether to exhaustively attempt to replicate these prior studies and identify conditions to rationalize the divergent results, or to focus on keeping conditions as holistic and relevant to the underlying cell biology, and study aspects of BAX biology not covered in the prior works. As such:

- 1. We avoided using reactive chemistries as part of our detection methods and instead focused on techniques to study BAX activation.*
- 2. We did not use millimolar concentrations of 2t16 because we aimed to study its effect on BAX distinct from covalent modifications and at concentrations that better represented a physiological cellular context.*
- 3. We chose not to incorporate 2t16 directly into liposomes as this had already been shown and may specifically/solely reflect the biology of the oligomeric conformer.*

Instead, we focused on how 2t16 affects monomeric BAX activation prior to membrane associations. To this end:

- 1. We designed experiments to specifically characterize the early steps of the BAX activation continuum, including approaches not utilized in prior works as well as developing new ones (e.g., our FLAMBE assay to specifically monitor "trigger"-induced intramolecular rearrangements).*

2. We extended our structure-function insights of BAX by generating several structural mutants and identifying the determinants of the interaction (such as BAF topology and demonstrating that short-chain aldehydes lack targeting specificity).
3. We explored the hypothesis that the disease-relevant P168 mutation allosterically alters the BAF resulting in the loss-of-function phenotype.
4. And we attempted to place our discoveries within the larger established literature of the BAX structure-function relationship by detailing how our results connect to a collection of discrete observations (our Discussion).

We hope that the take away for the reader is that each work studies distinct biology, and that BAX is potentially regulated both biophysically (interactions with the BAF) and biochemically (cysteine modification) by 2t16 in discrete stages and environments throughout the BAX activation continuum. We modified our Discussion to better highlight this interpretation of both our data and the prior works (**lines 494–533, 602–606**).

3. Given the known reactivity of alkenal metabolites, can the authors confirm the purity and structural integrity of the 2t-hexadecenal used in their experiments? Can they provide batch consistency and analytical validation (e.g., NMR, MS) to help rule out variability of the 2t-hexadecenal as a factor in observed differences compared to previous studies.

Our 2t16 is purchased from Avanti Polar Lipids and each lot comes with a certificate of analysis and is confirmed to be >99% pure. We prepare our 2t16 in single use aliquots to minimize oxidation or degradation. Though to directly address this question, we performed mass spectrometry on 2t16 and indeed observed a peak corresponding to the correct molecular weight; we have included this vital control (**Figure S1A**).

4. The manuscript briefly suggests that NP-40 (a non-ionic detergent) may explain some discrepancies with earlier findings. The authors should elaborate on this point. How might NP-40 influence lipid-protein interactions in lysates or buffers in a way that affects covalent modification?

As it specifically pertains to Jarugumilli et al., 2018, we question whether the inclusion of NP-40 during cell lysis may have affected BAX structure or exposure of the C62 residue. It is well established that common detergents (including NP-40) induce BAX activation and oligomerization when used above their CMC (e.g., Antonsson et al., 2000). Additionally, C62, which sits within the key BH3 residues of BAX $\alpha 2$, is predominantly core-facing and not likely to be solvent exposed (Suzuki et al., 2000; PDB 1F16). It is a known hallmark that these $\alpha 2$ residues get exposed following BAX activation (Gavathiotis et al., 2010) and are modeled to be cytosolic facing in the activated BAX dimer at the membrane (Czabotar et al., 2013; Bleicken et al., 2014; PDB 4BDU – N.B. C62 is replaced with serine in this structure and $\alpha 5$ is extended to predict C126). Therefore, we posit that NP-40 may have facilitated certain steps of the BAX activation continuum and that their 2t16 alkyne was targeting an activated conformer of BAX where C62 is more targetable. This may also explain the divergent result in Cohen et al., 2020, where they concluded C126 was the primary target residue.

We apologize for not detailing this thought process sufficiently and modified the Discussion to include this explanation (**lines 504–509, 523–526**). Additionally, we added structural models of the active BAX dimer to support these discussion points, including the recently published structure of oligomeric BAX (Zhang et al., 2025; PDB 9IXU)

(Figures S10A–C). We agree that these details are vital to properly articulating our conclusions and believe this addition will greatly improve the Discussion.

5. The authors note that the BAF cavity is present in previously reported structures of BAX bound to BID BH3 or BAX BH3 domains and the domain-swapped dimers of BAX. The authors should include a comparative structural Figure highlighting the BAF site across these BAX structures for additional clarity and support of this conclusion.

To address this excellent suggestion, we added several structures representing domain-swapped BAX dimers bound to various BH3 peptides with their BAF cavities visualized (Figure S10D). Of note, we did slightly modify the text to specifically highlight structures that featured progressive C-terminal truncations, which is a subsequent point we discuss; as such, the structure bound to BAX-BH3 is no longer included as one of the examples (lines 572–575)

Similarly, this comment inspired us to include a visualization of the BAF across the ensemble states of the BAX solution structures – namely, monomeric BAX (PDB 1F16) and BAX triggered by BIM (PDB 2K7W) (Figures S4F–G; lines 283–285, 297–300). We agree that these additional visualizations improve the revised manuscript and thank the reviewer for this suggestion.

6. Figures lack detailed information about the number of replicates and statistical analyses performed. The authors should include this information to Figure legends wherever applicable.

We apologize for this oversight and added additional details to our figure legends and methods section where needed.

Reviewer #3:

Gelles et al. investigate the mechanism by which the bioactive sphingolipid metabolite trans-2-hexadecenal (2t16) promotes BAX activation, a process relevant to mitochondrial outer membrane permeabilisation and apoptotic cell death. Employing biochemical, structural, and computational approaches, the authors identify a hydrophobic region of BAX as a putative binding site for non-covalent interaction with 2t16. The authors propose that this interaction, particularly in the presence of the BH3-only activator BIM, potentiates early BAX conformational changes and promotes membrane permeabilisation, without invoking the covalent mechanism elucidated in prior studies.

Whilst the study employs several creative indirect methods and identifies a potential lipid interaction site, the central claim that 2t16 operates principally through a non-covalent mechanism rests on incomplete and, at times, internally inconsistent evidence. Given the well-established reactivity of α,β -unsaturated aldehydes and the membrane-embedded nature of lipid regulation, much of the data—derived largely from high-concentration solution-phase lipid treatments and rigid-body docking to an inactive BAX structure—appears inconsistent with the proposed model. The findings may ultimately align better with a dual mechanism involving both covalent and non-covalent features, as is commonly observed in physiological and pharmacological covalent protein regulation.

Moreover, the study is marked by a considerable degree of speculation, with data interpreted selectively to support the authors' model, whilst alternative explanations receive insufficient consideration.

We respect and appreciate this Reviewer's critical analysis of our work and we believe that their perspective may be shared by a subset of the readership. As such, we wish to thoroughly and satisfactorily address the points made below and so we will separate the comments and provide our responses in more granular detail to specific concerns.

Overall, we believe that several of these comments arose from the Reviewer's interpretation that we contradict and/or dismiss the prior works describing a covalent modification by 2t16. This is undoubtedly a consequence of our attempt to defend our findings and a failure to articulate that we suggest a cohesive model in which BAX is likely being regulated both biophysically (interactions with the BAF) and biochemically (cysteine modification) by 2t16 at discrete stages and environments throughout the BAX activation continuum. As we will specifically remark below, we have substantially altered the text of our Discussion to address this.

Of note, this Reviewer appears to focus heavily on the structure, function, and regulation of oligomeric BAX localized to the mitochondrial outer membrane and several comments suggest less mechanistic attention on BAX biology prior to membrane association. There is ample and growing literature studying monomeric BAX activation and regulation in solution separate from the functionalization phase within a membrane. Here, we specifically focus on the biology and structure of monomeric BAX prior to membrane associations and multimeric structures. We do this in part to distinguish our efforts from prior works while still supporting and leaving space for their conclusions; additionally, it affords focus on complementary yet distinct approaches, questions, and investigations:

1. We designed experiments to specifically characterize the early steps of the BAX activation continuum, including approaches not utilized in prior works as well as developing new ones (e.g., our FLAMBE assay to specifically monitor "trigger"-induced intramolecular rearrangements).
2. We extended our structure-function insights of BAX by generating several structural mutants and identifying the determinants of the interaction (such as BAF topology and demonstrating that short-chain aldehydes lack targeting specificity).
3. We explored the hypothesis that the disease-relevant P168 mutation allosterically alters the BAF resulting in the loss-of-function phenotype.
4. And we attempted to integrate our discoveries within the larger established literature of the BAX structure-function relationship by detailing how our results connect to a collection of discrete observations (our Discussion).

Major Concerns:

1. Departure from Physiological Context:

Prior work established that 2t16, generated by ER lipid metabolism, is shuttled to mitochondria to regulate BAX. Building on these findings, subsequent studies documented a covalent mechanism of regulation for this lipid electrophile. In contrast, the present study departs from the physiologically relevant membrane context by applying high-dose 2t16 in solution to cells and liposomes (a "direct treatment model" that is insufficiently justified and moves further away from physiological relevance).

*There is a bit of a misunderstanding regarding the original work identifying the 2t16 requirement. While Chipuk et al., 2012 does detail the requirement of sphingolipids originating in heterotypic membrane populations, the results actually demonstrate that the catabolism of sphingomyelin into ceramide requires ER-localized sphingomyelinases; the subsequent catabolism to 2t16 is accomplished with enzymes found within the mitochondrial outer membrane. The reader may be inclined to restrict their conclusion to a model in which 2t16 must reside in the membrane. However, this is not the only interpretation and several experiments in this original work demonstrate that ectopic 2t16 (i.e., solution-localized) promotes BAX-mediated MOMP and there was no requirement to incorporate 2t16 into model liposomes. While other groups explored the hypothesis that membrane-localized 2t16 interacts with membrane-localized BAX, we chose to explore the open hypothesis that 2t16 interacts with monomeric BAX prior to association with membranes. Moreover, the majority of S1P studies highlight cytosolic and extracellular roles for this lipid and its metabolites, further providing a rationale that 2t16 is not limited to membranes. We have added this note to our text in case any readers similarly question the validity of 2t16 in the cytosol (**lines 530–533**).*

We believe that demonstrating a mechanistic role for 2t16 on monomeric conformers of BAX is both physiological and warranted. BAX activation is initiated in the cytosol and it is relevant to investigate how 2t16 may contribute to BAX early-activation distinct from investigations focused on BAX functionalization (e.g., oligomerization and pore formation).

The Reviewer critiques our approach as using "high-dose" 2t16 when in fact we limit our investigations to low micromolar concentrations. Compare this with prior publications such

as Cohen et al., 2020 in which they utilized millimolar concentrations (equivalent to 1000-fold molar excess of BAX), which, as shown in their own data, instigates the covalent modification in solution. Additionally, both Chipuk et al., 2012 and Cohen et al., 2020 demonstrate that millimolar concentrations of 2t16 can induce BAX oligomerization in a manner similar to detergents (i.e., the BAX laddering assays, which require either crosslinking, membranes, or micellar detergents to achieve). While our original submission did use the term "supraphysiological" to describe the higher end of our titrations, it was to highlight that direct activation by 2t16 at high concentrations was not likely to occur in cells; however, all doses are in the micromolar range and well-below that of prior publications. To avoid this confusion, we removed that term from our revised manuscript.

Further, the membrane environment may facilitate BAX conformations amenable to covalent derivatization, which would not be readily recapitulated by a solution system.

This is precisely the point we wish to make in the paper. It is our conclusion that when BAX is activated in the cytosol, the role of 2t16 is mediated by non-covalent interactions within the BAF; at the membrane, we believe the cysteines are more exposed and susceptible to modification by 2t16. Based on the structure of the BAX activated dimer (PDB: 4BDU), C62 is exposed to the cytosol and C126 is likely facing the mitochondrial outer membrane (though the structure does not include C126, the model is extended in Bleicken et al., 2014); the recent BAX oligomer structure further validates this orientation (PDB: 9IXU; Zhang et al., 2025). This model is supported by the compelling LUV data in Cohen et al., 2020 that demonstrates C126 modification when BAX is within the membrane.

We acknowledge that we may have not sufficiently articulated this argument or our support for the prior works, and so we have greatly revised the Discussion to better communicate these thoughts as requested by both Reviewer 2 and Reviewer 3 (lines 514–533, 604–606). Additionally, we have included the structures for the active BAX dimer and oligomer, highlighting the orientation of the relevant residues (Figure S10A–C).

The lipid, solubilised using DMSO and prewarming techniques, likely exists as heterogeneous micelles or aggregates, raising concerns regarding its bioavailability and mechanism of intracellular delivery. Intracellular entry beyond plasma membrane incorporation appears doubtful, and any 2t16 reaching the cytoplasm would likely be rapidly reduced or react with multiple off-target proteins.

Numerous publications treating cells with ectopic 2t16 do not support this point. While we cannot eliminate the possibility that 2t16 could form micelles at some critical concentration, 1) the lack of polar or charged region would likely weaken any micelle-forming behavior, thus requiring greater concentrations to induce; and 2) our deliberate avoidance of millimolar concentrations would aid in preventing any micelle or aggregate formation. We agree that any electrophile behavior of 2t16 would certainly target a myriad of cellular targets (as demonstrated in Jarugumilli et al., 2018), and this is precisely why we further interrogate the synergistic mechanism of 2t16 on BIM-activated BAX in additional model systems.

The observed cellular effects might therefore reflect a non-specific stress response culminating in apoptosis rather than direct BAX regulation. Furthermore, the use of serum starvation—a

known pro-apoptotic stimulus—as the delivery context for 2t16 further complicates the interpretation.

*We agree that the cellular data can have alternative interpretations and does not de facto prove 2t16 is directly activating BAX. We cannot dismiss the literature demonstrating that ectopic 2t16 can induce stress pathways (e.g., Kumar et al., 2011; Upadhyaya et al., 2012). Nevertheless, we believe that the rapid death kinetics suggest a direct mechanism with BAX; any induced stress response would take several hours to engage the apoptotic machinery – for instance, the time to induce BIM expression and neutralize the anti-apoptotic reservoir. Still, this is precisely why we move away from the cellular system and utilize model systems and techniques to specifically observe and analyze the direct effect of 2t16 on BAX. We have adjusted the text on to make this point more clear (**lines 121–127**).*

*Regarding the point about the serum, we believe that the Reviewer misunderstood our methodology. The cells are not serum-starved. For SPARKL assays, the growth media is removed and replaced by assay media in a two-step process: first, media containing treatments is added, then followed by media containing detection reagents. The treatments (including 2t16, ABT-737, and vehicle controls) were prepared in serum-free media to avoid albumin binding; the detection reagents were prepared in serum-containing media. The cells were first given the media containing treatments, permitted to uptake 2t16 for 30 minutes, and then were given the remaining media, which contained serum. Our controls do not indicate that this approach was detrimental to the cells. We have revised the SPARKL section of our Methods to better detail the experimental setup and avoid this misunderstanding (**lines 951–955**).*

Similar concerns extend to the in vitro data, where the observed effects could more plausibly derive from non-specific detergent-like action, as evidenced by the lack of distinction between wild-type and cysteine-mutant BAX.

This is only one interpretation of the cysteine-deficient BAX mutant data. Here, we demonstrate no loss of our phenotype when we remove the sites for aldehyde modification, which is a routine approach to investigate post-translational modifications. To view this critical control as somehow indicating a detergent-like behavior appears to be a reluctance to consider any non-covalent mechanism, which is the more straightforward interpretation of these data. We do not agree with the above interpretation for several reasons:

- 1. If 2t16 were capable of forming micelles and behaving like a detergent at the micromolar range, then this behavior would similarly call into question the results of Cohen et al., 2020, where it was used at 5 mM.*
- 2. If the observed BAX phenotypes were due to a non-specific detergent mechanism of 2t16, then we would expect to see an analogous disruption of BCL-xL/BAK^{TAMRA} in our fluorescence polarization assays, which is not observed (**Figure S3H; lines 229–231**).*
- 3. BAX activation by micellar detergent can form oligomers in solution and we do not observe this in our SEC experiments. This is a topic we have previously described in Gelles et al., 2022, and we have added this point to our text (**lines 248–250**).*
- 4. In Chipuk et al., 2012, 2t16 at 0.5 and 1 mM demonstrated an ability to directly induce BAX oligomerization in solution-based cross-linking studies. By comparison,*

the functional assays demonstrated BAX activity with 2t16 in the micromolar range. This example clearly articulates the importance of why we avoided millimolar 2t16 for mechanistic studies.

- 5. As stated above, the chemistry of 2t16 and lack of polar or charged region would reduce any micelle-forming or detergent-like behavior, thus increasing the hypothetical critical micellar concentration beyond the range that we utilize it in this work.*

We do acknowledge that readers may also share this Reviewer's concern and so we have made text adjustments to address the above points (as noted).

2. Limitations of the Mass Spectrometry Analysis:

The MS experiments relied on to discount covalent modification are not sensitive for detecting lipidated BAX due to technical limitations. Lipidated peptides are very hydrophobic and often poorly recovered in MS workflows, and activated BAX oligomers are typically resistant to proteolytic digestion and mass spectrometric detection. Thus, the absence of detectable lipid-modified peptides does not exclude the presence of covalent modifications, particularly given the methodological constraints and the limited experimental detail (e.g., lipid-to-protein ratios, incubation times) provided.

"[...] activated BAX oligomers are typically resistant to proteolytic digestion and mass spectrometric detection." This is not supported by the BAX literature. Papers describing altered proteolytic digestion of oligomeric BAX are describing membrane-bound structures of BAX (Goping et al., 1998; Lucken-Ardjomande et al., 2008; Montessuit et al., 2010). Furthermore, Jarugumilli et al., 2018 utilized trypsin digestion to analyze modified BAX by MALDI-TOF.

*However, as multiple reviewers have raised questions regarding the potential limitations of the "bottom-up" mass spectrometry approach, we conducted additional intact mass spectrometry experiments to assess the status of BAX modifications. Due to limitations of sample preparation (e.g., protein concentrations, sample injection, etc.), experiments utilizing wildtype BAX protein did not provide consistent and accurate data. Instead, we utilized oligomerization-deficient BAX mutants (BAX^{G108V}, BAX^{R109D}) to confirm no modification in response to 2t16; as a control, we did observe cysteine modification in response to iodoacetamide treatment (**Figures 2B–C; lines 167–174**).*

*While we believe the data from both mass spectrometry approaches are compelling, we agree that they are not conclusive. As such, we corroborated our conclusions by utilizing a cysteine-deficient BAX mutant (BAX^{2S}) in mechanistic and functional assays throughout the study, thereby demonstrating that synergistic activation of BAX by 2t16 is independent of cysteines (**Figures 2, 3, 5, S2, S3, and S7**). Additionally, we added experiments demonstrating that wildtype BAX modified with iodoacetamide remains sensitive to 2t16 (**Figures 2D–F; lines 175–179**). Collectively, these diverse approaches all corroborate our conclusion that, at this concentration range, 2t16 results in BAX activation through a non-covalent interaction.*

3. Inappropriate Use of Rigid-Body Docking:

The docking studies are performed using the structure of inactive BAX, whereas the proposed binding site is thought to emerge following BIM-induced conformational activation. As such, the docking is based on an inappropriate structural template, and the conclusions drawn are therefore questionable. For a protein as conformationally dynamic as BAX, molecular dynamics simulations would offer a more appropriate approach than rigid-body docking.

The model we describe is one in which $\alpha 8$ is mobilized following BIM triggering and the subsequent accessibility of the BAF permits 2t16 binding. As such, we predominantly utilize the structure of quasi-active BAX bound by BIM-SAHB with $\alpha 8$ removed (PDB 2K7W). We also include in silico investigations utilizing the inactive monomer structure (PDB 1F16) to explore the topography of the BAF in both inactive and triggered monomeric BAX. We are not investigating membrane-associated BAX and therefore do not utilize active dimer structures in our docking studies. We believe we are utilizing the most relevant and appropriate structures for our studies and are curious what structure would be considered more appropriate by this Reviewer.

We would like to remind the Reviewer (and future readers) that BAX does not immediately convert from a monomer to the active dimer, and that there is a growing appreciation for distinct conformers along the activation continuum (e.g., loop displacement, $\alpha 9$ mobilization, latch/core separation, etc.). However, these conformers are flexible and mobile, and therefore not conducive to structural biology determinations; at present, there are no structures available to represent these events prior to the active dimer.

While we absolutely agree that molecular dynamics simulations may provide additional insights and could be more informative than rigid body docking, this would represent a significant departure from our current capabilities and would require substantial time to develop. As such, we plan to explore these methodologies in future works and believe that our conclusions are properly supported by the approaches and experiments already included within this work.

4. Technical Concerns with the NMR Data and Its Interpretation Based on the Structure of Inactive BAX:

The interpretation of the NMR data raises additional concerns. First, the data are analyzed in the context of the inactive BAX structure, whereas the proposed model places non-covalent interaction of 2t16 with BAX downstream of BIM activation. This is problematic because BIM activation is known to induce conformational changes that could alter the proposed binding site, currently mapped to and evaluated based on the inactive state. Second, the timing parameters of the NMR experiments—specifically, the duration of protein-ligand incubation and the time point of NMR data acquisition—are not disclosed, limiting the ability to assess the relevance of the observations to the proposed model. Since BAX activation by 2t16 promotes dimer (as suggested by the SEC data) and ultimately oligomer formation, such species that could be induced by covalent modification would not be visible by NMR due to signal attenuation, biasing the solution NMR data toward evaluable monomeric species only for the undisclosed conditions that were applied.

Similar to the preceding comment, we are unclear which structure would be considered more appropriate by this Reviewer. BAX protein was treated with 2t16 and subjected to

1H-15N HSQC. Therefore, we utilize the monomeric structure to illustrate the positions of the residues exhibiting significant CSPs (Figure 4A). As the Reviewer states, attempting to capture a BIM-bound "triggered" BAX structure by HSQC is extremely complex and typically requires additional detergents and stabilizers to prevent BAX oligomerization during sample collection; for this reason, we studied 2t16 on unstimulated BAX protein.

Despite this, we are confident that our viewpoint does not invalidate this experimental approach. The HSQC data demonstrates a congregation of residue shifts around $\alpha 8$, which suggests the location of 2t16 interactions. Moreover, as suggested by Reviewer 1, we have added information regarding the alteration of peak morphology in the residues proximal to $\alpha 8$ (Figure 4A, right panel; lines 267–270). Our interpretation of these data is that this is where 2t16 demonstrates preferential interactions, and we strengthen this conclusion using unconstrained in silico docking (Figure 4C). Furthermore, we confirm $\alpha 8$ mobilization as a requirement for 2t16-induced BAX activation through a disulfide-tethered structural mutant (Figures 4G–I, S4H–M). Critically, while these approaches were what led to our exploration of the BAX cavity, we do not suggest that the observed CSPs represented the binding of 2t16 within the BAF; the interaction residues were further explored in later figures (Figures 6 and 7).

Regarding the specifics of the sample preparation, we apologize for this omission. Samples were prepared and immediately analyzed by HSQC, which took ~500 minutes to perform the 128 scans at the resolution of complex points we were observing. The length of the collection time is a consequence of the BAX concentration, which cannot be increased without risking auto-activation and precipitation of the sample; this is also consistent with the BAX literature. Therefore, the "treatment time" encompasses and surpasses those of our functional assays. We did not observe significant 2t16-induced loss of signal or peak intensities suggestive of protein loss (due to either activation-induced multimerization or degradation). We have adjusted the Methods section to include these details (lines 1192–1206) and added a comment about sample stability to the Results (lines 270–272).

5. Control Compound and Binding Orientation Issues:

Based on the authors' non-covalent ligand binding model, there is little mechanistic rationale for why the reduced form (16CHO) would fail to bind and activate BAX, whereas the α,β -unsaturated aldehyde (2t16) would. Both compounds are featureless hydrophobic chains differing only at the terminal functional group. If the entirety of the binding mechanism was dependent on the presence of the double bond (however unlikely), a key control beyond 16CHO would be the alcohol derivative (trans-2-hexadecen-1-ol), which maintains the double bond but removes the electrophilic aldehyde functionality.

This is an important suggestion and we thank the reviewer. While we can replicate the lack of BAX activation with the saturated aldehyde, hexadecanal, we have not yet discovered the underlying mechanism. While we could not source the molecule specifically suggested by the Reviewer (trans-2-hexadecen-1-ol), we did explore this suggestion in principle. We utilized a hydrazide molecule that would react with the aldehyde head group of 2t16 and observed that the resulting molecule (2t16-NN-EtOH) was incapable of activating BAX or synergizing with BIM (Figures 1J–K, S1F; lines 141–143). Later in the paper, we perform in silico docking simulations with

2t16-NN-EtOH and observed altered orientations within the BAF due to the new larger headgroup (Figures S8A–B; lines 389–396). These data corroborate our interpretation that the functional consequence of 2t16 is a result of residing in the neck of the BAF, which is lost with 2t16-NN-EtOH molecule.

Furthermore, the docking poses suggest an unnatural orientation, with the lipid tail (ordinarily embedded within the membrane) proposed to insert into the hydrophobic groove—an orientation inconsistent with membrane biophysics. Instead, a model involving covalent reaction between a cysteine on the BAX surface and a membrane-exposed and reactive lipid head group appears more plausible, as previously detected and validated in the native membrane context.

The Reviewer is perhaps inadvertently contradicting themselves with this statement, and appears to be confusing the details of prior works. Both Jarugumilli et al., 2018 and Cohen et al., 2020 describe a Michael addition via the β -carbon of the double bond; neither work suggests a Schiff reaction with the aldehyde headgroup. Moreover, neither previous work nor our own implicates the BAX hydrophobic groove (formed by helices α 3, α 4, and α 5). Finally, the lipid-protein overlay assay described in Chipuk et al., 2012 would not have formed a covalent bond and is the most direct example that 2t16 interacts with monomeric BAX structures. Therefore, the model as suggested here by the Reviewer is not supported by the published data.

"[...] the native membrane context." Again, we remind the Reviewer that native BAX resides in the cytosol – not the membrane – and that BAX activation occurs in solution prior to translocation to the mitochondrial outer membrane. Our perspective is that regulation of BAX activation occurs predominantly at this stage in the cytosol. Therefore, this work explores the role of 2t16 interactions on monomeric BAX activation prior to translocation, association with membranes, or oligomerization.

We agree that the question regarding 2t16 orientation is a good point. All of our molecular docking investigations resulted in a "tail-first" orientation of 2t16 in the BAF. We were only able to observe a "head-first" orientation if we introduced a constraint forcing the oxygen atom to be in proximity with select BAF residues. Curiously, this did result in a slight benefit to docking score even though the pose could not be replicated in the unconstrained attempts. In light of these results, we are incorporating statements in the Results describing this outcome and that we cannot exclude the possibility of the reverse orientation (lines 397–402). However, as our model describes 2t16 residence in the depth of the BAF as the pro-apoptotic phenomenon – which is supported by the new data with the loss-of-function molecule, 2t16-NN-EtOH (Figures 1J–K, S1F, S8A–B; lines 141–143, 389–396) – we do not believe that this open question of orientation undermines the conclusions of this work.

6. Interpretation of Mutagenesis Data:

The mutagenesis experiments do not conclusively validate the proposed lipid-binding mechanism. Mutations within the putative binding site could alter BAX activation through a range of alternative structural or functional mechanisms. For example, the P186G mutant is known to exhibit reduced activity, but attributing this to impaired lipid binding is speculative without further rigorous characterisation.

We agree that our mutagenesis approaches may not be 100% conclusive – no singular approach ever is – but collectively, we posit our complementary assays are compelling. This comment is selectively highlighting a known loss-of-function mutant while omitting the relevant controls and data we provide for each of the mutants throughout the paper.

- 1. We corroborated the synergistic effect of 2t16 on a cysteine-deficient mutant of BAX (BAX^{2S}) (Figures 2, 3, 5, S2, S3, and S7).*
- 2. We test our hypothesis that 2t16 binds the BAF through the identification of 3 separate mutants, each of which we validate has having no impaired response to BIM activation or ability to permeabilize LUVs (Figures S8C–D).*
- 3. Regarding P168G, we explicitly comment that this mutant has reduced activity (lines 454–456) and demonstrate the reduced, but not abrogated, response to BIM and ability to permeabilize LUVs (Figures S9B).*
- 4. In the revised manuscript, we now include Bliss synergy calculations to quantitatively determine that the BAF mutants exhibit loss of sensitivity to 2t16-induced synergy (Figure 6H).*
- 5. In the case of P168G, the Bliss calculations quantify the loss of 2t16 synergy while accounting for the reduced sensitivity to BIM (Figure 7F–G).*
- 6. We specifically study the loss of 2t16 sensitivity on P168G in our BAX activation assays (e.g., FLAMBE, LUV permeabilization) prior to experimentation in cellular models in order to contextualize the results from cellular model systems.*

While we agree with this Reviewer that the loss of sensitivity to 2t16 in reconstituted double knockout MEFs could be the result of a general reduced activity of the P168G mutant, this is why we first assessed the response of P168G to 2t16 in our BAX activation assays (Figures 7D–G). We have adjusted the language of our results section to avoid unwarranted overinterpretation of the results (lines 463–467).

Summary:

This manuscript offers interesting insights into potential non-covalent interactions between 2t16 and BAX that may influence BAX function. However, the evidence provided does not sufficiently support the conclusion that 2t16 acts predominantly via a non-covalent mechanism. The data appear more consistent with a complementary non-covalent contribution within an overarching covalent regulatory process, particularly in light of established principles of lipid electrophile biology (including diminished activity of the reduced analog) and lipid orientation within membranes.

The study would benefit from additional experimentation in membrane systems, a broader exploration of covalent interactions under physiologically relevant conditions, cellular proof of direct non-covalent interaction with BAX, and a more cautious framing of its conclusions.

Finally, the characterisation of prior work as limited or flawed, whilst employing even more supra-physiological and indirect methodologies in solution, appears unnecessarily dismissive and detracts from the overall scientific narrative. Further, indicating that “structural and molecular mechanisms remain largely unknown” does not accurately reflect the prior literature on 2t16 and BAX. A more balanced and integrative perspective—that covalent and non-covalent features

likely cooperate in BAX regulation by 2t16—would better reflect the available data and provide a stronger foundation for future studies.

We greatly regret and apologize if our original submission was interpreted as dismissive of prior works. This was not our intention, and we may have been a bit overzealous while advocating for our findings and providing distinction between our approach and the approaches of others. We have adjusted our language throughout the manuscript, with particular attention to sections remarking on the other works (lines 494–533). Though as requested by Reviewer 2, we do include commentary on how the different approaches may have resulted in divergent conclusions; however, the language has been adjusted to avoid a negative or dismissive interpretation. (lines 494–513). Similarly, we have adjusted the language in our introduction to better articulate what remains "an open question" (lines 84–86).

*To be clear and reiterate: we believe that each work studies distinct biology, and that BAX is likely being regulated both biophysically (interactions with the BAF) and biochemically (cysteine modification) by 2t16 in discrete stages and environments throughout the BAX activation continuum. It is our conclusion that when BAX is activated in the cytosol, the role of 2t16 is mediated by non-covalent interactions within the BAF; at the membrane, we believe the cysteines are more exposed and susceptible to modification by 2t16, which is supported by structures of BAX at the membrane. We have reworked our Discussion to explore this cohesive model more thoroughly, as requested by both Reviewer 2 and Reviewer 3 (lines 523–529, 602–606), and included relevant structures of active multimeric BAX to support this more inclusive conclusion (**Figure S10A–C**).*

We respectfully request that the Reviewer re-evaluates if our defined and faithful model systems soundly represent a means to study the BAX activation continuum and its relevant physiological biology – model systems that established and support the majority of what we currently know about pro-apoptotic BCL-2 proteins. Again,

- 1. BAX is activated and regulated in the cytosol, and therefore studying BAX prior to membrane associations is valid and warranted.*
- 2. 2t16 is a soluble long-chain aldehyde and not restricted only to membranes.*
- 3. Chipuk et al., 2012 included several studies suggestive of interaction between 2t16 and BAX prior to membrane association or oligomerization.*
- 4. While millimolar concentrations of 2t16 exhibit covalent binding against BAX and a myriad of cellular substrates, the cellular concentration is orders of magnitude lower; we therefore utilize concentrations that are more representative of the cellular environment (low micromolar).*

We believe that we represent a fundamentally different philosophy of BAX biology compared with this Reviewer, and therefore it appears we have differing opinions on how best to study BAX – and this is ok! We may not share the same perspective, but we hope that our revised manuscript and this thorough response document help explain our position and approach, and that we can respectfully agree to disagree on a few points. That being said, we absolutely agree with the tone check and thank this Reviewer for pushing us to prepare a more cohesive, comprehensive, and inclusive paper, and we believe the revised Discussion is much improved.

References

- Antonsson B, Montessuit S, Lauper S, Eskes R, Martinou JC. Bax oligomerization is required for channel-forming activity in liposomes and to trigger cytochrome c release from mitochondria. *Biochem J.* 2000 Jan 15;345 Pt 2(Pt 2):271-8. PMID: 10620504; PMCID: PMC1220756.
- Bleicken S, Jeschke G, Stegmüller C, Salvador-Gallego R, García-Sáez AJ, Bordignon E. Structural model of active Bax at the membrane. *Mol Cell.* 2014 Nov 20;56(4):496-505. doi: 10.1016/j.molcel.2014.09.022. Epub 2014 Oct 30. PMID: 25458844; PMCID: PMC4869853.
- Bleicken S, Assafa TE, Stegmüller C, Wittig A, Garcia-Saez AJ, Bordignon E. Topology of active, membrane-embedded Bax in the context of a toroidal pore. *Cell Death Differ.* 2018 Nov;25(10):1717-1731. doi: 10.1038/s41418-018-0184-6. Epub 2018 Sep 5. PMID: 30185826; PMCID: PMC6180131.
- Chipuk JE, McStay GP, Bharti A, Kuwana T, Clarke CJ, Siskind LJ, Obeid LM, Green DR. Sphingolipid metabolism cooperates with BAK and BAX to promote the mitochondrial pathway of apoptosis. *Cell.* 2012 Mar 2;148(5):988-1000. doi: 10.1016/j.cell.2012.01.038. PMID: 22385963; PMCID: PMC3506012.
- Cohen DT, Wales TE, McHenry MW, Engen JR, Walensky LD. Site-Dependent Cysteine Lipidation Potentiates the Activation of Proapoptotic BAX. *Cell Rep.* 2020 Mar 10;30(10):3229-3239.e6. doi: 10.1016/j.celrep.2020.02.057. PMID: 32160532; PMCID: PMC7343539.
- Czabotar PE, Westphal D, Dewson G, Ma S, Hockings C, Fairlie WD, Lee EF, Yao S, Robin AY, Smith BJ, Huang DC, Kluck RM, Adams JM, Colman PM. Bax crystal structures reveal how BH3 domains activate Bax and nucleate its oligomerization to induce apoptosis. *Cell.* 2013 Jan 31;152(3):519-31. doi: 10.1016/j.cell.2012.12.031. PMID: 23374347.
- Gavathiotis E, Reyna DE, Davis ML, Bird GH, Walensky LD. BH3-triggered structural reorganization drives the activation of proapoptotic BAX. *Mol Cell.* 2010 Nov 12;40(3):481-92. doi: 10.1016/j.molcel.2010.10.019. PMID: 21070973; PMCID: PMC3050027.
- Gelles JD, Mohammed JN, Chen Y, Sebastian TM, Chipuk JE. A kinetic fluorescence polarization ligand assay for monitoring BAX early activation. *Cell Rep Methods.* 2022 Mar 28;2(3):100174. doi: 10.1016/j.crmeth.2022.100174. Epub 2022 Mar 9. PMID: 35419554; PMCID: PMC9004659.
- Goping IS, Gross A, Lavoie JN, Nguyen M, Jemmerson R, Roth K, Korsmeyer SJ, Shore GC. Regulated targeting of BAX to mitochondria. *J Cell Biol.* 1998 Oct 5;143(1):207-15. doi: 10.1083/jcb.143.1.207. PMID: 9763432; PMCID: PMC2132805.
- Jarugumilli GK, Choi JR, Chan P, Yu M, Sun Y, Chen B, Niu J, DeRan M, Zheng B, Zoeller R, Lin C, Wu X. Chemical Probe to Identify the Cellular Targets of the Reactive Lipid Metabolite 2-trans-Hexadecenal. *ACS Chem Biol.* 2018 May 18;13(5):1130-1136. doi: 10.1021/acschembio.7b01063. Epub 2018 Apr 6. PMID: 29608264; PMCID: PMC5959771.
- Kumar A, Byun HS, Bittman R, Saba JD. The sphingolipid degradation product trans-2-hexadecenal induces cytoskeletal reorganization and apoptosis in a JNK-dependent manner.

Cell Signal. 2011 Jul;23(7):1144-52. doi: 10.1016/j.cellsig.2011.02.009. Epub 2011 Mar 6. Erratum in: *Cell Signal.* 2012 Jan;24(1):351. PMID: 21385609; PMCID: PMC3086202.

Lucken-Ardjomande S, Montessuit S, Martinou JC. Bax activation and stress-induced apoptosis delayed by the accumulation of cholesterol in mitochondrial membranes. *Cell Death Differ.* 2008 Mar;15(3):484-93. doi: 10.1038/sj.cdd.4402280. Epub 2007 Dec 14. PMID: 18084240.

Montessuit S, Somasekharan SP, Terrones O, Lucken-Ardjomande S, Herzig S, Schwarzenbacher R, Manstein DJ, Bossy-Wetzel E, Basañez G, Meda P, Martinou JC. Membrane remodeling induced by the dynamin-related protein Drp1 stimulates Bax oligomerization. *Cell.* 2010 Sep 17;142(6):889-901. doi: 10.1016/j.cell.2010.08.017. PMID: 20850011; PMCID: PMC4115189.

Suzuki M, Youle RJ, Tjandra N. Structure of Bax: coregulation of dimer formation and intracellular localization. *Cell.* 2000 Nov 10;103(4):645-54. doi: 10.1016/s0092-8674(00)00167-7. PMID: 11106734.

Upadhyaya P, Kumar A, Byun HS, Bittman R, Saba JD, Hecht SS. The sphingolipid degradation product trans-2-hexadecenal forms adducts with DNA. *Biochem Biophys Res Commun.* 2012 Jul 20;424(1):18-21. doi: 10.1016/j.bbrc.2012.06.012. Epub 2012 Jun 19. PMID: 22727907; PMCID: PMC3402648.

Zhang Y, Tian L, Huang G, Ge X, Kong F, Wang P, Xu Y, Shi Y. Structural basis of BAX pore formation. *Science.* 2025 Jun 26;388(6754):eadv4314. doi: 10.1126/science.adv4314. Epub 2025 Jun 26. PMID: 40570108.

RESPONSE TO REVIEWERS

Reviewer #1:

The authors have addressed my comments adequately.

We are thrilled that the Reviewer supports the revised manuscript.

Reviewer #2:

The authors have satisfactorily addressed my comments with new data, clarifications, and revisions of the text.

They investigated two BAX mutants with reduced oligomerization capacity for evidence of covalent adducts formed by 2t-hexadecenal binding and evaluated the BAX-2S mutant (C62S/C126S), which lacks reactive cysteines and therefore cannot form covalent adducts. These new experiments and data strengthen the authors' conclusions and underscore a role for 2t-hexadecenal in mediating early BAX activation in the cytosol. This is also consistent with the structures of BAX conformations available. While prior studies demonstrated covalent lipid adduction to BAX, the present work offers an alternative and potentially complementary mechanism for 2t-hexadecenal. The authors have clearly articulated the study's conclusions and limitations in the revised manuscript and reviewer's responses. Overall, this is a coherent and well-executed study and I recommend publication.

We appreciate this Reviewer's summary of our new data and revised manuscript, and agree that the revised work has been greatly improved in response to the Reviewers' comments.

Reviewer #3:

I appreciate the revision efforts and the clarifying remarks with respect to rationale and methodology. The work introduces interesting tools and concepts regarding early BAX activation, but several interpretive and methodological aspects remain overstated and highly speculative relative to the data.

We thank the reviewer for appreciating our revision efforts and clarifying remarks, and for the encouraging perspectives on our experimental tools and concepts. We apologize that our first resubmission was not satisfactory for this Reviewer, and are pleased to have the opportunity to improve the manuscript in a second round of revision. Below, we provide detailed responses to the Reviewer's lingering concerns, and hope that this persuades the Reviewer to agree with Reviewers 1 and 2 that our conclusions are properly supported by the approaches, experiments, and controls included within this work.

1. Positioning of prior work.

It would be appropriate to acknowledge up front that covalent modification of BAX by 2t-hexadecenal (2t16) at Cys126 has already been demonstrated in membrane and cellular contexts with functional consequences for MOMP (as opposed to indicating in the abstract and introduction that the molecular mechanism involving 2t16 is “largely unknown” and “remains an open question”). Introducing and acknowledging prior work does not undermine the current work. In accordance with this handling, the discussion concentrates chiefly on the in vitro concentrations used in earlier covalent studies whilst overlooking the accompanying physiological findings that validated these mechanisms in native mitochondria and intact cells in a cysteine-specific manner. The selective emphasis on mM dosing in prior in vitro studies gives an incomplete picture and may inadvertently mislead readers regarding the existing mechanistic evidence and sets a tone that authors are trying hard to “sell” their current work as a controversial alternative. It would be more objective and balanced to frame the present work as exploring non-covalent contributions that complement covalent mechanisms already confirmed in physiologic settings.

We genuinely appreciate the reviewer's supportive position for distinct, yet potentially, complementary activities of 2t16 on BAX. Therefore, we agree that updating the language to align with the above statements is an excellent suggestion. We edited the Abstract and Introduction (specifically, lines 84–88) to capture these concepts and introduce the prior works earlier in the manuscript, and we further honed the Discussion to include additional interpretations from the previous publications. However, we do believe it is accurate to state that the role of 2t16 on monomeric BAX activation, distinct from membrane-associated multimers, remains an open question worthy of further investigation as it has not been decisively studied by prior works.

As an aside, the physiological findings referred to above are more biochemistry leaning than cell biology and/or physiological as no interactions between endogenous BAX and endogenous 2t16 were published. The use of “native mitochondria” (i.e., isolated mitochondria) was an extension of the original study we published in Cell where recombinant BAX was treated with exogenous 2t16 or recombinant S1P lyase in the presence of BIM peptide. It is worth noting that the protocol used for isolating mitochondria also includes heterotypic membranes (this was the premise of the original Cell publication).

Finally, the "emphasis on mM dosing" is important to include as (we believe) it provides the nuance necessary to explain the different observations between the works. Indeed, Reviewer 2 specifically requested this additional commentary on experimental details that may have resulted in the distinct phenotypes, and we agree that this is valuable to the readership who may otherwise believe that the mechanism of 2t16 on BAX activation is fully described. Furthermore, highlighting the role of substrate concentration in a proposed chemical reaction is entirely warranted and relevant. While this Reviewer feels these details are disparaging in nature, we believe they are constructive to the larger discourse and we explicitly conclude this discussion by positing that both non-covalent and covalent contributions of 2t16 are biologically relevant to BAX activation (lines 516–535, 606–608).

2. Plausibility of cytosolic delivery.

The proposal that exogenously applied 2t16 freely enters the cytosol and engages BAX to account for the observed results remains speculative and unlikely – and the authors provide no evidence that lipid-BAX binding in the cytosol is actually occurring (nor that 2t16 even reached the cytosol in reacted or unreacted form). It is so well known that α,β -unsaturated aldehydes are highly reactive electrophiles which, if present in the media and if even able to dislodge from the cell membrane to access the cytosol, would be expected to form adducts with numerous nucleophilic targets, including glutathione and abundant soluble proteins (the authors refer to hundreds of potential reactive targets). In this light, a BAX-related effect to explain the Figure 1A-B data appears so implausible as to undermine the subsequent structure-function work. Starting the manuscript on such thin ice (as in, this is our interpretation yet here are the multitude of caveats that more likely make it not a reliable interpretation) is undermining. The early data is among the weakest data (lack of actual dose-responses, many alternative interpretations, etc.).

To briefly summarize Figures 1A–C, we generated kinetic cellular data suggesting that 2t16-dependent cell death requires BAX and BIM, and this phenotype is regulated by anti-apoptotic BCL-2 proteins. Importantly, our stated conclusion is broad and only suggests that 2t16 "induces apoptotic cell death by acting on effector BCL-2 family proteins", but we have further softened the language to avoid overinterpretation (lines 121–122).

Immediately after describing these observations, we stated, "There are reports that ectopic hexadecenal can form adducts with DNA and generate oxidative stress resulting in apoptosis, and we could not eliminate the possibility that ectopic hexadecenal was targeting additional substrates or stress pathways. Therefore, we interrogated whether hexadecenal directly promoted BAX-mediated pore formation by utilizing recombinant BAX protein and large unilamellar vesicles (LUVs), which are biochemically-defined liposomes that mimic the major lipid composition of the OMM, and assessed BAX activation by measuring LUV permeabilization" (lines 123–129). Within the previous revised document, we fully acknowledged that alternatives exist, and we focused our investigations into a singular pathway to explore BAX biology. In the remainder of the revised manuscript, we then identified and linked direct biochemical, biophysical, and structural relationships between 2t16 and BAX. Given the word limitations for the journal, we cannot exhaustively discuss experimental limitations of the work at each figure panel. We propose that this approach is no different from adding a growth factor to a cell and exploring one node of a signaling pathway or cellular phenotype in isolation; there is no purpose in overly undermining the work with copious alternative interpretations within the Results section. As such, we believe that the progression of our experimental approaches

and results is the most appropriate and effective order to thoroughly characterize and report our findings.

The comment that “...exogenously applied 2t16 freely enters the cytosol...” is the Reviewer’s personal synthesis, which is not based on our data, interpretation, or discussion. In the 2t16 literature, it is known that the aldehyde forms adducts (albeit potentially reversible) with glutathione and serum albumin, yet it is commonly added to cells to study its activities (the doses utilized account for the proportion binding to glutathione, for example). Similarly, we noted that lowering the serum aided in the cellular response to 2t16. As the focus of the current work is not the mechanism of 2t16 uptake, we did not mention proposed uptake scenarios because it is a frequently used approach to study 2t16; however, this would be interesting biology to pursue, and would likely include investigations of endocytosis, micropinocytosis, and carrier proteins. Furthermore, the sphingolipid field’s early work in the 1990s that identified the first intersections with cell death were based on extracellular/ectopic delivery of numerous amphipathic reactive species, which also served as the foundation for our understanding of how extracellular sphingolipid gradients activate GPCRs and inform the immune compartment.

Finally, in contrast to the Reviewer’s note, we indeed included multiple 2t16 doses in all cellular experiments (Figures 1A–C, 7G–I) and extensively throughout the approach to ensure scientific rigor.

3. Clarification regarding lipid orientation and membrane context.

In the response to point 5, the authors appear to have misunderstood an important aspect of the critique. The concern was not about Schiff-base formation with an aldehyde given the Michael addition mechanism, but instead about lipid orientation: the suggestion that the lipid tail of 2t16 inserts into the hydrophobic cleft of BAX (the hydrophobic BAF site not the BC groove) is difficult to reconcile with the disposition of endogenous ER-to-mitochondrial transfer of these lipids, whose hydrophobic tails are embedded within or associated with membranes. A soluble, tail-inserted lipid electrophile acting on cytosolic BAX is conceptually inconsistent with known membrane biology. I appreciate that the authors believe that perhaps 2t16 can find its way to the solution phase, but they never test or show it by fractionation MS or otherwise. In addition, there is no data presented from cells or cell lysates that show this lipid is associated with soluble BAX. Such results are so fundamental to the study – so proceeding without them makes this a really hard sell.

At present, there is no generally accepted mechanism for how lipid orientation or membrane contexts support S1P metabolism into 2t16 nor release of 2t16 from a hydrophobic environment. The S1P lyase active site faces the cytosol, suggesting that the catalysis of S1P occurs in solution and in trans-membrane structures. For 2t16 metabolism by FALDH, there are implications of enzyme-membrane juxtaposition, 2t16 “sliding” along an ER membrane to enter at the cytosolic/membrane protein interface, and potentially 2t16 transient binding to/from glutathione, but none are uncontested as to how the active site accesses 2t16 prior to catalysis. Additionally, the Reviewer’s assumptions regarding the localization and suspected membrane orientation of 2t16 is based on behavior of bulky and complex amphipathic lipids (e.g., phospholipids, glycerolipids) and does not reflect that 1) S1P and α,β -unsaturated aldehydes are soluble moieties and 2) they target cellular substrates that are not associated with membranes. As there is no “consistent with

membrane biology” literature to explain how 2t16 orientation supports its metabolism, this is clearly an important area of investigation for the future, but this knowledge gap should not hinder the present study as we provide an additional premise of protein biochemistry to better examine 2t16 metabolism and its consequences.

Despite this, we addressed this comment during the prior round of revision by: 1) introducing additional constraints to our molecular docking to generate a "head-first" orientation (though we were not able to replicate this result without the forced constraints); and 2) adding a section to the Results describing the additional modeling and explicitly stating that the orientation of 2t16 could not be conclusively determined (lines 399–404). Aside from attempting protein crystallography of 2t16 and BAX (which involves significant technical hurdles and limitations), we are unclear how else this comment could be addressed. As such, we provide a rational interpretation based on this literature and in silico tools to glean interpretations, but we clearly indicate in our manuscript that this mechanism is not defined.

The reviewer indicates several times that the cytosolic interaction between 2t16 and BAX has been convincingly published (which was originally implicated by our functional data in Cell using isolated primary mitochondria from fresh liver and endogenous S1P lyase). We reproduce this interaction in vitro and extended with new insights into a binding domain and functional consequences – also with a disease related BAX mutant. Regarding the use of cell lysates to observe the complex, there is currently no approach to detect endogenous 2t16; all published methods utilized exogenous and derivatized 2t16 to measure S1P lyase activity. These approaches significantly change the shape of 2t16 and involve multiple organics that exclude detecting a protein complex. Furthermore, we remind the Reviewer that we propose a non-covalent interaction between 2t16 and BAX, and, as such, attempts to isolate cellular BAX would not preserve this interaction.

4. Non-covalent versus covalent framing.

The failure of non-reactive analogues (saturated or hydrazide-modified aldehydes) to activate BAX supports the relevance of the α,β -unsaturated moiety to a covalent mechanism, not a non-covalent binding mechanism. The reader will know this, so to interpret this result otherwise is inconsistent with biochemical principles. Along these lines, the cysteine-mutant data do not disprove a covalent mechanism, they only allow for study of an isolated non-covalent contribution. Therefore, the result section subtitles and interpretations in the abstract and conclusion should emphasize the study of a non-covalent contribution rather than position the results as proving a non-covalent only mechanism, which the data simply do not support (and more often suggest the opposite). The NMR data show that the non-reactive 16CHO compound causes NMR CSPs, which would be expected as the non-covalent component of a covalent process, because the specificity of physiologic covalent reactivity and signaling relies on non-covalent interaction to template and reinforce selectivity of the covalent component of the interaction.

We would like to point out that the non-reactive hydrazide product retains the reactive trans-double bond (a Michael addition as detailed in both Jarugumilli et al., 2018 and Cohen et al., 2020), so the suggestion that the hydrazide reaction supports covalent reaction is inconsistent with the experimental design, results, and scientific literature. We also disagree with this Reviewer's extremely selective interpretation of the NMR data and feel compelled to remind them that, in addition to 16CHO demonstrating distinct CSP

patterns, short-chain aldehydes (which retain reactivity for cysteine residues via the trans-double bond) similarly demonstrated altered CSP profiles. Critically, neither cysteine residue demonstrated significant CSPs in response to 2t16, so the Reviewer's proposed explanation of the NMR data is not supported by our data.

*To reiterate our position on covalent versus non-covalent mechanisms (as discussed in response to other comments in this document, and in the previous revision cycle): we are not denying the covalent mechanism, but given cysteine-free monomeric BAX cooperates with BIM and 2t16 to activate and permeabilize membranes, this suggests that alternative, non-covalent mechanisms warrant exploration and these data should be available to the scientific community. Our section subtitles and conclusion text appropriately reflect that our data support (we do not use the term "prove") the non-covalent mechanism, and we provide ample space and consideration in the manuscript to describing the relevance of the covalent mechanism as well (**lines 516–535, 606–608**). We have made multiple additional adjustments to the text to ensure that the reader does not interpret our conclusions as discounting the existence of a covalent mechanism (for example, **lines 164–165, 186–188, 482–485**).*

As a larger point, numerous mechanisms explaining several fundamentals within the apoptotic cascade have driven our field for decades. For example, the decade-long disagreement regarding the role of BH3-only proteins in BAX/BAK activation, which ultimately revealed the direct activator and sensitizer/de-repressor BCL-2 classes; or the ongoing 20+ year debate as to whether or not the BH3-only protein PUMA is a direct activator. Similar to these examples, previous literature on 2t16/BAX and our current manuscript will support scientific engagement and the establishment of elegant model systems to explore the interactions between the entire BCL-2 family and lipids, in which very little is known. We hope the reviewer appreciates the critical importance of the scientific literature to not only reinforce past work but also challenge and evolve.

5. Mass-spectrometry limitations and docking language.

The authors' negative MS data do not exclude covalent modification (particularly since two independent groups demonstrated otherwise), and the docking studies would be better presented as hypothesis-generating rather than structural evidence. Since MD simulations are not performed, "docking simulations" terminology should be changed to "rigid-body docking" or "static docking calculations".

*The Reviewer appears to interpret our revised manuscript as continuing to argue against and invalidate prior works, which we simply do not, as based on our Introduction and Discussion sections. "The authors' negative MS data do not exclude covalent modification...", "the cysteine-mutant data do not disprove a covalent mechanism..." – we use these approaches to demonstrate that our observed phenotype is not due to covalent modification, and we explicitly incorporate the conclusions of the prior works in our interpretations and discussion. These comments were raised in the previous revision cycle, and our response to these concerns are maintained in the re-revised manuscript (**lines 160–164, 186–188, 480–482, 516–521, 525–531, 606–608**). It is unrealistic to include a qualifying statement regarding the prior works for every single panel of data that indicates we are studying a non-covalent mechanism throughout the Results section; instead, the*

Discussion is the appropriate section of the paper to compare, contrast, and consider these differences.

Furthermore, we do indeed use the docking studies as hypothesis generating and then experimentally examine the in silico data with several structural mutants (both non-functional and functional) using numerous experimental model systems (Figures 4, 6, 7, S4, S8, S9). For clarity, we doubled checked and refined the language to ensure our scientific process is more obvious to the reader.

Finally, we agree that since MD simulations are not performed, the terminology was updated to remove any erroneous use of the term "simulation". Thank you for this correction.

6. Clarifying the dosing context.

The authors emphasize that they are using micromolar rather than millimolar dosing for in vitro studies, but in the response the authors also indicate that in one set of experiments the molar excess is up to 40,000-fold compared to BAX, which undermines the argument that this study is more conservative or physiologic with respect to dosing levels compared to prior ones. The in vitro setting is just that – it is non-physiologic by definition and is used to hypothesis generate for ultimate testing and validation in cells and tissues.

We firmly believe that utilizing micromolar and not millimolar concentrations of 2t16 more closely resembles cellular contexts – there is no argument here as nanomolar levels in cells are reported in the literature, yet levels can be amplified to micromolar upon inhibition of the S1P lyase. The experiment with 40,000-fold 2t16 is to saturate the binding curve and demonstrate that the effect does NOT happen despite the incredible molar excess, which is both a rational and anticipated investigation. This singular technical approach does not undermine the study as suggested by the Reviewer's distortion of the methodologies. While the Reviewer appears to make the assertion that the specifics of in vitro model systems are unimportant because they are never truly physiological, we wish to remind them that the role of substrate concentration is extremely relevant to a proposed covalent reaction and thus warrants highlighting. To ensure our intentions are clear in the manuscript, we adjusted the text accordingly.

7. Lingering structural methodology concerns.

The manuscript remains heavily reliant on static docking calculations, which constitute a minimal and low-rigor approach given the central role of computation in their conclusions. While the authors argue that molecular dynamics (MD) simulations are beyond scope, this is unpersuasive; MD is precisely the method needed to capture conformational variability, address structural perturbations (e.g., $\alpha 8$ deletion), and evaluate stability of binding modes. The current strategy does not adequately represent the relevant ensembles or intermediates, and the terminology conflating docking with simulation is misleading. At minimum, the authors should employ explicit solvent MD sampling of existing structures, relax mutant models, and dock to representative ensembles. Without this, the work does not meet basic standards for a high-impact publication.

We certainly appreciate the suggestion to enhance our manuscript with additional in silico data. Yet, molecular dynamics simulations are inherently theoretical and rely on force fields that approximate molecular behavior, and without experimental validation (e.g.,

NMR), the results remain purely speculative. For our approach, we instead performed NMR as the first step prior to rigid-docking and the identification of BAX mutants to investigate in the manuscript. We realize this may be perceived as a departure from the normal workflow, but as early conformational changes of monomeric BAX beyond BIM stimulation are not well characterized at the structural level, we focused on the structure of quasi-active BAX bound by BIM-SAHB with $\alpha 8$ removed (PDB 2K7W), and supported these observations to include the inactive monomer structure (PDB 1F16) to explore BAF topology in both inactive and triggered monomeric BAX.

In good faith, and to ensure our strategy was acceptable, we recently discussed this Reviewer's molecular dynamics comment with several experts in the field, and all concluded that given our workflow to first perform NMR and then subsequent experimentation with several mutants to support the NMR and rigid docking, the addition of molecular dynamics would not influence the conclusions of our paper, yet it would require months of effort with parameters that are largely speculative and largely assumed based on biochemical systems rather than cell biology. If we were asserting specific binding affinities, then advanced MD simulations would likely provide more precise metrics; however, our docking-based comparisons are relative in nature (e.g., a modification results in disordered binding predictions), are used as hypothesis-generating, and are experimentally validated with structural mutants. As such, the Reviewer's suggestion would not change the following conclusions:

1. The BAF is present in all ensembles of the structures modeled in this work (**lines 285–287, 299–302, Figures S4F–G**);
2. The BAF cavity is represented in several BAX structures, including crystal structures and solution NMR structures (**lines 574–579, Figure S10D**);
3. Mutagenesis of BAF-forming residues selectively loses sensitivity to 2t16 while remaining sensitive to BIM activation (**lines 416–424; Figures 6G–H, S8C–F**).

Given the utilization of more than 15 techniques and analytical tools, integration of numerous cell biology concepts, and the description of a new domain with the BCL-2 family, we do not agree with the specious comment that our applied expertise within the manuscript fails to meet basic standards for a high-impact publication.

To summarize, the authors work so hard at justifying the new model that they undermine their credibility by making comparisons and justifications that aren't objectively effective or necessary based on the data provided. Making the suggested textual refinements, with an overall more objective approach to data interpretation would allow the manuscript to stand as a more constructive extension of previous models rather than a competing reinterpretation, whilst fully crediting established physiological data and situating the proposed mechanism within the known biochemistry of lipid electrophiles. It may be "OK" to directly say, this is what we speculate based on our results despite the caveats, some of our own data (and prior data) are inconsistent with our new model, and even quite inconsistent with what the reader might expect from lipid electrophiles and how they are used in the study. At the end of the day, a more circumspect handling seems not only justified, but could ultimately come across more genuine and compelling.

We appreciate the reviewer's expert and candid advice, and we believe that the integration of these comments has improved the accuracy, tone, and data interpretation. Thank you.

RESPONSE TO REVIEWERS

Reviewer #3:

The authors' responses address several points, but many rely more on rhetorical framing than on directly engaging the underlying scientific issues. To be constructive and avoid unnecessary back-and-forth, I offer a few specific examples where clarification or correction would strengthen the manuscript. My intention is sincerely to help refine the discourse to ensure that the authors' interpretations align with biochemical principles and the data presented.

We thank the Reviewer for their constructive criticism and suggestions for how we may further clarify and strengthen the manuscript. We interpret that several of the Reviewer's comments regarding our word choices were perceived as weakening published works, but our intentions were to aid readers in following our experimental progression. In good faith, we have made additional text adjustments to help avoid any misinterpretations of our language and address several of the points raised by the Reviewer

Line 4, Title: Consider whether “lipids” should be singular, as the study focuses on one lipid species.

We understand the Reviewer's comment. However, we demonstrate (using in silico and experimental data) that similar long-chain alkenals also activate BAX, thus leading to our conclusion that the alkenal length is a determinant of the interaction (likely due to increased specificity for the BAF) (Figures 5, S5, S6, and S7). While hexadecenal is likely the predominant bioactive lipid in the cell, other long-chain aldehydes may similarly interact with BAX. As such, we believe the current title is accurate to the findings and keeps the field open to additional interactions and discovery.

Lines 34–35, Abstract: The phrase “mechanisms remain underexplored” is not strictly accurate, as prior work included detailed structure–function studies. The sentence reads clearly without this clause.

We have removed the phrase as suggested.

Line 84, Introduction: The phrase “requirement for 2t-hexadecenal in BAX pore formation” mischaracterizes earlier covalent studies, which examined BAX monomer activation rather than strictly pore formation.

The Reviewer has misunderstood this sentence. The quoted phrase refers to the original Cell 2012 paper, which primarily assessed BAX activation by pore formation, not the covalent studies; the second half of the sentence describes the recent covalent studies. Nevertheless, we have changed the phrase to "the requirement for 2t-hexadecenal in BAX-mediated MOMP..." to avoid this misunderstanding.

Lines 106–107, Results: The external-treatment approach remains difficult to interpret given the compound's high intrinsic reactivity. It is unknown whether sufficient cytosolic levels can be reached to achieve a non-covalent interaction with BAX, as opposed to non-specific induction of stress. The term “validated ectopic hexadecenal” is not supported experimentally.

The term "validated ectopic hexadecenal" refers to the control mass spectrometry experiment in which we validate the molecular weight of our reagent, as previously requested in the review process (Figure S1A). There is no suggestion that "validated" refers to the results of the ectopic treatment experiment. To avoid this confusion, we have removed the word "validated".

The Reviewer maintains that we overstate the conclusions of these assays and we politely do not agree. We previously addressed this comment and remind the Reviewer that we explicitly state the limitations of this approach and its conclusions in the text. Moreover, these limitations are the rationale for directly investigating the effect of hexadecenal on BAX using model membrane systems and several other experimental approaches using BAX protein. We believe that our approaches (including limitations), interpretations, and reasoning are transparently delivered to the reader and that our conclusions evolve with each experiment over the course of the paper.

Lines 130–131, Results: The statement that the authors use LUVs “without altering the biochemistry of the LUV membrane” doesn’t really apply; LUVs are synthetic systems defined entirely by their constructed compositions.

The lipid species and ratios used to make LUVs are widely standardized in the apoptosis field as they were specifically selected to mimic the major lipids comprising the outer mitochondrial membrane (as published in Kuwana et al., 2002). Our statement that we did not deviate from the standard composition is specifically to distinguish our approach from prior works (in which hexadecenal was incorporated into the LUV formulation) and to make it clear to the reader that we are not merely replicating prior investigations. We believe this is an important distinction especially when our conclusions propose an alternative mechanism to what has previously been published. We have added the appropriate reference to contextualize our statement and the importance of LUV composition.

Lines 142–144, Results: This is a key area needing clarification. Both control compounds—hexadecenal (double bond reduced) and the hydrazone (aldehyde eliminated)—lack electrophilicity because the distinctly reactive α,β -unsaturated aldehyde moiety is disrupted in each case. Although the hydrazone species still contains a double bond, it is not “reactive” (electrophilic) without conjugation to the aldehyde. Readers will recognize that both controls remove Michael acceptor capacity. Thus, these negative results are much more consistent with loss of covalent reactivity than with selective loss of a non-covalent interaction that happens to require both of these molecular features. This is important because both compounds are treated as specificity controls across multiple assays (thermophoresis [lines 158-159], FLAMBE [lines 216-217], SEC [lines 247-248], results paragraph introduction [lines 335-336], etc.), so accurate interpretation here is especially important.

The Reviewer’s interpretation of these experiments and controls has evolved over the review process. The decision to use a hydrazone derived molecule was in response to this Reviewer, who had requested a control molecule "which maintains the double bond but removes the electrophilic aldehyde functionality". This was specifically accomplished by the 2t16-NN-EtOH hydrazone molecule; the Reviewer has now changed their interpretation, thus undermining their original concern we addressed.

In this third set of comments, the Reviewer suggests that the ability for Michael reaction requires both the aldehyde moiety and a proximal double bond, and therefore the

2t16-NN-EtOH molecule may not exhibit the same reactivity despite conservation of the double bond. This is not our area of expertise, and we cannot comment on the validity of this point. Therefore, in good faith, we added a comment discussing this possibility and that the loss-of-function of the two molecules could suggest a chemical reaction mechanism.

We remind the Reviewer that the limitation of these molecules and their conclusions is precisely why we then study a cysteine-deficient BAX mutant to demonstrate that our observed phenotype is not due to covalent modification. This critical control is what clarifies the interpretation of the modified molecules and corroborates the existence of a non-covalent interaction. In our opinion, we present no misinterpretation in subsequent assays as suggested by the Reviewer – the use of these molecules (and the cysteine-deficient mutant) is to demonstrate conservation of the phenotype between assays. In service of this goal, we have moved the 2t16 binding curve into the main figures (now Figure 2B) and put the comparison of 2t16 and hexadecanal in the supplement (now Figure S2B).

As a separate consideration, we point out that the original work studying the role of hexadecenal in BAX activation utilized cis-11-hexadecenal. While the Reviewer now suggests that the double bond must be proximal to the aldehyde moiety, it is worth noting that this is not true in cis-11-hexadecenal, yet the aldehyde activated BAX regardless. This would suggest: 1) that the original work was in fact studying the non-covalent interaction, thus supporting our conclusions; and/or 2) that the Reviewer's supposition regarding the origin of the electrophilicity is not accurate. We chose not to explore this point within the work as it was a divergence from the main biology and instead we focused on the structure-function relationship of BAX utilizing experimental and computational approaches corroborated with numerous rationally-designed loss-of-function mutations.

Line 196, Results: The distinction made regarding “direct treatment” is not meaningful, as prior studies also used direct treatment models.

In this section of the Results, we posit that it is important to remind the reader that we utilized a model in which BAX encounters the aldehyde in solution prior to membranes (i.e., when the BAX conformer resembles the structures used later in the work – namely, 1F16 and 2K7W). The sentence goes on to state that membrane incorporation methods would have hexadecenal interacting with the conformers that BAX adopts at the membrane, which include multimeric and oligomeric structures. These conformers are vastly different, and it is important for our readership to be reminded which part of the BAX activation continuum we are investigating with each experimental approach. Furthermore, this point is provided to explain our progression into a different BAX activation model system – namely, one that observes BAX activation in solution prior to membrane interactions.

Lines 254–255, Results: A concluding statement about lipid-BAX association at a specific stage of the activation pathway is made without evidence of the interaction in a cellular or cell lysate context.

The sentence in question reads: "These data collectively demonstrate that hexadecenal promotes monomeric BAX activation downstream of BCL-2 protein interactions, following activation by direct activators, and before interactions with the OMM." The data at this

point demonstrate a requirement for BIM-triggering and increased multimerization in solution prior to interactions with a membrane (Figure 4). Therefore, the referenced sentence accurately summarizes the data and conclusions at that point in the paper. There is no mention or suggestion of the larger cellular context because, as the Reviewer points out, those were not experimentally examined. This does not undermine the data or conclusions as the underlying biology of the BCL-2 family was discovered, and continues to be studied, using biochemical approaches, and our interpretation is supported by the scientific literature and current understanding of how the BCL-2 family functions. However, to avoid the possibility of inaccuracy, we changed "the OMM" to "a membrane" in case this was the focus of the Reviewer's concern.

Lines 354–355, 402–404, 433–435, Results: The authors appropriately acknowledge the limitations of rigid docking, including artifact risk and inability to determine ligand orientation. However, given these limitations, it is premature to draw a declarative conclusion that the aldehyde “resides in the depth of the BAF” without more informative structural or computational approaches. Incorporating even basic molecular dynamics simulations is widely considered an industry standard for such claims in a high-impact journal.

The Reviewer raised this comment twice before and we provided a detailed and thorough response as well as incorporating the requested modifications into the text (as noted above and in the previous two Reviewer's Response documents).

To briefly reiterate our position on this matter:

- 1. Molecular dynamics is fundamentally a theoretical computational approach, and instead we provide experimental validation through a variety of techniques and rational design of specific and selective loss-of-function mutations. Any additional insights gained by MD would be entirely speculative in nature.*
- 2. Prior, this Reviewer suggested MD for solvent sampling, model relaxation, and then docking to a representative ensemble. Critically, this would only potentially alter the BAX model, and the docking of hexadecenal would then be subsequently conducted using the same method as the current data – the MD simulation does not perform or replace the docking efforts. Given that the BAX models utilized in this work were selected from solution NMR ensembles, it is unclear how docking against a single "representative ensemble" would be fundamentally different than the deposited structures that were experimentally determined.*
- 3. Fundamentally, the addition of MD as previously suggested by this Reviewer would not change the following conclusions: i) the BAF is present in all ensembles of the structures modeled in this work (Figures S4F–G); ii) the BAF cavity is represented in several BAX structures, including crystal and solution NMR structures (Figure S10D); iii) studies mutating BAF-forming residues demonstrated a selective loss-of-function for hexadecenal while remaining sensitive to BIM activation (Figures 6G–H, S8C–F).*

Given the utilization of more than 15 techniques, approaches, and analytical tools, as well as the description of a new domain within the BCL-2 family, we vehemently disagree with the Reviewer's suggestion that our work is unfit for a high impact journal.

Lines 500–501 and 522–523, Discussion: The suggestion that prior results may reflect “the use of millimolar concentrations of 2t-hexadecenal”, coupled with the assertion that BAX is “unlikely to experience millimolar concentrations...during activation in the cytosol,” is misleading. Earlier studies also employed micromolar concentrations as well as endogenous, physiologic levels present in mitochondrial membranes. It is worth noting that covalent mechanisms generally require far lower effective concentrations than non-covalent interactions to exert biological effects in a cellular context, so the focus on millimolar levels does not meaningfully address the underlying issue.

The Reviewer has provided their opinion on this matter previously, and we reiterate that our discussion on the topic is valid and warranted when attempting to rationalize how each of the manuscripts – both prior and our own – result in divergent conclusions while not being mutually exclusive. While this Reviewer holds this opinion, it is worth noting that another reviewer (Reviewer 2) explicitly requested commentary on the difference in approaches and how it may have resulted in different findings. We believe that the Discussion is the appropriate place for these types of considerations as it is how we can best position our findings in the context of prior works, while supporting the conclusion that the different results are without conflict.

The assertion that covalent interactions require "far lower effective concentrations than non-covalent interactions to exert biological effects in a cellular context" is a mischaracterization of the point we raise in the text. In a cellular context, there are several factors that may facilitate covalent interactions (for example, enrichment and sidechain positioning at membranes), and this is precisely the point we make – that the structure of activated, membrane-associated BAX structures have exposed cysteines for targeting and likely sit at sites enriched in hexadecenal (Figure S10). By comparison, the prior work demonstrated that inactive cytosolic BAX was not modified by hexadecenal at micromolar concentrations, whereas millimolar concentrations instigated the covalent reaction. There is no disagreement here, as the point in question compares the results from experiments studying inactive, monomeric BAX in solution; we go on to posit that membrane-associated BAX would be more primed for modification, thus aligning with the approaches and results of the prior work. Additionally, we would like to point out that in both examples that the Reviewer cites, we explicitly state that we are discussing BAX in solution/cytosol.

Finally, we believe that our position is clear – in both the manuscript text and in multiple responses to the Reviewer's comments – we do not dispute the previous findings and offer consideration and key methodological differences to rationalize how each manuscript identified distinct (and non-conflicting) conclusions regarding BAX biology. We edited the text to avoid reasonable misunderstanding or misinterpretation of our logic, approach, data, or conclusions. Finally, we acknowledge that the manuscript has been improved during the revision process as we have refined our discussion and interpretations at multiple steps to appease this Reviewer's concerns.